# Diffusion Models as Cartoonists: The Curious Case of High Density Regions

**Rafał Karczewski, Markus Heinonen**
Aalto University
{rafal.karczewski, markus.o.heinonen}@aalto.fi

**Vikas Garg**
YaiYai Ltd and Aalto University
vgarg@csail.mit.edu

## Abstract

We investigate what kind of images lie in the high-density regions of diffusion models. We introduce a theoretical mode-tracking process capable of pinpointing the exact mode of the denoising distribution, and we propose a practical high-density sampler that consistently generates images of higher likelihood than usual samplers. Our empirical findings reveal the existence of significantly higher likelihood samples that typical samplers do not produce, often manifesting as cartoon-like drawings or blurry images depending on the noise level. Curiously, these patterns emerge in datasets devoid of such examples. We also present a novel approach to track sample likelihoods in diffusion SDEs, which remarkably incurs no additional computational cost. Code is available at https://github.com/Aalto-QuML/high-density-diffusion

## 1 Introduction

Recently, Karras et al. (2024a) attributed the empirical success of guided diffusion models to their ability to limit outliers, i.e. samples $\boldsymbol{x}_0 \sim p_0$ with low likelihood $p_0(\boldsymbol{x}_0)$. We argue that these models in fact also elude samples with very high likelihoods. Our assertion stems from investigating a key question: what manifests if we bias the sampler towards high-density regions of $p_0$? However, an immediate hurdle is the inability of stochastic diffusion models to track their own likelihood (Song et al., 2021c;b).

First, we show that likelihood can be tracked in diffusion models with novel augmented stochastic differential equations (SDE), which govern the evolution of a sample with its log-density $\log p_t(\boldsymbol{x}_t)$

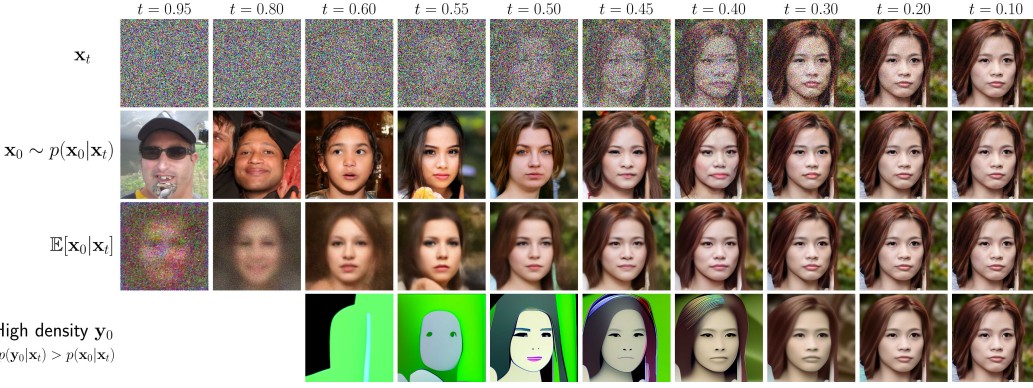

Figure 1: **High-density samples $\boldsymbol{y}_0|\boldsymbol{x}_t$ resemble cartoon drawings.** This is in contrast to regular denoising samples $\boldsymbol{x}_0 \sim p(\boldsymbol{x}_0|\boldsymbol{x}_t)$ or expectations $\mathbb{E}[\boldsymbol{x}_0|\boldsymbol{x}_t]$. The data contains no cartoons.

| Summary of our contributions | |
|---|---|
| **Augmented SDEs (section 2 + section 3)** | |
| Exact $\log p_t(x_t)$ when $\nabla \log p_t(x)$ and $p_T$ are known | subsection 2.2 |
| Estimated $\log p_t^{\text{SDE}}(x_t)$ for approximate $\nabla \log p_t(x)$ | subsection 2.3 |
| Variational gap $\log p_0^{\text{SDE}}(\boldsymbol{x}_0) - \text{ELBO}(\boldsymbol{x}_0)$ formula | Equation 13 |
| Estimation error analysis | subsection 3.2 |
| **Mode-tracking (section 4):** | |
| Evolution of the mode of $p(x_s\|x_t)$ | Theorem 5 |
| High-Density ODE approximation | Remark 1 |
| General instantaneous change of variables | Lemma 1 |
| **Diffusion probability landscape (section 5):** | |
| Unrealistic images have the highest likelihoods | Figure 1 |
| Blurring an image increases its likelihood | Figure 10 |
| Likelihood estimates correlate with PNG size | Figure 8 |

Figure 2: Our contributions.

under the optimal (unknown) model. For approximate models, we provide a formula for the bias in the log-density estimate. The evaluation of the log-density estimate comes at no additional cost, and can be used with any pretrained model without further tuning.

Then, we introduce a theoretical mode-tracking process, which finds the exact mode of the denoising distribution $p(\boldsymbol{x}_0|\boldsymbol{x}_t)$ under some technical assumptions, albeit at a high computational cost. We propose an approximative *high density sampler* whose images almost always have higher likelihood than regular samples.

We leverage these findings to give insights into the diffusion model probability landscape. We find that dramatically higher likelihood samples exist that a regular sampler never returns in practise. We observe that the high density samples tend to be (i) blank images for high noise levels, (ii) cartoon drawings for moderate noise, and (iii) blurry images for low noise (See Figure 1). *This is despite the datasets not containing any cartoon drawings.* Curiously, on FFHQ-256 we observe 97% correlation between model's likelihood estimates and the amount of information in the image.

We summarize our contributions in Figure 2.

## 2 LIKELIHOOD ESTIMATION IN SDE MODELS

We model the data distribution with a diffusion process (Song et al., 2021c), which gradually transforms the data into a Gaussian. We define a forward process $p(\boldsymbol{x}_t|\boldsymbol{x}_0) = \mathcal{N}(\boldsymbol{x}_t; \alpha_t \boldsymbol{x}_0, \sigma_t^2 \boldsymbol{I}_D)$, where $D$ is the dimensionality of the data, $\alpha_t$ and $\sigma_t$ are schedule parameters with a decreasing signal-to-noise ratio $\text{SNR}(t) = \alpha_t^2/\sigma_t^2 = e^{\lambda_t}$. Song et al. (2021c) showed an equivalence to a linear stochastic differential equation (SDE) with $\boldsymbol{x}_0 \sim p_0$ - data distribution and

$$d\boldsymbol{x}_t = f(t)\boldsymbol{x}_t dt + g(t)d\text{W}_t, \tag{1}$$

with drift $f(t) = \frac{d \log \alpha_t}{dt}$, diffusion $g^2(t) = -\frac{d\lambda_t}{dt}\sigma_t^2$, and $\{\text{W}_t\}$ a Wiener process. Most of our theoretical results also hold for non-linear SDEs such as the one proposed by Bartosh et al. (2024). We discuss the impact of the linearity of the drift in section 6 and provide the general formulas in the appendices. Anderson (1982) showed that there exists a corresponding (sharing the distribution over trajectories) *reverse-time* SDE

$$d\boldsymbol{x}_t = \Big(f(t)\boldsymbol{x}_t - g^2(t) \boxed{\nabla_{\boldsymbol{x}} \log p_t(\boldsymbol{x}_t)} \Big)dt + g(t)d\overline{\text{W}}_t, \tag{2}$$

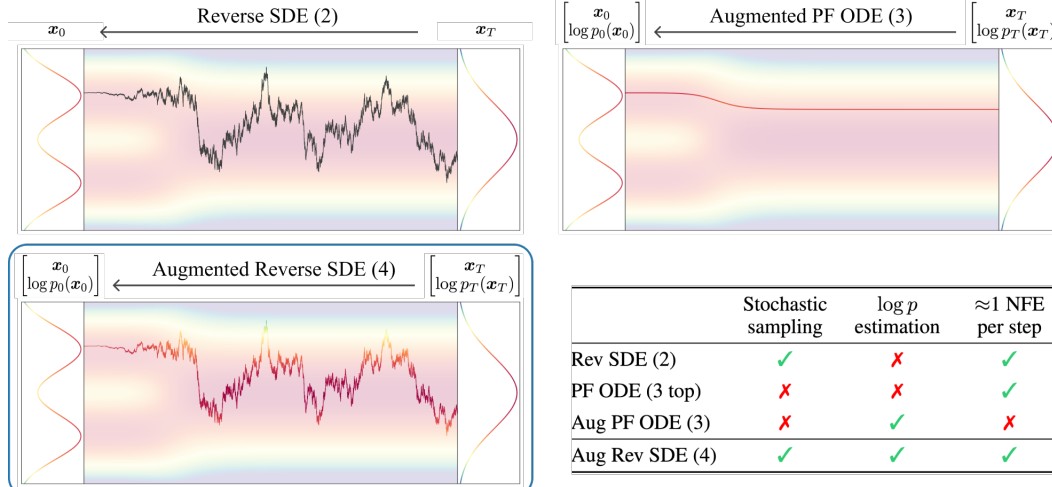

Figure 3: **Tracking stochastic sampling likelihood.** Estimation of $\log p_t(\boldsymbol{x}_t)$ (colored trajectory) for stochastic sampling via Augmented Reverse SDE (equation 4) on a Gaussian mixture with known $\nabla_{\boldsymbol{x}} \log p_t(\boldsymbol{x})$ and $p_T$. Evaluation of $d \log p_t(\boldsymbol{x}_t)$ requires only the score function.

where $p_t$ is the marginal density of the forward process (equation 1), and $\{\overline{W}_t\}$ is a Wiener process going backwards in time. Training of diffusion models centers on efficient estimation of the *score* $\nabla_{\boldsymbol{x}} \log p_t(\boldsymbol{x})$ (Hyvärinen & Dayan, 2005; Vincent, 2011; Song et al., 2020; Kingma & Gao, 2024).

## 2.1 DIFFUSION ODE

We are interested in sampling high-density regions of diffusion models and for that, we need ability to calculate the likelihood $p_0(\boldsymbol{x})$ of a sample image $\boldsymbol{x} \sim p_0$. As the reverse-SDE (equation 2) does not track the likelihood evolution $d \log p_t(\boldsymbol{x}_t)$, a typical method for obtaining sample likelihoods is instead via the Probability-Flow ODE (Chen et al., 2018; Song et al., 2021c)

$$d \begin{bmatrix} \boldsymbol{x}_t \\ \log p_t(\boldsymbol{x}_t) \end{bmatrix} = \begin{bmatrix} f(t)\boldsymbol{x}_t - \frac{1}{2}g^2(t) \, \nabla_{\boldsymbol{x}} \log p_t(\boldsymbol{x}_t) \\ -f(t)D + \frac{1}{2}g^2(t) \, \mathrm{div}_{\boldsymbol{x}} \, \nabla_{\boldsymbol{x}} \log p_t(\boldsymbol{x}_t) \end{bmatrix} dt, \tag{3}$$

where $\mathrm{div}_{\boldsymbol{x}} = \sum_i \frac{\partial}{\partial x_i}$ is the divergence operator. The PF-ODE is a continuous normalizing flow (CNF) that shares marginal distributions $p_t(\boldsymbol{x}_t)$ with both forward (1) and reverse SDE (2), when they share $p_0$ (Song et al., 2021c). The PF-ODE tracks and returns both the sample $\boldsymbol{x}_0$ and its likelihood $\log p_0(\boldsymbol{x}_0)$ from $\boldsymbol{x}_T \sim p_T$. It requires second-order derivatives $\mathrm{div} \, \nabla$, which are roughly twice as expensive as the score to evaluate (Hutchinson, 1989; Grathwohl et al., 2019).

## 2.2 AUGMENTED STOCHASTIC DYNAMICS

It has been reported that a stochastic sampler (2) has superior sample quality to the PF-ODE (3) (Song et al., 2021c; Karras et al., 2022). However, the absence of density-augmented SDEs forbids the stochastic sampler from describing the likelihood of the samples it yields (Song et al., 2021b; Lu et al., 2022; Zheng et al., 2023; Lai et al., 2023). We bridge this gap by generalizing equation 3 to account for the stochastic evolution of $\boldsymbol{x}$.

**Theorem 1** (Augmented reverse SDE). *Let $\boldsymbol{x}$ be a random process defined by equation 2. Then*

$$d \begin{bmatrix} \boldsymbol{x}_t \\ \log p_t(\boldsymbol{x}_t) \end{bmatrix} = \begin{bmatrix} f(t)\boldsymbol{x}_t - g^2(t) \, \nabla_{\boldsymbol{x}} \log p_t(\boldsymbol{x}_t) \\ -f(t)D - \frac{1}{2}g^2(t)\| \, \nabla_{\boldsymbol{x}} \log p_t(\boldsymbol{x}_t) \, \|^2 \end{bmatrix} dt + g(t) \begin{bmatrix} \boldsymbol{I}_D \\ \nabla_{\boldsymbol{x}} \log p_t(\boldsymbol{x}_t) \end{bmatrix}^T d\overline{W}_t. \tag{4}$$

The proof (Appendix E) consists of applying Itô's lemma and the Fokker-Planck equation. Note that $d \log p_t(\boldsymbol{x}_t)$ is economical to track as it only needs access to the first-order score. Figure 3

visualises a particle following a stochastic reverse trajectory while tracking its log-density. This result contributes a new useful and economical tool to generate a sample together with its density estimate, which previously was done using PF-ODE (Jing et al., 2022). Similarly to the reverse SDE, we also introduce density-tracking forward SDE.

**Theorem 2** (Augmented forward SDE). *Let $\boldsymbol{x}$ be a random process defined by equation 1. Then*

$$d \begin{bmatrix} \boldsymbol{x}_t \\ \log p_t(\boldsymbol{x}_t) \end{bmatrix} = \begin{bmatrix} f(t)\boldsymbol{x}_t \\ F(t, \boldsymbol{x}_t) \end{bmatrix} dt + g(t) \begin{bmatrix} \boldsymbol{I}_D \\ \nabla_{\boldsymbol{x}} \log p_t(\boldsymbol{x}_t)^T \end{bmatrix} d\mathbf{W}_t, \qquad (5)$$

*where*

$$F(t, \boldsymbol{x}_t) = -\operatorname{div}_{\boldsymbol{x}} \left( f(t)\boldsymbol{x}_t - g^2(t)\, \nabla_{\boldsymbol{x}} \log p_t(\boldsymbol{x}_t) \right) + \frac{1}{2} g^2(t) \| \nabla_{\boldsymbol{x}} \log p_t(\boldsymbol{x}_t) \|^2.$$

Proof is similar to the reverse case and can be found in Appendix D. The forward augmented SDE will prove useful for estimating the density of an arbitrary input point $\boldsymbol{x}$ (not necessarily sampled from the model).

Karras et al. (2022) proposed a more general forward SDE than equation 1 for which we also derive the dynamics of $\log p_t(\boldsymbol{x}_t)$ in both directions (Appendix F). Interestingly, we show that equation 1 is the only case for which the dynamics of $\log p_t(\boldsymbol{x}_t)$ in the reverse direction do not involve any higher order derivatives of $\log p_t(\boldsymbol{x}_t)$.

We can now track the likelihood for forward and reverse SDEs under the theoretical true $\nabla_{\boldsymbol{x}} \log p_t(\boldsymbol{x})$. However, under score approximation $\boldsymbol{s}(t, \boldsymbol{x}) \approx \nabla_{\boldsymbol{x}} \log p_t(\boldsymbol{x})$ the density solutions $\log p_t(\boldsymbol{x}_t)$ of Theorems 1 and 2 become biased, which we study next.

## 2.3 APPROXIMATE REVERSE DYNAMICS

We can substitute the approximate score $\boldsymbol{s}(t, \boldsymbol{x}) \approx \nabla_{\boldsymbol{x}} \log p_t(\boldsymbol{x})$ and assume $p_T^{\mathrm{ODE}} = p_T^{\mathrm{SDE}} = \mathcal{N}(\boldsymbol{0}, \sigma_T^2 \boldsymbol{I}_D)$. The resulting SDE and ODE models are no longer equivalent, i.e. $p_t^{\mathrm{ODE}} \neq p_t^{\mathrm{SDE}}$ (Song et al., 2021b; Lu et al., 2022). The PF-ODE becomes

$$d \begin{bmatrix} \boldsymbol{x}_t \\ \log p_t^{\mathrm{ODE}}(\boldsymbol{x}_t) \end{bmatrix} = \begin{bmatrix} f(t)\boldsymbol{x}_t - \frac{1}{2}g^2(t)\, \boldsymbol{s}(t, \boldsymbol{x}_t) \\ -f(t)D + \frac{1}{2}g^2(t)\, \operatorname{div}_{\boldsymbol{x}}\, \boldsymbol{s}(t, \boldsymbol{x}_t) \end{bmatrix} dt, \qquad (6)$$

which can track its marginal log-density $\log p_t^{\mathrm{ODE}}(\boldsymbol{x}_t)$ exactly. However, substituting the true score with $\boldsymbol{s}(t, \boldsymbol{x})$ in the augmented reverse SDE 4 incurs estimation error in $\log p_0^{\mathrm{SDE}}(\boldsymbol{x}_0)$. We characterise the error formally in a novel theorem:

**Theorem 3** (Approximate Augmented Reverse SDE). *Let $\boldsymbol{s}(t, \boldsymbol{x})$ be an approximation of the score function. Let $\boldsymbol{x}_T \sim p_T$ and define an auxiliary process $r$ starting at $r_T = \log p_T^{\mathrm{SDE}}(\boldsymbol{x}_T)$. If*

$$d \begin{bmatrix} \boldsymbol{x}_t \\ r_t \end{bmatrix} = \begin{bmatrix} f(t)\boldsymbol{x}_t - g^2(t)\, \boldsymbol{s}(t, \boldsymbol{x}_t) \\ -f(t)D - \frac{1}{2}g^2(t)\| \boldsymbol{s}(t, \boldsymbol{x}_t) \|^2 \end{bmatrix} dt + g(t) \begin{bmatrix} \boldsymbol{I}_D \\ \boldsymbol{s}(t, \boldsymbol{x}_t)^T \end{bmatrix} d\overline{\mathbf{W}}_t, \qquad (7)$$

*then $\boldsymbol{x}_0 \sim p_0^{\mathrm{SDE}}(\boldsymbol{x}_0)$ and*

$$r_0 = \log p_0^{\mathrm{SDE}}(\boldsymbol{x}_0) + \mathrm{X}, \qquad (8)$$

*where $\mathrm{X}$ is a random variable such that the bias of $r_0$ is given by*

$$\mathbb{E}\mathrm{X} = \frac{T}{2} \mathbb{E}_{t \sim \mathcal{U}(0,T), \boldsymbol{x}_t \sim p_t^{\mathrm{SDE}}(\boldsymbol{x}_t)} \left[ g^2(t) \| \boldsymbol{s}(t, \boldsymbol{x}_t) - \nabla_{\boldsymbol{x}} \log p_t^{\mathrm{SDE}}(\boldsymbol{x}_t) \|^2 \right] \geq 0. \qquad (9)$$

See Appendix G for the proof and the definition of X. Intuitively, the true density evolution of $p_t^{\mathrm{SDE}}(\boldsymbol{x}_t)$ is induced by $d\boldsymbol{x}_t$ in equation 7, and the auxiliary variable $r_t$ does not follow it perfectly. Since the equation 9 has an intractable score, we seek more practical alternatives to measuring the accuracy of $r_0$ in the next Section.

> The new augmented SDE of equation 7 can be used with any score-based model without further tuning to provide sample likelihood estimates $\log p_0^{\mathrm{SDE}}(\boldsymbol{x}_0)$ for no extra cost.

## 3 ACCURACY OF THE DENSITY ESTIMATION

We analyse the accuracy of the $\log p_0^{\text{SDE}}(\boldsymbol{x}_0)$ estimates in equation 8. We begin with the approximate forward dynamics which will provide a method for bounding the estimation error of $r_0$.

### 3.1 APPROXIMATE FORWARD DYNAMICS

In contrast to Theorem 3, when we replace the true score function with $\boldsymbol{s}$ in the forward direction, we underestimate $\log p_0^{\text{SDE}}(\boldsymbol{x}_0)$ on average.

**Theorem 4** (Approximate Augmented Forward SDE). *Let $\boldsymbol{s}(t, \boldsymbol{x}_t)$ be the model approximating the score function and $\boldsymbol{x}_0 \in \mathbb{R}^D$ given. Define an auxiliary process $\omega$ starting at $\omega_0 = 0$. If*

$$d \begin{bmatrix} \boldsymbol{x}_t \\ \omega_t \end{bmatrix} = \begin{bmatrix} f(t)\boldsymbol{x}_t \\ -f(t)D + g^2(t)\left(\frac{1}{2}\| \boldsymbol{s}(t,\boldsymbol{x}_t) \|^2 + \operatorname{div}_{\boldsymbol{x}} \boldsymbol{s}(t,\boldsymbol{x}_t)\right) \end{bmatrix} dt + g(t) \begin{bmatrix} \boldsymbol{I}_D \\ \boldsymbol{s}(t,\boldsymbol{x}_t)^T \end{bmatrix} d\mathrm{W}_t. \tag{10}$$

*Then*

$$\omega_T = \log p_T^{\text{SDE}}(\boldsymbol{x}_T) - \log p_0^{\text{SDE}}(\boldsymbol{x}_0) + \mathrm{Y}_{\boldsymbol{x}_0}, \tag{11}$$

*where $\mathrm{Y}_{\boldsymbol{x}_0}$ is a random variable such that*

$$\mathbb{E}\mathrm{Y}_{\boldsymbol{x}_0} = \frac{T}{2}\mathbb{E}_{t \sim \mathcal{U}(0,T)}\mathbb{E}_{\boldsymbol{x}_t \sim p(\boldsymbol{x}_t|\boldsymbol{x}_0)}g^2(t)\| \boldsymbol{s}(t,\boldsymbol{x}_t) - \nabla_{\boldsymbol{x}}\log p_t^{\text{SDE}}(\boldsymbol{x}_t) \|^2 \geq 0. \tag{12}$$

See Appendix G for the proof and the definition of $\mathrm{Y}_{\boldsymbol{x}_0}$. Due to the drift div operator, the evaluation of $d\omega_t$ is computationally comparable to $d\log p_t^{\text{ODE}}(\boldsymbol{x}_t)$. Interestingly, Theorem 4 completes a known lower bound into a novel identity,

$$\log p_0^{\text{SDE}}(\boldsymbol{x}_0) = \underbrace{\mathbb{E}\mathrm{Y}_{\boldsymbol{x}_0}}_{\geq 0} \underbrace{-\frac{e^{\lambda_{\min}}}{2}\|\boldsymbol{x}_0\|^2 + \frac{T}{2}\mathbb{E}_{t,\boldsymbol{\varepsilon}}\left[-\frac{d\lambda_t}{dt}\|\sigma_t \, \boldsymbol{s}(t,\alpha_t\boldsymbol{x}_0 + \sigma_t\boldsymbol{\varepsilon}) + \boldsymbol{\varepsilon}\|^2\right] + C}_{\text{ELBO}(\boldsymbol{x}_0) \text{ (Song et al., 2021b; Kingma et al., 2021)}}, \tag{13}$$

where $t \sim \mathcal{U}(0,T)$, $\boldsymbol{\varepsilon} \sim \mathcal{N}(\boldsymbol{0}, \boldsymbol{I}_D)$ and $C = -\frac{D}{2}\left(1 + \log(2\pi\sigma_0^2)\right)$ (see Corollary 1). $\text{ELBO}(\boldsymbol{x})$ is the standard tool for estimating the SDE's model likelihood of an arbitrary point $\boldsymbol{x}$.

### 3.2 ESTIMATION BIAS OF $\log p_0^{\text{SDE}}$

We are interested in using the stochastic sampler to obtain high-quality sample-density pairs $(\boldsymbol{x}_0, \log p_t^{\text{SDE}}(\boldsymbol{x}_0))$. We have now discussed two ways of estimating $\log p_0^{\text{SDE}}(\boldsymbol{x}_0)$: with $r_0$ (equation 7) and $\text{ELBO}(\boldsymbol{x}_0)$ (equation 13). Their estimation errors $\varepsilon_r(r_0, \boldsymbol{x}_0) = r_0 - \log p_0^{\text{SDE}}(\boldsymbol{x}_0)$ and $\varepsilon_{\text{ELBO}}(\boldsymbol{x}_0) = \log p_0^{\text{SDE}}(\boldsymbol{x}_0) - \text{ELBO}(\boldsymbol{x}_0)$ are intractable to estimate due to the presence of unknown score $\nabla \log p_t^{\text{SDE}}(\boldsymbol{x}_t)$.

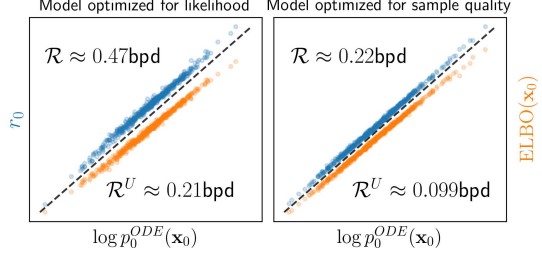

Figure 4: $r_0 > \log p_0^{\text{ODE}}(\boldsymbol{x}_0) > \text{ELBO}(\boldsymbol{x}_0)$ correlate strongly.

However, we can provide an upper bound on the bias of both estimators,

$$\underbrace{\mathbb{E}_{\boldsymbol{x}_0 \sim p_0^{\text{SDE}}(\boldsymbol{x}_0)}[\varepsilon_{\text{ELBO}}(\boldsymbol{x}_0)]}_{\geq 0 \text{ (equation 12)}} + \underbrace{\mathbb{E}_{(\boldsymbol{x}_0, r_0)}[\varepsilon_r(r_0, \boldsymbol{x}_0)]}_{\geq 0 \text{ (equation 9)}} = \underbrace{\mathbb{E}_{(\boldsymbol{x}_0, r_0)}\left[r_0 - \text{ELBO}(\boldsymbol{x}_0)\right]}_{\mathcal{R}(\boldsymbol{s}) \text{ (tractable)}}. \tag{14}$$

We can thus sample $(\boldsymbol{x}_0, r_0)$ using equation 7 and the average difference $r_0 - \text{ELBO}(\boldsymbol{x}_0)$ gives an upper bound on the bias of both $r_0$ and $\text{ELBO}(\boldsymbol{x}_0)$. This can be useful to assess the accuracy of both $r_0$ and $\text{ELBO}(\boldsymbol{x}_0)$ as density estimates for stochastic samples $\boldsymbol{x}_0$. Furthermore, we can estimate how much $p_0^{\text{SDE}}$ differs from $p_0^{\text{ODE}}$ by providing bounds for $\text{KL}[p_0^{\text{SDE}}||p_0^{\text{ODE}}]$

$$\underbrace{\mathbb{E}_{\boldsymbol{x}_0 \sim p^{\text{SDE}}(\boldsymbol{x}_0)}\left[\text{ELBO}(\boldsymbol{x}_0) - \log p_0^{\text{ODE}}(\boldsymbol{x}_0)\right]}_{\mathcal{R}^L(\boldsymbol{s}) \text{ (tractable)}} \leq \text{KL}\left[p_0^{\text{SDE}}||p_0^{\text{ODE}}\right] \leq \underbrace{\mathbb{E}_{(\boldsymbol{x}_0, r_0)}\left[r_0 - \log p_0^{\text{ODE}}(\boldsymbol{x}_0)\right]}_{\mathcal{R}^U(\boldsymbol{s}) \text{ (tractable)}}. \tag{15}$$

$\mathcal{R}^U$ and $\mathcal{R}^L$ are novel practical tools for measuring the difference between $p_0^{\text{SDE}}$ and $p_0^{\text{ODE}}$. As a demonstration, we train two versions of a diffusion model on CIFAR-10 (Krizhevsky et al., 2009), one with maximum likelihood training (Kingma et al., 2021; Song et al., 2021b) and one optimized for sample quality (Kingma & Gao, 2024). Please see Appendix M for implementation details. We then generate 512 samples of $(\boldsymbol{x}_0, r_0)$ with equation 7 and for each $\boldsymbol{x}_0$ we estimated $\log p_0^{\text{ODE}}(\boldsymbol{x}_0)$ using equation 6 and $\text{ELBO}(\boldsymbol{x}_0)$ using equation 13.

For both models we found very high correlations between $r_0$ and $\log p_0^{\text{ODE}}(\boldsymbol{x}_0)$ at 0.996 and 0.999. Surprisingly, it is the model optimized for sample quality that yielded both a higher correlation and lower values of $\mathcal{R}^U(\boldsymbol{s})$ and $\mathcal{R}(\boldsymbol{s})$ suggesting a smaller difference between $p_0^{\text{SDE}}$ and $p_0^{\text{ODE}}$ and a lower bias of $\log p_0^{\text{SDE}}(\boldsymbol{x}_0)$ estimation (Figure 4). We also estimated $\mathcal{R}^L(\boldsymbol{s})$, which was negative for both models which is a trivial lower bound for $\text{KL}[p_0^{\text{SDE}}||p_0^{\text{ODE}}] \geq 0$.

> The density estimates $r_0 \approx \log p_0^{\text{SDE}}(\boldsymbol{x}_0)$ from equation 7 empirically form an upper bound on $\log p_0^{\text{ODE}}(\boldsymbol{x}_0)$ and correlate with it very strongly ($> 0.99$).

# 4 MODE ESTIMATION

Many diffusion models are trained by implicitly maximizing weighted ELBO (Kingma & Gao, 2024), and can be interpreted as likelihood-based models. Karras et al. (2024a) emphasized the role of likelihood by explaining the empirical success of guided diffusion models by their ability to limit low density samples $p_0(\boldsymbol{x}_0)$. We explore this idea further by asking: *what if we aim for samples with the highest $p_0(\boldsymbol{x}_0)$?*

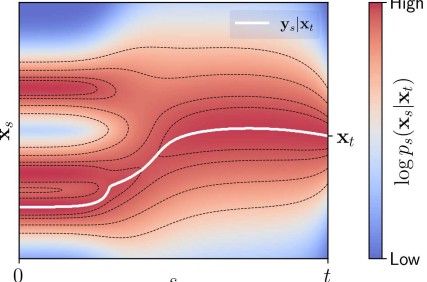

Figure 5: Equation 16 accurately recovers the mode-tracking curve.

## 4.1 MODE-TRACKING CURVE

We first go back to assuming a perfectly estimated score function $\nabla_{\boldsymbol{x}} \log p_t(\boldsymbol{x})$. Let $p(\boldsymbol{x}_0|\boldsymbol{x}_t)$ denote the denoising distribution given by solving the reverse SDE (equation 2) from $\boldsymbol{x}_t$ to $\boldsymbol{x}_0$. If we knew how to estimate the *denoising mode* $\boldsymbol{y}_0(\boldsymbol{x}_t) = \arg\max p(\boldsymbol{x}_0|\boldsymbol{x}_t)$, we could bias the sampler towards higher likelihood regions by first taking a regular noisy samples $\boldsymbol{x}_t \sim p_t(\boldsymbol{x}_t)$ at various times $t$, and pushing them to deterministic modes $\boldsymbol{y}_0(\boldsymbol{x}_t)$.

We approach this by asking a seemingly more difficult question: can we find a *mode-tracking curve*, i.e. $\boldsymbol{y}_s$ such that $p(\boldsymbol{y}_s|\boldsymbol{x}_t) = \max_{\boldsymbol{x}_s} p(\boldsymbol{x}_s|\boldsymbol{x}_t)$ for all $s < t$ (See Figure 5)? We show that whenever such a smooth curve exists it is given by an ODE:

**Theorem 5** (Mode-tracking ODE). *Let* $t \in (0, T]$ *and* $\boldsymbol{x}_t \in \mathbb{R}^D$ *a noisy sample. If there exists a smooth curve* $s \mapsto \boldsymbol{y}_s$ *such that* $p(\boldsymbol{y}_s|\boldsymbol{x}_t) = \max_{\boldsymbol{x}_s} p(\boldsymbol{x}_s|\boldsymbol{x}_t)$, *then* $\boldsymbol{y}_t = \boldsymbol{x}_t$ *and for* $s < t$

$$\frac{d}{ds}\boldsymbol{y}_s = f(s)\boldsymbol{y}_s - g^2(s) \, \nabla_{\boldsymbol{y}} \log p_s(\boldsymbol{y}_s) \, - \frac{1}{2}g^2(s) \, \boldsymbol{A}(s, \boldsymbol{y}_s)^{-1} \, \nabla_{\boldsymbol{y}}\Delta_{\boldsymbol{y}} \log p_s(\boldsymbol{y}_s) , \qquad (16)$$

*where* $\boldsymbol{A}(s, \boldsymbol{y}) = \left(\nabla_{\boldsymbol{y}}^2 \log p_s(\boldsymbol{y}) - \psi(s)\boldsymbol{I}_D\right)$, $\psi(s) = \frac{1}{\sigma_s^2}\frac{e^{\lambda_t}}{e^{\lambda_s} - e^{\lambda_t}}$, *and* $\Delta_{\boldsymbol{y}} = \sum_i \frac{\partial^2}{\partial y_i^2}$ *is the Laplace operator. In particular:*

$$p(\boldsymbol{y}_0|\boldsymbol{x}_t) = \max_{\boldsymbol{x}_0} p(\boldsymbol{x}_0|\boldsymbol{x}_t). \qquad (17)$$

The proof and the statement without assuming invertibility of $\boldsymbol{A}$ can be found in Appendix H. We visualize on a mixture of 1D Gaussians in Figure 5 how the solution of the mode tracking ODE (equation 16) correctly recovers the denoising mode curve, i.e. the mode of $p(\boldsymbol{x}_s|\boldsymbol{x}_t)$ for all $s < t$.

## 4.2 HIGH-DENSITY SEEKING ODE

Using Theorem 5 in high-dimensional data is problematic. A smooth *mode-tracking* curve needs to exist, which is difficult to verify and need not hold (Appendix J). Moreover, equation 16 requires computing and inverting the Hessian and estimating third-order derivatives, which is prohibitively

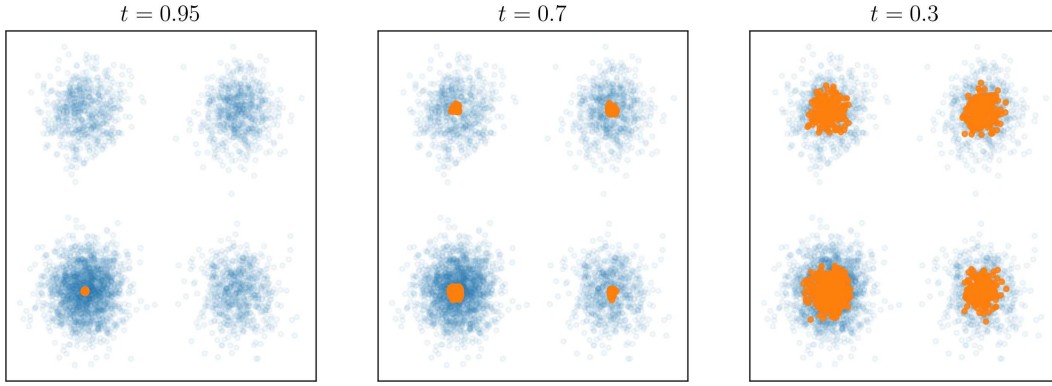

Figure 6: **Equation 16 enables controlling the likelihood-diversity tradeoff.** Blue dots are samples from the data distribution, i.e. a mixture of four Gaussians with the bottom left component having the highest weight. Orange samples are generated with Algorithm 1.

expensive (Meng et al., 2021). Please see Appendix K for more details. We propose an approximation of equation 16 by noting its high-order terms disappear under Gaussian data.

**Remark 1** (High-density ODE or HD-ODE)**.** *If $p_0$ is Gaussian, then equation 16 becomes*

$$d\boldsymbol{y}_s = \Big( f(s)\boldsymbol{y}_s - g^2(s)\, \boxed{\nabla_{\boldsymbol{x}} \log p_s(\boldsymbol{y}_s)}\, \Big)\, ds, \qquad (18)$$

*i.e. the drift term of reverse SDE equation 2. If $\boldsymbol{y}_t = \boldsymbol{x}_t$ and equation 18 holds for $s < t$, then*

$$\boldsymbol{y}_0 = \arg\max_{\boldsymbol{x}_0} p(\boldsymbol{x}_0|\boldsymbol{x}_t) + \mathcal{O}(e^{-\lambda_{\max}}). \qquad (19)$$

See Appendix I for the proof. Even though equation 18 finds the mode of $p(\boldsymbol{x}_0|\boldsymbol{x}_t)$ only in the Gaussian case (for sufficiently small $e^{-\lambda_{\max}}$), we will empirically show that even for non-Gaussian data it finds points with much higher likelihoods than regular samples.

In Algorithm 1 we propose a novel high-density sampler that uses equation 18 to bias the sampling towards higher likelihood regions. We choose a threshold time $t$ and sample $\boldsymbol{x}_t \sim p_t(\boldsymbol{x}_t)$, and then estimate $\boldsymbol{y}_0(\boldsymbol{x}_t)$ by solving equation 18 from $s = t$ to $s = 0$. Figure 6 shows how the threshold controls the tradeoff between likelihood and diversity in a toy mixture of Gaussians.

---

**Algorithm 1** High density sampling

1: **Input:** Threshold $t \in (0, T]$
2: Initial $\boldsymbol{x}_T \sim \mathcal{N}(\boldsymbol{0}, \sigma_T^2 \boldsymbol{I}_D)$
3: Sample $\boldsymbol{x}_t \sim p_t(\boldsymbol{x}_t)$     eq. 6 or 7
4: $\boldsymbol{y}_0 \leftarrow$ HD-ODE$(t, 0, \boldsymbol{x}_t)$     eq. 18
5: Return $\boldsymbol{y}_0$

---

### 4.3  ESTIMATING MODE DENSITIES FOR REAL-WORLD DATA

We will next discuss how to evaluate the density of the high-density samples $\boldsymbol{y}_0|\boldsymbol{x}_t$ and regular samples $\boldsymbol{x}_0|\boldsymbol{x}_t$ to empirically show that $p(\boldsymbol{y}_0|\boldsymbol{x}_t) > p(\boldsymbol{x}_0|\boldsymbol{x}_t)$ for $\boldsymbol{x}_0 \sim p(\boldsymbol{x}_0|\boldsymbol{x}_t)$. For any $\boldsymbol{x}_0$, the denoising likelihood can be decomposed using Bayes' rule

$$\log p_{0|t}(\boldsymbol{x}_0|\boldsymbol{x}_t) = \underbrace{\log p_{t|0}(\boldsymbol{x}_t|\boldsymbol{x}_0)}_{\mathcal{N}(\boldsymbol{x}_t|\alpha_t\boldsymbol{x}_0, \sigma_t^2\boldsymbol{I}_D)} + \log p_0(\boldsymbol{x}_0) - \log p_t(\boldsymbol{x}_t). \qquad (20)$$

For $\boldsymbol{x}_0 \sim p(\boldsymbol{x}_0|\boldsymbol{x}_t)$ we can use equation 4 from $(\boldsymbol{x}_t, r_t = 0)$ to obtain $r_0 = \log p_0(\boldsymbol{x}_0) - \log p_t(\boldsymbol{x}_t)$. To estimate $\log p(\boldsymbol{y}_0|\boldsymbol{x}_t)$ we could use the PF-ODE, but that is inefficient.[1] Instead, we can obtain $d \log p_s(\boldsymbol{y}_s)$ under HD-ODE with a convenient, and to our knowledge novel, lemma:

**Lemma 1** (General instantaneous change of variables)**.** *Consider a CNF given by $d\boldsymbol{x}_t = f_1(t, \boldsymbol{x}_t)dt$ with prior $p_T$ and marginal distributions $p_t$, and a particle following some different dynamical system $d\boldsymbol{z}_t = f_2(t, \boldsymbol{z}_t)dt$. Then, if $f_1$ and $f_2$ are uniformly Lipschitz in the second argument and continuous in the first, we have:*

$$\frac{d \log p_t(\boldsymbol{z}_t)}{dt} = -\,\mathrm{div}_{\boldsymbol{z}}\, f_1(t, \boldsymbol{z}_t) + \big(f_2(t, \boldsymbol{z}_t) - f_1(t, \boldsymbol{z}_t)\big)^T \nabla_{\boldsymbol{z}} \log p_t(\boldsymbol{z}_t). \qquad (21)$$

---

[1] On top of solving HD-ODE we would need to solve the augmented PF-ODE twice: once for $\boldsymbol{y}_t$ from $t$ to $T$ and once for $\boldsymbol{y}_0$ from $0$ to $T$. Instead, we perform one *augmented* ((18) + (22)) HD-ODE solve from $t$ to $0$.

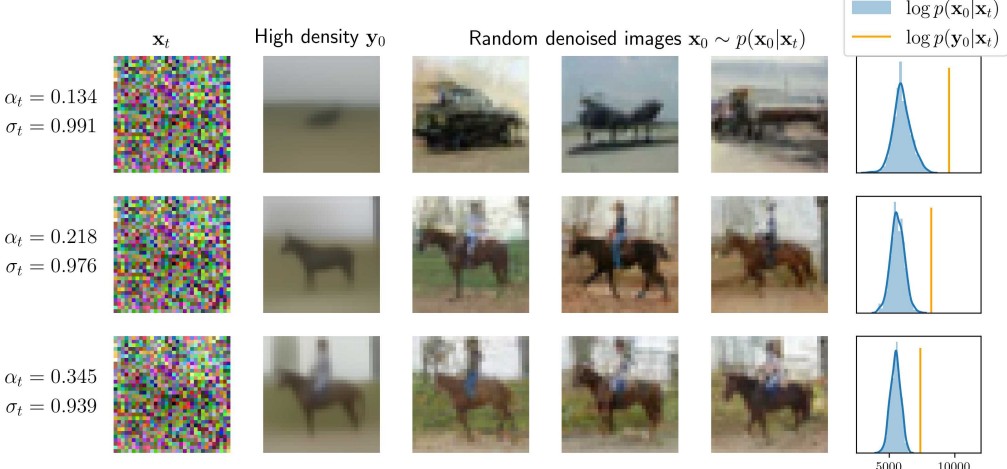

Figure 7: **Algorithm 1 generates images with higher likelihoods than regular samples.** For different noise levels $t$, compare the high density $\boldsymbol{y}_0$ (equation 18) with random samples $\boldsymbol{x}_0$ on CIFAR-10. We find that the percentage of samples with higher likelihoods than $\boldsymbol{y}_0$ is 0.

The proof is in Appendix L. When $f_1 = f_2$ we recover the standard formula Chen et al. (2018). By setting $f_1$ =PF-ODE (3) and $f_2$ =HD-ODE (18) we augment the HD-ODE with its density evolution

$$\frac{d \log p_s(\boldsymbol{y}_s)}{ds} = -f(s)D + \frac{1}{2}g^2(s)\,\text{div}_{\boldsymbol{y}}\;\nabla_{\boldsymbol{y}} \log p_s(\boldsymbol{y}_s)\;-\frac{1}{2}g^2(s)\big\|\;\nabla_{\boldsymbol{y}} \log p_s(\boldsymbol{y}_s)\;\big\|^2. \quad (22)$$

We trained a diffusion model on CIFAR-10 and generated $\boldsymbol{x}_t$ for different noise levels $t$ and compared the high-density samples $\boldsymbol{y}_0|\boldsymbol{x}_t$ (Algorithm 1) against 512 regular samples $\boldsymbol{x}_0 \sim p(\boldsymbol{x}_0|\boldsymbol{x}_t)$. We found that $p(\boldsymbol{y}_0|\boldsymbol{x}_t) > p(\boldsymbol{x}_0|\boldsymbol{x}_t)$ for all samples across different noise levels $t$ (See Figure 7).

Additionally, we compared the likelihoods of regular samples and the ones obtained with algorithm 1 for different models, and values of the threshold parameter $t$. We found that algorithm 1 samples have higher likelihoods than regular samples in all cases. For details, please refer to Appendix N.

## 5 HIGH-RESOLUTION DIFFUSION PROBABILITY LANDSCAPE

We demonstrated that algorithm 1 can generate images with much higher likelihoods than regular samples. However, after visually inspecting the high density samples $\boldsymbol{y}_0$ in Figure 7 we found that they correspond to blurry images with much less detail than regular samples.

To gain more insight we analyze high-density samples on higher-resolution diffusion models.[2] We found that the samples with the highest likelihood were blurry images. Surprisingly, for higher values of $t$, the samples were unnatural cartoons but still received higher likelihoods than regular samples. The training datasets do not contain any cartoon images. See Figure 9. A similar phenomenon occurs for latent diffusion models Appendix O.

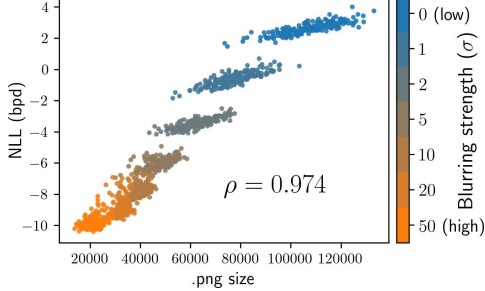

Figure 8: **Density estimates correlate with information content (`.jpg` size)**

Inspired by the empirical observation that the highest likelihood generated samples are blurry images we performed the following two experiments. First, we add different amounts of blur to FFHQ-256 test images and measure the likelihood of the distorted image. We found that blurring always increases the likelihood and that the increase is proportional to the strength of blurring (Figure 10).

---

[2]We used FFHQ-256 and Churches-256 models from `github.com/yang-song/score_sde_pytorch` and ImageNet-64 from `github.com/NVlabs/edm`

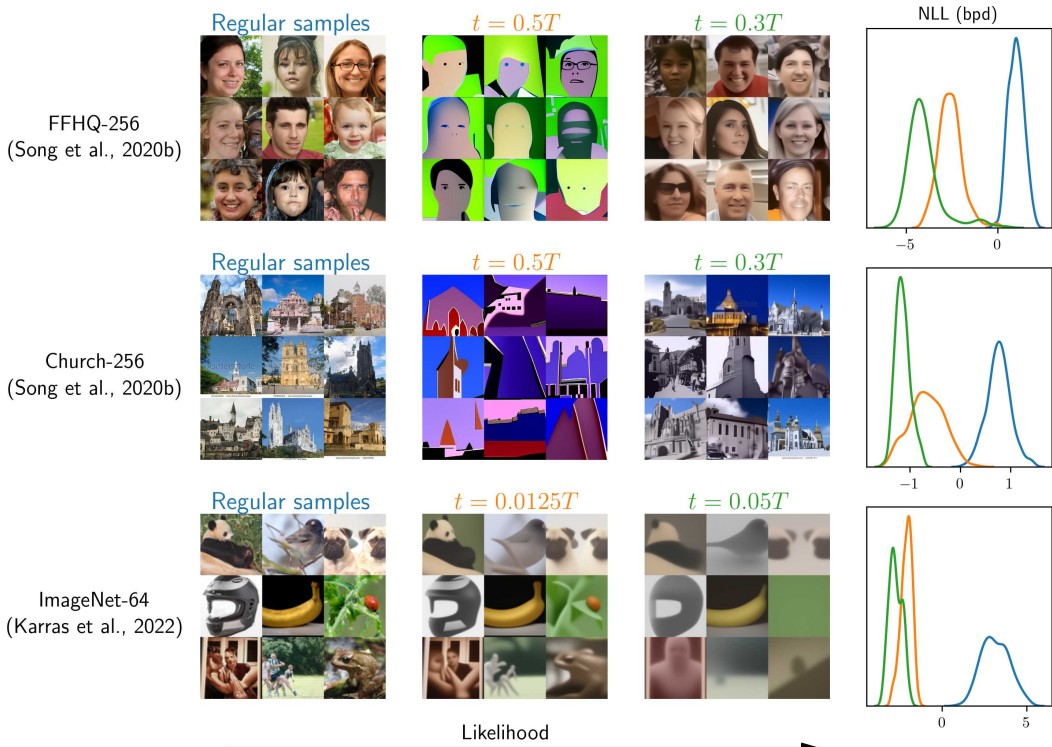

Figure 9: **Diffusion models assign highest likelihoods to unrealistic images.** Left: High-density samples generated with Algorithm 1 for various values of the threshold parameter $t$. Right: distributions of negative log-likelihood (in bits-per-dim). Across different models and datasets, Algorithm 1 finds unnatural images, which have higher densities than regular samples.

Second, we compare the model's likelihood estimation with the image's file size after PNG compression. The smaller the PNG file size, the less information in the image. For FFHQ-256, we found a 97% correlation between $\log p_0(\boldsymbol{x}_0)$ and the amount of information in an image (Figure 8). This hints at why cartoons and blurry images have the highest densities (Appendix P). We used 192 samples for each of 7 blur strengths $\sigma \in \{0, 1, 2, 5, 10, 20, 50\}$, resulting in 7·192=1344 images.

## 6 DISCUSSION

**Variance of $\log p_0^{\mathrm{SDE}}(\boldsymbol{x}_0)$ estimate.** In section 3 we discussed the accuracy of $r_0$, our novel estimate of $\log p_0^{\mathrm{SDE}}(\boldsymbol{x}_0)$. Specifically, we provided tools to estimate the bound of its bias for stochastic samples. Based on the empirically measured correlation between $r_0$ and $\log p_0^{\mathrm{ODE}}(\boldsymbol{x}_0)$ at over 0.99,

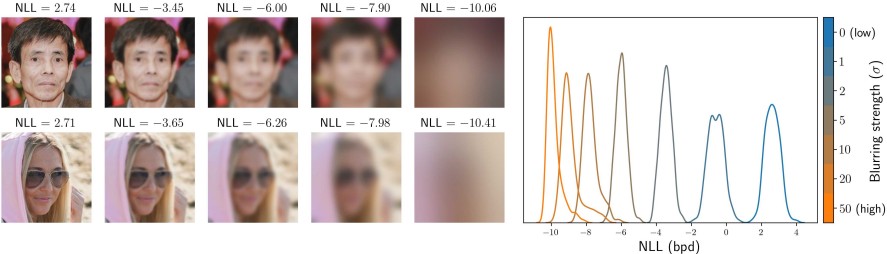

Figure 10: **Blurring increases likelihood.** Left: Two FFHQ-256 images with different amounts of blur and corresponding negative loglikelihoods (NLL). Right: Distributions of NLL for different amounts of added blur ($\sigma \in \{0, 1, 2, 5, 10, 20, 50\}$, 192 samples each).

we hypothesize that the variance of $r_0 - \log p_0^{\text{SDE}}(\boldsymbol{x}_0)$ is low whenever $\boldsymbol{s}(t, \boldsymbol{x}) \approx \nabla \log p_t^{\text{SDE}}(\boldsymbol{x})$. However, we do not provide theoretical guarantees, or empirical estimates.

$p_0^{\text{SDE}}$ **vs** $p_0^{\text{ODE}}$. In equation 15 we show that $\mathcal{R}^U(\boldsymbol{s}) = \mathbb{E}_{(\boldsymbol{x}_0, r_0)}[r_0 - \log p_0^{\text{ODE}}(\boldsymbol{x}_0)]$ is an upper bound on $\text{KL}[p_0^{\text{SDE}} \| p_0^{\text{ODE}}] \geq 0$. In particular, for $(\boldsymbol{x}_0, r_0)$ sampled with equation 7 we must have *on average* $r_0 \geq \log p_0^{\text{ODE}}(\boldsymbol{x}_0)$. However, for two different models, we found that $r_0 \geq \log p_0^{\text{ODE}}(\boldsymbol{x}_0)$ holds *for every sample*. We hypothesize this is a more widespread phenomenon, but do not prove it.

**Towards exact estimates of** $\log p_0^{\text{SDE}}(\boldsymbol{x}_0)$. We showed in equation 9 that the bias of $r_0$ is given by $\mathbb{E}(r_0 - \log p_0^{\text{SDE}}(\boldsymbol{x}_0)) \propto \mathbb{E}_{t, \boldsymbol{x}_t} g^2(t) \| \boldsymbol{s}(t, \boldsymbol{x}_t) - \nabla_{\boldsymbol{x}} \log p_t^{\text{SDE}}(\boldsymbol{x}_t) \|^2$ for $\boldsymbol{x}_t \sim p_t^{\text{SDE}}(\boldsymbol{x}_t)$. Similarly, equation 12 shows that $\log p_0^{\text{SDE}}(\boldsymbol{x}_0) - \text{ELBO}(\boldsymbol{x}_0) \propto \mathbb{E}_{t, \boldsymbol{x}_t} g^2(t) \| \boldsymbol{s}(t, \boldsymbol{x}_t) - \nabla_{\boldsymbol{x}} \log p_t^{\text{SDE}}(\boldsymbol{x}_t) \|^2$ for $\boldsymbol{x}_t \sim p(\boldsymbol{x}_t | \boldsymbol{x}_0)$. Both these errors could then be reduced to zero if $\boldsymbol{s}(t, \boldsymbol{x}) = \nabla_{\boldsymbol{x}} \log p_t^{\text{SDE}}(\boldsymbol{x})$ for all $t, \boldsymbol{x}$. However, for SDEs with linear drift (equation 1), this can only happen if $p_t^{\text{SDE}}$ is Gaussian for all $t$ (Proposition B.1 in Lu et al. (2022)). This is because an SDE with linear drift cannot transform a non-Gaussian $p_0$ into a Gaussian in finite time $T$.

To unlock exact likelihood estimation in diffusion SDEs, non-linear drift is necessary, such as the one proposed in Bartosh et al. (2024). There it is possible to have $p_T$ Gaussian for finite $T$ and $\boldsymbol{s}(t, \boldsymbol{x}) = \nabla_{\boldsymbol{x}} \log p_t^{\text{SDE}}(\boldsymbol{x})$ for all $t, \boldsymbol{x}$, in which case both $r_0$ and $\text{ELBO}(\boldsymbol{x}_0)$ become exact (Theorem 7 and Proposition 3).

## 7 RELATED WORK

Park et al. (2024) show that the temperature in the diffusion samplers relates to the sample density and can result in cartoon-like samples. See Appendix Q for a discussion on cartoon generation.

**Likelihood estimation for diffusion models.** As discussed in subsection 2.3, there is a distinction between diffusion ODEs and SDEs. For diffusion SDEs, only lower bounds on likelihood are reported (Ho et al., 2020; Vahdat et al., 2021; Nichol & Dhariwal, 2021; Huang et al., 2021; Kingma et al., 2021; Kim et al., 2022). Exact likelihoods, on the other hand, are available for diffusion ODEs (Song et al., 2021c;a;b; Dockhorn et al., 2022) and some works explicitly optimize for ODE likelihood (Lu et al., 2022; Zheng et al., 2023; Lai et al., 2023). For a comprehensive survey, we refer to Yang et al. (2023). We provide a novel tool for estimating the likelihood of diffusion SDE samples. Concurrent work (Skreta et al., 2025) independently discovered Theorem 1 and Lemma 1 without considering the forward process (Theorem 2) or imperfect score functions (subsection 2.3).

**Typicality vs likelihood.** Theis et al. (2016) observed that likelihoods do not correlate with image quality. Furthermore, deep generative models can assign higher likelihoods to out-of-distribution (OOD) data than training data (Choi et al., 2018; Nalisnick et al., 2019a; Kirichenko et al., 2020) and perform poorly at OOD detection. Nalisnick et al. (2019b); Choi et al. (2018) analyze this phenomenon with *typicality*, arguing that typical samples do not have the highest likelihoods. Ben-Hamu et al. (2024) observed that explicit distortion of an image, like inserting a gray patch, increases the likelihood assigned by a flow-based model. Contrary to these reports, we explicitly study regions of highest likelihood and shed light on the probability landscape of diffusion models.

## 8 CONCLUSION

We provide novel tools for estimating the likelihood for Diffusion SDE samples. Additionally, we theoretically and empirically analyze the estimation error and discuss when exact likelihood estimation for diffusion SDEs might be possible. These tools, combined with a theoretical mode-seeking analysis, allowed us to study high-density regions of diffusion models. We made a surprising observation that unnatural and blurry images occupy the highest-density regions of diffusion models. While Karras et al. (2024a) argued that avoiding low-density regions is crucial for the success of diffusion models, our analysis reveals that high-density regions should also be avoided in high-quality image generation. We discuss the limitations of this work in Appendix R.

This work not only enhances the understanding of diffusion model probability landscapes but also opens avenues for improved sample generation strategies.

ACKNOWLEDGMENTS

This work was supported by the Finnish Center for Artificial Intelligence (FCAI) under Flagship R5 (award 15011052). VG also acknowledges the support from Saab-WASP (grant 411025), Academy of Finland (grant 342077), and the Jane and Aatos Erkko Foundation (grant 7001703). RK thanks Paulina Karczewska for her help with preparing figures.

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

## A NOTATION AND ASSUMPTIONS

Table 1: Summary of notation and abbreviations.

| Notation | Description |
|---|---|
| $D$ | dimensionality of the data |
| $\|\boldsymbol{x}\|^2 = \sum_i x_i^2$ | Squared Euclidean norm of $\boldsymbol{x} \in \mathbb{R}^D$ |
| $\nabla_{\boldsymbol{x}}$, or $\frac{\partial}{\partial \boldsymbol{x}}$ | gradient w.r.t $\boldsymbol{x}$ |
| $\mathrm{div}_{\boldsymbol{x}} = \sum_i \frac{\partial}{\partial x_i}$ | divergence operator w.r.t $\boldsymbol{x}$ |
| $\mathbb{E}_{\boldsymbol{x} \sim p} f(\boldsymbol{x})$ | expectation $= \int f(\boldsymbol{x}) p(\boldsymbol{x}) d\boldsymbol{x}$ |
| $\boldsymbol{I}_D$ | $D$-dimensional identity matrix |
| $D_{KL}(p\|\|q)$ | Kullback-Leibler divergence $= \mathbb{E}_{\boldsymbol{x} \sim p} (\log p(\boldsymbol{x}) - \log q(\boldsymbol{x}))$ |
| $f \in \mathcal{C}^k$ | function $f$ has continuous derivatives of up to order $k$ |
| $\Delta_{\boldsymbol{x}} = \sum_i \frac{\partial^2}{\partial x_i^2}$ | Laplace operator |
| $p_0$ | data distribution |
| $p_t$ | marginal distribution of a process defined by Forward SDE equation 1 |
| $p(\boldsymbol{x}_t\|\boldsymbol{x}_0) = p_{t\|0}(\boldsymbol{x}_t\|\boldsymbol{x}_0)$ | Forward transition distribution. For linear drift SDE (equation 1): $p(\boldsymbol{x}_t\|\boldsymbol{x}_0) = \mathcal{N}(\boldsymbol{x}_t\|\alpha_t \boldsymbol{x}_0, \sigma_t^2 \boldsymbol{I}_D)$ |
| $p(\boldsymbol{x}_0\|\boldsymbol{x}_t) = p_{0\|t}(\boldsymbol{x}_0\|\boldsymbol{x}_t)$ | Denoising distribution $p_{0\|t}(\boldsymbol{x}_0\|\boldsymbol{x}_t) = p_{t\|0}(\boldsymbol{x}_t\|\boldsymbol{x}_0)p_0(\boldsymbol{x}_0)/p_t(\boldsymbol{x}_t)$ |
| $\mathrm{SNR}(t) = \frac{\alpha_t^2}{\sigma_t^2}$ | signal to noise ratio |
| $\lambda_t = \log \mathrm{SNR}(t)$ | log signal to noise ratio |
| $\nabla_{\boldsymbol{x}} \log p_t(\boldsymbol{x})$ | "True" score function |
| $\boldsymbol{s}(t, \boldsymbol{x})$ | "approximate" score function |
| $p_t^{\mathrm{SDE}}$ | marginal distribution of a process defined by the approximate reverse SDE equation 7 |
| $p_t^{\mathrm{ODE}}$ | marginal distribution of a process defined by the approximate ODE equation 6 |
| $\mathrm{W}$ | Wiener process running forward in time |
| $\overline{\mathrm{W}}$ | Wiener process running backwards in time |

In equation 1, the drift term is linear in $\boldsymbol{x}$, which corresponds to most commonly used SDEs, because it admits Gaussian transition kernels $p_{t\|s}$ for $s < t$ (Song et al., 2021c;b; Kingma et al., 2021; Kingma & Gao, 2024) and covers many common implementations of diffusion models (Ho et al., 2020; Song et al., 2021a; Nichol & Dhariwal, 2021; Dhariwal & Nichol, 2021; Karras et al., 2024b).

However, more general SDEs have been proposed that do not assume linear drift (Zhang & Chen, 2021; Bartosh et al., 2024):

$$d\boldsymbol{x}_t = f(t, \boldsymbol{x}_t)dt + g(t)d\mathrm{W}_t \tag{23}$$

with $f$ and $g$ satisfying assumptions below. We define the approximate reverse SDE in the general case

$$d\boldsymbol{x}_t = \left(f(t, \boldsymbol{x}_t) - g^2(t)\boldsymbol{s}(t, \boldsymbol{x}_t)\right) dt + g(t)d\overline{\mathrm{W}}_t. \tag{24}$$

**In our theorems in section 2 and section 3, we do not assume the linearity of the drift and provide more general formulas. However, results in section 4 only hold for linear drift SDE.** We follow Song et al. (2021b) and Lu et al. (2022) and make the following assumptions in our proofs to ensure existence of reverse-time SDEs and correctness of integration by parts.

1. $p_0 \in \mathcal{C}^3$ and $\mathbb{E}_{\boldsymbol{x} \sim p_0}[\|\boldsymbol{x}\|^2] < \infty$.

2. $\forall t \in [0, T] : f(\cdot, t) \in \mathcal{C}^2$. And $\exists C > 0, \forall \boldsymbol{x} \in \mathbb{R}^D, t \in [0, T] : \|f(t, \boldsymbol{x})\|_2 \le C(1 + \|\boldsymbol{x}\|_2)$.

3. $\exists C > 0, \forall \boldsymbol{x}, \boldsymbol{y} \in \mathbb{R}^D : \|f(t, \boldsymbol{x}) - f(t, \boldsymbol{y})\|_2 \le C\|\boldsymbol{x} - \boldsymbol{y}\|_2$.

4. $g \in \mathcal{C}$ and $\forall t \in [0, T], |g(t)| > 0$.

5. For any open bounded set $\mathcal{O}$, $\int_0^T \int_{\mathcal{O}} \left( \|p_t(\boldsymbol{x})\|^2 + D \cdot g(t)^2 \|\nabla p_t(x)\|^2 d\boldsymbol{x} \right) dt < \infty$.

6. $\exists C > 0, \forall \boldsymbol{x} \in \mathbb{R}^D, t \in [0, T] : \|\nabla p_t(x)\|^2 \le C(1 + \|\boldsymbol{x}\|)$.

7. $\exists C > 0, \forall \boldsymbol{x}, \boldsymbol{y} \in \mathbb{R}^D : \|\nabla \log p_t(\boldsymbol{x}) - \nabla \log p_t(\boldsymbol{y})\| \le C\|\boldsymbol{x} - \boldsymbol{y}\|$.

8. $\exists C > 0, \forall \boldsymbol{x} \in \mathbb{R}^D, t \in [0, T] : \|\boldsymbol{s}(t, \boldsymbol{x})\| \le C(1 + \|\boldsymbol{x}\|)$.

9. $\exists C > 0, \forall \boldsymbol{x}, \boldsymbol{y} \in \mathbb{R}^D : \|\boldsymbol{s}(t, \boldsymbol{x}) - \boldsymbol{s}(t, \boldsymbol{y})\| \le C\|\boldsymbol{x} - \boldsymbol{y}\|$.

10. Novikov's condition: $\mathbb{E}\left[ \exp\left( \frac{1}{2} \int_0^T \|\nabla \log p_t(\boldsymbol{x}) - \boldsymbol{s}(t, \boldsymbol{x})\|^2 dt \right) \right] < \infty$.

11. $\forall t \in [0, T], \exists k > 0 : p_t(\boldsymbol{x}), p_t^{\mathrm{SDE}}(\boldsymbol{x}), p_t^{\mathrm{ODE}}(\boldsymbol{x}) \in O(e^{-\|\boldsymbol{x}\|^k})$ as $\|\boldsymbol{x}\| \to \infty$.

Additionally, we assume

12. $\forall \boldsymbol{x}_0 \in \mathbb{R}^D : \mathbb{E}_{\boldsymbol{x} \sim \nu(\boldsymbol{x}_0)} \left( \int_0^T g^2(t) \|\nabla_{\boldsymbol{x}} \log p_t^{\mathrm{SDE}}(\boldsymbol{x}_t)\|^2 dt \right) < \infty$, where $\nu(\boldsymbol{x}_0)$ is the path measure of equation 1 starting at $\boldsymbol{x}_0$.

13. $\mathbb{E}_{\boldsymbol{x} \sim \nu^{\mathrm{SDE}}} \left( \int_0^T g^2(t) \|\nabla_{\boldsymbol{x}} \log p_t^{\mathrm{SDE}}(\boldsymbol{x}_t) - \boldsymbol{s}(t, \boldsymbol{x}_t)\|^2 dt \right) < \infty$, where $\nu^{\mathrm{SDE}}$ is the path measure of equation 7.

14. $\forall \boldsymbol{x}_0 \in \mathbb{R}^D, \forall t \in [0, T], \exists k > 0 : p_{t|0}(\boldsymbol{x}|\boldsymbol{x}_0) \in O(e^{-\|\boldsymbol{x}\|^k})$ as $\|\boldsymbol{x}\| \to \infty$. This trivially holds for the linear drift SDE (equation 1), where $p_{t|0}(\boldsymbol{x}|\boldsymbol{x}_0)$ is Gaussian.

## B  FOKKER PLANCK EQUATION

A useful tool in some of the proofs is the Fokker-Planck equation (Fokker, 1914; Planck, 1917; Øksendal & Øksendal, 2003; Särkkä & Solin, 2019), a partial differential equation (PDE) governing the evolution of the marginal density of a diffusion process. For a process described in Equation 23, the evolution of marginal density is described by the Fokker-Planck Equation:

$$\frac{\partial}{\partial t} p_t(\boldsymbol{x}) = - \operatorname{div}\left( f(t, \boldsymbol{x}) p_t(\boldsymbol{x}) \right) + \frac{1}{2} g^2(t) \Delta_{\boldsymbol{x}} p_t(\boldsymbol{x}), \tag{25}$$

where $p_t$ is the marginal density at time $t$, which holds for all $t \in (0, T)$ and $\boldsymbol{x} \in \mathbb{R}^D$. Equivalently,

$$\begin{aligned} \frac{\partial}{\partial t} \log p_t(\boldsymbol{x}) = &- \operatorname{div}\left( f(t, \boldsymbol{x}) \right) + \frac{1}{2} g^2(t) \Delta_{\boldsymbol{x}} \log p_t(\boldsymbol{x}) \\ &- \nabla_{\boldsymbol{x}} \log p_t(\boldsymbol{x})^T \left( f(t, \boldsymbol{x}) - \frac{1}{2} g^2(t) \nabla_{\boldsymbol{x}} \log p_t(\boldsymbol{x}) \right). \end{aligned} \tag{26}$$

## C  ITÔ'S LEMMA

The main tool for studying the dynamics of log-density of stochastic processes we will use in our proofs is Itô's lemma (Itô, 1951), which states that for a stochastic process

$$d\boldsymbol{x}_t = f(t, \boldsymbol{x}_t) dt + g(t) d\mathrm{W}_t \tag{27}$$

and a smooth function $h : \mathbb{R} \times \mathbb{R}^D \to \mathbb{R}$ it holds that

$$dh(t, \boldsymbol{x}_t) = \left( \frac{\partial}{\partial t} h(t, \boldsymbol{x}_t) + \frac{\partial}{\partial \boldsymbol{x}} h(t, \boldsymbol{x}_t)^T f(t, \boldsymbol{x}_t) + \frac{1}{2} g^2(t) \Delta_{\boldsymbol{x}} h(t, \boldsymbol{x}_t) \right) dt + g(t) \frac{\partial}{\partial \boldsymbol{x}} h(t, \boldsymbol{x}_t) d\mathrm{W}_t. \tag{28}$$

A more general version of Itô's lemma holds, which does not assume isotropic diffusion, but we do not need it in our proofs.

## D  PROOF OF THEOREM 2

**Theorem 2** (Augmented forward SDE). *Let $\boldsymbol{x}$ be a random process defined by equation 1. Then*

$$d \begin{bmatrix} \boldsymbol{x}_t \\ \log p_t(\boldsymbol{x}_t) \end{bmatrix} = \begin{bmatrix} f(t)\boldsymbol{x}_t \\ F(t, \boldsymbol{x}_t) \end{bmatrix} dt + g(t) \begin{bmatrix} \boldsymbol{I}_D \\ \nabla_{\boldsymbol{x}} \log p_t(\boldsymbol{x}_t) \end{bmatrix}^T d\mathrm{W}_t, \tag{5}$$

*where*

$$F(t, \boldsymbol{x}_t) = -\operatorname{div}_{\boldsymbol{x}} \left( f(t)\boldsymbol{x}_t - g^2(t) \nabla_{\boldsymbol{x}} \log p_t(\boldsymbol{x}_t) \right) + \frac{1}{2}g^2(t)\| \nabla_{\boldsymbol{x}} \log p_t(\boldsymbol{x}_t) \|^2.$$

*For the non-linear drift (equation 23), we have (the difference highlighted in* blue*)*

$$F(t, \boldsymbol{x}_t) = -\operatorname{div}_{\boldsymbol{x}} \left( f(t, \boldsymbol{x}_t) - g^2(t) \nabla_{\boldsymbol{x}} \log p_t(\boldsymbol{x}_t) \right) + \frac{1}{2}g^2(t)\| \nabla_{\boldsymbol{x}} \log p_t(\boldsymbol{x}_t) \|^2$$

*Proof.* We will apply Itô's lemma (Appendix C) to $h(t, \boldsymbol{x}) := \log p_t(\boldsymbol{x})$. From Equation 26, we have

$$\frac{\partial}{\partial t} h(t, \boldsymbol{x}) = -\operatorname{div}_{\boldsymbol{x}} f(t, \boldsymbol{x}) + \frac{1}{2}g^2(t)\Delta_{\boldsymbol{x}} \log p_t(\boldsymbol{x}) - \nabla_{\boldsymbol{x}} \log p_t(\boldsymbol{x})^T \left( f(t, \boldsymbol{x}) - \frac{1}{2}g^2(t)\nabla_{\boldsymbol{x}} \log p_t(\boldsymbol{x}) \right) \tag{29}$$

and

$$\frac{\partial}{\partial t} h(t, \boldsymbol{x}_t) + \frac{\partial}{\partial \boldsymbol{x}} h(t, \boldsymbol{x}_t)^T f(t, \boldsymbol{x}_t) + \frac{1}{2}g^2(t)\Delta_{\boldsymbol{x}} h(t, \boldsymbol{x}_t)$$

$$= -\operatorname{div}_{\boldsymbol{x}} f(t, \boldsymbol{x}) + \frac{1}{2}g^2(t)\Delta_{\boldsymbol{x}} \log p_t(\boldsymbol{x}) - \nabla_{\boldsymbol{x}} \log p_t(\boldsymbol{x})^T \left( \cancel{f(t, \boldsymbol{x})} - \frac{1}{2}g^2(t)\nabla_{\boldsymbol{x}} \log p_t(\boldsymbol{x}) \right)$$

$$+ \cancel{\nabla_{\boldsymbol{x}} \log p_t(\boldsymbol{x})^T f(t, \boldsymbol{x}_t)} + \frac{1}{2}g^2(t)\Delta_{\boldsymbol{x}} \log p_t(\boldsymbol{x}_t)$$

$$= -\operatorname{div}_{\boldsymbol{x}} f(t, \boldsymbol{x}_t) + \frac{1}{2}g^2(t)\|\nabla_{\boldsymbol{x}} \log p_t(\boldsymbol{x}_t)\|^2 + g^2(t)\Delta_{\boldsymbol{x}} \log p_t(\boldsymbol{x}_t)$$

$$= -\operatorname{div}_{\boldsymbol{x}} \left( f(t, \boldsymbol{x}_t) - g^2(t)\nabla_{\boldsymbol{x}} \log p_t(\boldsymbol{x}_t) \right) + \frac{1}{2}g^2(t)\|\nabla_{\boldsymbol{x}} \log p_t(\boldsymbol{x}_t)\|^2. \tag{30}$$

Finally, we get

$$d \log p_t(\boldsymbol{x}_t) = \left( -\operatorname{div}_{\boldsymbol{x}} \left( f(t, \boldsymbol{x}_t) - g^2(t)\nabla_{\boldsymbol{x}} \log p_t(\boldsymbol{x}_t) \right) + \frac{1}{2}g^2(t)\|\nabla_{\boldsymbol{x}} \log p_t(\boldsymbol{x}_t)\|^2 \right) dt$$

$$+ \nabla_{\boldsymbol{x}} \log p_t(\boldsymbol{x}_t)^T d\mathrm{W}_t. \tag{31}$$

$\square$

## E  PROOF OF THEOREM 1

**Theorem 1** (Augmented reverse SDE). *Let $\boldsymbol{x}$ be a random process defined by equation 2. Then*

$$d \begin{bmatrix} \boldsymbol{x}_t \\ \log p_t(\boldsymbol{x}_t) \end{bmatrix} = \begin{bmatrix} f(t)\boldsymbol{x}_t - g^2(t) \nabla_{\boldsymbol{x}} \log p_t(\boldsymbol{x}_t) \\ -f(t)D - \frac{1}{2}g^2(t)\| \nabla_{\boldsymbol{x}} \log p_t(\boldsymbol{x}_t) \|^2 \end{bmatrix} dt + g(t) \begin{bmatrix} \boldsymbol{I}_D \\ \nabla_{\boldsymbol{x}} \log p_t(\boldsymbol{x}_t) \end{bmatrix}^T d\overline{\mathrm{W}}_t. \tag{4}$$

*For the non-linear drift (equation 23), we have (the difference highlighted in* blue*)*

$$d \begin{bmatrix} \boldsymbol{x}_t \\ \log p_t(\boldsymbol{x}_t) \end{bmatrix} = \begin{bmatrix} f(t, \boldsymbol{x}_t) - g^2(t) \nabla_{\boldsymbol{x}} \log p_t(\boldsymbol{x}_t) \\ -\operatorname{div}_{\boldsymbol{x}} f(t, \boldsymbol{x}_t) - \frac{1}{2}g^2(t)\| \nabla_{\boldsymbol{x}} \log p_t(\boldsymbol{x}_t) \|^2 \end{bmatrix} dt + g(t) \begin{bmatrix} \boldsymbol{I}_D \\ \nabla_{\boldsymbol{x}} \log p_t(\boldsymbol{x}_t) \end{bmatrix}^T d\overline{\mathrm{W}}_t.$$

*Proof.* The reverse SDE (Equation 2) can be equivalently written, as

$$d\boldsymbol{x}_t = \underbrace{\left(-f(T-t,\boldsymbol{x}_t) + g^2(T-t)\nabla_{\boldsymbol{x}}\log p_{T-t}(\boldsymbol{x}_t)\right)}_{=:\mu(t,\boldsymbol{x}_t)} dt + g(T-t)d\mathrm{W}_t \quad (32)$$

for a forward running brownian motion W and positive $dt$. We will apply Itô's lemma (Appendix C) to $g(t,\boldsymbol{x}) = \log p_{T-t}(\boldsymbol{x})$. Since forward and reverse SDEs share marginals we can use Equation 29:

$$\frac{\partial}{\partial t}g(t,\boldsymbol{x}) = \mathrm{div}_{\boldsymbol{x}}(f(T-t,\boldsymbol{x})) - \frac{1}{2}g^2(T-t)\Delta_{\boldsymbol{x}}\log p_{T-t}(\boldsymbol{x})$$
$$+ \nabla_{\boldsymbol{x}}\log p_{T-t}(\boldsymbol{x})^T\left(f(T-t,\boldsymbol{x}) - \frac{1}{2}g^2(T-t)\nabla_{\boldsymbol{x}}\log p_{T-t}(\boldsymbol{x})\right) \quad (33)$$

and

$$\frac{\partial}{\partial t}g(t,\boldsymbol{x}_t) + \frac{\partial}{\partial\boldsymbol{x}}g(t,\boldsymbol{x}_t)^T\mu(t,\boldsymbol{x}_t) + \frac{1}{2}g^2(T-t)\Delta_{\boldsymbol{x}}g(t,\boldsymbol{x}_t)$$
$$= \mathrm{div}_{\boldsymbol{x}}(f(T-t,\boldsymbol{x})) - \frac{1}{2}g^2(T-t)\Delta_{\boldsymbol{x}}\log p_{T-t}(\boldsymbol{x}_t)$$
$$+ \nabla_{\boldsymbol{x}}\log p_{T-t}(\boldsymbol{x}_t)^T\left(f(T-t,\boldsymbol{x}_t) - \frac{1}{2}g^2(T-t)\nabla_{\boldsymbol{x}}\log p_{T-t}(\boldsymbol{x}_t)\right)$$
$$- \nabla_{\boldsymbol{x}}\log p_{T-t}(\boldsymbol{x}_t)^T\left(f(T-t,\boldsymbol{x}_t) - g^2(T-t)\nabla_{\boldsymbol{x}}\log p_{T-t}(\boldsymbol{x}_t)\right)$$
$$+ \frac{1}{2}g^2(T-t)\Delta_{\boldsymbol{x}}\log p_{T-t}(\boldsymbol{x}_t)$$
$$= \mathrm{div}_{\boldsymbol{x}}(f(T-t,\boldsymbol{x}_t)) + \frac{1}{2}g^2(T-t)\|\nabla_{\boldsymbol{x}}\log p_{T-t}(\boldsymbol{x}_t)\|^2. \quad (34)$$

Remarkably, the terms involving the higher order derivatives: $\Delta_{\boldsymbol{x}}\log p_{T-t}(\boldsymbol{x}_t)$ cancel out. Thus, we have

$$d\log p_{T-t}(\boldsymbol{x}_t) = \left(\mathrm{div}_{\boldsymbol{x}}(f(T-t,\boldsymbol{x}_t)) + \frac{1}{2}g^2(T-t)\|\nabla_{\boldsymbol{x}}\log p_{T-t}(\boldsymbol{x}_t)\|^2\right)dt$$
$$+ g(T-t)\nabla_{\boldsymbol{x}}\log p_{T-t}(\boldsymbol{x}_t)^T d\mathrm{W}_t, \quad (35)$$

which can equivalently be written as

$$d\log p_t(\boldsymbol{x}_t) = \left(-\mathrm{div}_{\boldsymbol{x}}(f(t,\boldsymbol{x}_t)) - \frac{1}{2}g^2(t)\|\nabla_{\boldsymbol{x}}\log p_t(\boldsymbol{x}_t)\|^2\right)dt + g(t)\nabla_{\boldsymbol{x}}\log p_t(\boldsymbol{x}_t)^T d\overline{\mathrm{W}}_t, \quad (36)$$

where $\overline{\mathrm{W}}$ is running backwards in time and $dt$ is negative. $\qquad\square$

## F  GENERAL SDEs

Karras et al. (2022) showed that there is a more general SDE formulation than Equation 23, which can be interpreted as a continuum between the PF-ODE and SDE formulations. Specifically, they showed that for any choice of $\beta : \mathbb{R}_+ \to \mathbb{R}_+$ the SDE

$$d\boldsymbol{x}_t = \left(f(t,\boldsymbol{x}_t) - \left(\frac{1}{2} - \beta(t)\right)g^2(t)\nabla_{\boldsymbol{x}}\log p_t(\boldsymbol{x}_t)\right)dt + \sqrt{2\beta(t)}g(t)d\mathrm{W}_t \quad (37)$$

has the same marginals as Equation 23 and it has a reverse-time SDE

$$d\boldsymbol{x}_t = \left(f(t,\boldsymbol{x}_t) - \left(\frac{1}{2} + \beta(t)\right)g^2(t)\nabla_{\boldsymbol{x}}\log p_t(\boldsymbol{x}_t)\right)dt + \sqrt{2\beta(t)}g(t)d\overline{\mathrm{W}}_t. \quad (38)$$

One can therefore replace the original model ($\beta \equiv \frac{1}{2}$) with any choice of non-negative $\beta$. We now derive the augmented dynamics of $\log p_t(\boldsymbol{x}_t)$ for any $\beta$.

### F.1 General Forward Augmented Dynamics

**Proposition 1.** *For $\boldsymbol{x}$ following Equation 37 we have*

$$d \log p_t(\boldsymbol{x}_t) = \left( -\mathrm{div}_{\boldsymbol{x}} f(t, \boldsymbol{x}_t) + \left( \frac{1}{2} + \beta(t) \right) g^2(t) \Delta_{\boldsymbol{x}} \log p_t(\boldsymbol{x}_t) + \beta(t) g^2(t) \|\nabla_{\boldsymbol{x}} \log p_t(\boldsymbol{x}_t)\|^2 \right) dt$$
$$+ \sqrt{2\beta(t)} g(t) \nabla_{\boldsymbol{x}} \log p_t(\boldsymbol{x}_t)^T d\mathrm{W}_t.$$

$$(39)$$

Note that since $\beta(t) \geq 0$, the higher order term $\Delta_{\boldsymbol{x}} \log p_t(\boldsymbol{x}_t)$ is always non-zero for any $\beta$.

*Proof.* We proceed in the same way as the proof of Theorem 2 and apply Itô's lemma (Appendix C) to $h(t, \boldsymbol{x}) = \log p_t(\boldsymbol{x})$. Since Equation 37 shares marginals with Equation 23 we can reuse the derivation of $\frac{\partial}{\partial t} h(t, \boldsymbol{x})$ from Equation 29:

$$\frac{\partial}{\partial t} h(t, \boldsymbol{x}) = -\mathrm{div}_{\boldsymbol{x}} f(t, \boldsymbol{x}) + \frac{1}{2} g^2(t) \Delta_{\boldsymbol{x}} \log p_t(\boldsymbol{x}) - \nabla_{\boldsymbol{x}} \log p_t(\boldsymbol{x})^T \left( f(t, \boldsymbol{x}) - \frac{1}{2} g^2(t) \nabla_{\boldsymbol{x}} \log p_t(\boldsymbol{x}) \right).$$

Therefore for $\boldsymbol{x}$ following Equation 37:

$$d \log p_t(\boldsymbol{x}_t) = \left( \frac{\partial}{\partial t} h(t, \boldsymbol{x}_t) + \nabla_{\boldsymbol{x}} \log p_t(\boldsymbol{x}_t)^T \left( f(t, \boldsymbol{x}_t) - \left( \frac{1}{2} - \beta(t) \right) g^2(t) \nabla_{\boldsymbol{x}} \log p_t(\boldsymbol{x}_t) \right) \right.$$
$$\left. + \beta(t) g^2(t) \Delta \log p_t(\boldsymbol{x}_t) \right) dt + \sqrt{2\beta(t)} g(t) \nabla_{\boldsymbol{x}} \log p_t(\boldsymbol{x}_t)^T d\mathrm{W}_t$$
$$= \left( -\mathrm{div}_{\boldsymbol{x}} f(t, \boldsymbol{x}_t) + \left( \frac{1}{2} + \beta(t) \right) g^2(t) \Delta_{\boldsymbol{x}} \log p_t(\boldsymbol{x}_t) + \beta(t) g^2(t) \|\nabla_{\boldsymbol{x}} \log p_t(\boldsymbol{x}_t)\|^2 \right) dt$$
$$+ \sqrt{2\beta(t)} g(t) \nabla_{\boldsymbol{x}} \log p_t(\boldsymbol{x}_t)^T d\mathrm{W}_t.$$

$$(40)$$

$\square$

### F.2 General Reverse Augmented Dynamics

**Proposition 2.** *For $\boldsymbol{x}$ following Equation 38, we have*

$$d \log p_t(\boldsymbol{x}_t) = - \left( \mathrm{div}_{\boldsymbol{x}} f(t, \boldsymbol{x}_t) + \left( \beta(t) - \frac{1}{2} \right) g^2(t) \Delta_{\boldsymbol{x}} \log p_t(\boldsymbol{x}_t) + \beta(t) g^2(t) \|\nabla_{\boldsymbol{x}} \log p_t(\boldsymbol{x}_t)\|^2 \right) dt$$
$$+ \sqrt{2\beta(t)} g(t) \nabla_{\boldsymbol{x}} \log p_t(\boldsymbol{x}_t)^T d\overline{\mathrm{W}}_t$$

$$(41)$$

Note that $\beta \equiv \frac{1}{2}$ (corresponding to Equation 2) is the only choice for which the higher order term involving $\Delta_{\boldsymbol{x}} \log p_t(\boldsymbol{x})$ disappears.

*Proof.* Similarly to the proof of Theorem 1 rewrite the general reverse SDE with positive $dt$ and W going forward in time

$$d\boldsymbol{x}_t = - \left( f(T - t, \boldsymbol{x}_t) - \left( \frac{1}{2} + \beta(T - t) \right) g^2(T - t) \nabla_{\boldsymbol{x}} \log p_{T-t}(\boldsymbol{x}_t) \right) dt$$
$$+ \sqrt{2\beta(T - t)} g(T - t) d\mathrm{W}_t$$

$$(42)$$

We apply Itô's lemma to $g(t, x) = \log p_{T-t}(\boldsymbol{x})$:

$$
\begin{aligned}
dg(t, \boldsymbol{x}_t) = & \frac{\partial}{\partial t} g(t, \boldsymbol{x}_t) dt \\
& - \nabla_{\boldsymbol{x}} g(t, \boldsymbol{x}_t)^T \left( f(T-t, \boldsymbol{x}_t) - \left( \frac{1}{2} + \beta(T-t) \right) g^2(T-t) \nabla_{\boldsymbol{x}} \log p_{T-t}(\boldsymbol{x}_t) \right) dt \\
& + \beta(T-t) g^2(T-t) \Delta_{\boldsymbol{x}} g(t, \boldsymbol{x}_t) dt \\
& + \sqrt{2\beta(T-t)} g(T-t) \nabla_{\boldsymbol{x}} \log p_{T-t}(\boldsymbol{x}_t)^T d\mathbf{W}_t \\
= & \left( \operatorname{div}_{\boldsymbol{x}} f(T-t, \boldsymbol{x}_t) + \left( \beta(T-t) - \frac{1}{2} \right) g^2(T-t) \Delta_{\boldsymbol{x}} \log p_{T-t}(\boldsymbol{x}_t) \right) dt \\
& + \beta(T-t) g^2(T-t) \| \nabla_{\boldsymbol{x}} \log p_{T-t}(\boldsymbol{x}_t) \|^2 dt \\
& + \sqrt{2\beta(T-t)} g(T-t) \nabla_{\boldsymbol{x}} \log p_{T-t}(\boldsymbol{x}_t)^T d\mathbf{W}_t,
\end{aligned}
\tag{43}
$$

which we rewrite equivalently with $dt < 0$ and $\overline{\mathbf{W}}$ running backward in time to obtain Equation 41. $\square$

## G  APPROXIMATE MODEL DYNAMICS

Analogously to Theorem 2 and Theorem 1 we can derive the dynamics of $\log p_t^{\mathrm{SDE}}(\boldsymbol{x})$.

**Theorem 6** (Approximate augmented forward SDE). *Let $\boldsymbol{x}$ be a random process defined by Equation 23. Then*

$$
d \log p_t^{\mathrm{SDE}}(\boldsymbol{x}_t) = G(t, \boldsymbol{x}_t) dt + \nabla_{\boldsymbol{x}} \log p_t^{\mathrm{SDE}}(\boldsymbol{x})^T d\mathbf{W}_t,
\tag{44}
$$

*where*

$$
G(t, \boldsymbol{x}) = -\operatorname{div}_{\boldsymbol{x}} \left( f(t, \boldsymbol{x}) - g^2(t) \boldsymbol{s}(t, \boldsymbol{x}) \right) + \frac{1}{2} g^2(t) \| \boldsymbol{s}(t, \boldsymbol{x}) \|^2 - \frac{1}{2} g^2(t) \| \nabla_{\boldsymbol{x}} \log p_t^{\mathrm{SDE}}(\boldsymbol{x}) - \boldsymbol{s}(t, \boldsymbol{x}) \|^2
\tag{45}
$$

*Proof.* Lu et al. (2022) showed that the corresponding forward SDE to Equation 24 is given by

$$
d\boldsymbol{x}_t = \left( f(t, \boldsymbol{x}_t) + g^2(t) \left( \nabla_{\boldsymbol{x}} \log p_t^{\mathrm{SDE}}(\boldsymbol{x}_t) - \boldsymbol{s}(t, \boldsymbol{x}_t) \right) \right) dt + g(t) d\mathbf{W}_t.
\tag{46}
$$

We will apply Itô's lemma (Appendix C) to $h(t, \boldsymbol{x}) = \log p_t^{\mathrm{SDE}}(\boldsymbol{x})$ for $\boldsymbol{x}$ following Equation 23 (not Equation 46, which is intractable due to presence of $\nabla_{\boldsymbol{x}} \log p_t^{\mathrm{SDE}}$). $\frac{\partial}{\partial t} h(t, \boldsymbol{x})$ can be evaluated using Equation 26

$$
\begin{aligned}
\frac{\partial}{\partial t} h(t, \boldsymbol{x}) = & -\operatorname{div}_{\boldsymbol{x}} f(t, \boldsymbol{x}) - g^2(t) \left( \Delta_{\boldsymbol{x}} \log p_t^{\mathrm{SDE}}(\boldsymbol{x}) - \operatorname{div}_{\boldsymbol{x}} \boldsymbol{s}(t, \boldsymbol{x}) \right) + \frac{1}{2} g^2(t) \Delta_{\boldsymbol{x}} \log p_t^{\mathrm{SDE}}(\boldsymbol{x}) \\
& - \nabla_{\boldsymbol{x}} \log p_t^{\mathrm{SDE}}(\boldsymbol{x})^T \left( f(t, \boldsymbol{x}) + g^2(t) \left( \nabla_{\boldsymbol{x}} \log p_t^{\mathrm{SDE}}(\boldsymbol{x}) - \boldsymbol{s}(t, \boldsymbol{x}) \right) - \frac{1}{2} g^2(t) \nabla_{\boldsymbol{x}} \log p_t^{\mathrm{SDE}}(\boldsymbol{x}) \right) \\
= & -\operatorname{div}_{\boldsymbol{x}} f(t, \boldsymbol{x}) - \frac{1}{2} g^2(t) \Delta_{\boldsymbol{x}} \log p_t^{\mathrm{SDE}}(\boldsymbol{x}) + g^2(t) \operatorname{div}_{\boldsymbol{x}} \boldsymbol{s}(t, \boldsymbol{x}) \\
& - \nabla_{\boldsymbol{x}} \log p_t^{\mathrm{SDE}}(\boldsymbol{x})^T \left( f(t, \boldsymbol{x}_t) - g^2(t) \boldsymbol{s}(t, \boldsymbol{x}) + \frac{1}{2} g^2(t) \nabla_{\boldsymbol{x}} \log p_t^{\mathrm{SDE}}(\boldsymbol{x}) \right)
\end{aligned}
\tag{47}
$$

Therefore, we have

$$
\begin{aligned}
& \frac{\partial}{\partial t} h(t, \boldsymbol{x}) + \nabla_{\boldsymbol{x}} h(t, \boldsymbol{x}) f(t, \boldsymbol{x}) + \frac{1}{2} g^2(t) \Delta_{\boldsymbol{x}} h(t, \boldsymbol{x}) \\
= & -\operatorname{div}_{\boldsymbol{x}} f(t, \boldsymbol{x}) + g^2(t) \operatorname{div}_{\boldsymbol{x}} \boldsymbol{s}(t, \boldsymbol{x}) \\
& - \nabla_{\boldsymbol{x}} \log p_t^{\mathrm{SDE}}(\boldsymbol{x})^T \left( -g^2(t) \boldsymbol{s}(t, \boldsymbol{x}) + \frac{1}{2} g^2(t) \nabla_{\boldsymbol{x}} \log p_t^{\mathrm{SDE}}(\boldsymbol{x}) \right) \\
= & -\operatorname{div}_{\boldsymbol{x}} \left( f(t, \boldsymbol{x}) - g^2(t) \boldsymbol{s}(t, \boldsymbol{x}) \right) + \frac{1}{2} g^2(t) \| \boldsymbol{s}(t, \boldsymbol{x}) \|^2 - \frac{1}{2} g^2(t) \| \nabla_{\boldsymbol{x}} \log p_t^{\mathrm{SDE}}(\boldsymbol{x}) - \boldsymbol{s}(t, \boldsymbol{x}) \|^2.
\end{aligned}
\tag{48}
$$

Thus for $\boldsymbol{x}$ following Equation 1, we have

$$d \log p_t^{\mathrm{SDE}}(\boldsymbol{x}_t) = G(t, \boldsymbol{x}_t)dt + \nabla_{\boldsymbol{x}} \log p_t^{\mathrm{SDE}}(\boldsymbol{x}_t)^T d\mathrm{W}_t, \tag{49}$$

where

$$G(t, \boldsymbol{x}) = -\mathrm{div}_{\boldsymbol{x}}\left(f(t, \boldsymbol{x}) - g^2(t)\boldsymbol{s}(t, \boldsymbol{x})\right) + \frac{1}{2}g^2(t)\|\boldsymbol{s}(t, \boldsymbol{x})\|^2 - \frac{1}{2}g^2(t)\|\nabla_{\boldsymbol{x}} \log p_t^{\mathrm{SDE}}(\boldsymbol{x}) - \boldsymbol{s}(t, \boldsymbol{x})\|^2 \tag{50}$$

$\square$

Interestingly, Theorem 6 can be used to derive a lower bound for the likelihood of an individual data point $\boldsymbol{x}_0$ (Kingma et al., 2021; Song et al., 2021b).

**Proposition 3** (ELBO for non-linear SDE). *For any $\boldsymbol{x}_0 \in \mathbb{R}^D$ and $p_t^{\mathrm{SDE}}$ marginal distribution of a process defined by some $p_T^{\mathrm{SDE}}$ and equation 24 for $t < T$, we have*

$$\log p_0^{\mathrm{SDE}}(\boldsymbol{x}_0) = \underbrace{\frac{T}{2}\mathbb{E}_{t,\boldsymbol{x}_t}g^2(t)\|\boldsymbol{s}(t, \boldsymbol{x}_t) - \nabla_{\boldsymbol{x}} \log p_t^{\mathrm{SDE}}(\boldsymbol{x}_t)\|^2}_{\geq 0} + \mathrm{ELBO}(\boldsymbol{x}_0), \tag{51}$$

*where $t \sim \mathcal{U}(0, T)$, $\boldsymbol{x}_t \sim p_{t|0}(\boldsymbol{x}_t|\boldsymbol{x}_0)$ and*

$$\mathrm{ELBO}(\boldsymbol{x}_0) = \mathbb{E}_{\boldsymbol{x}_T \sim p_{T|0}(\boldsymbol{x}_T|\boldsymbol{x}_0)}[\log p_T^{\mathrm{SDE}}(\boldsymbol{x}_T)] + T\mathbb{E}_{t,\boldsymbol{x}_t}L(t, \boldsymbol{x}_t) \tag{52}$$

*and $L(t, \boldsymbol{x}) = -\frac{1}{2}g^2(t)\|\boldsymbol{s}(t, \boldsymbol{x})\|^2 + L_i(t, \boldsymbol{x})$, where one may choose any of the following $L_1, L_2, L_3$ (one could also have different definitions depending on $t$):*

$$L_1(t, \boldsymbol{x}) = \mathrm{div}_{\boldsymbol{x}}\left(f(t, \boldsymbol{x}) - g^2(t)\boldsymbol{s}(t, \boldsymbol{x})\right) \tag{53}$$

$$L_2(t, \boldsymbol{x}) = -\left(f(t, \boldsymbol{x}_t) - g^2(t)\boldsymbol{s}(t, \boldsymbol{x}_t)\right)^T \nabla_{\boldsymbol{x}_t} \log p_{t|0}(\boldsymbol{x}_t|\boldsymbol{x}_0) \tag{54}$$

$$L_3(t, \boldsymbol{x}) = \mathrm{div}_{\boldsymbol{x}}(f(t, \boldsymbol{x})) + g^2(t)\boldsymbol{s}(t, \boldsymbol{x}_t)^T \nabla_{\boldsymbol{x}_t} \log p_{t|0}(\boldsymbol{x}_t|\boldsymbol{x}_0) \tag{55}$$

*Proof.* Using Equation 44:

$$\log p_0^{\mathrm{SDE}}(\boldsymbol{x}_0) = \log p_T^{\mathrm{SDE}}(\boldsymbol{x}_T) - \int_0^T G(t, \boldsymbol{x}_t)dt - \int_0^T g(t)\nabla_{\boldsymbol{x}} \log p_t^{\mathrm{SDE}}(\boldsymbol{x}_t)^T d\mathrm{W}_t, \tag{56}$$

for $\boldsymbol{x}$ being a random trajectory following Equation 23 starting at $\boldsymbol{x}_0$. Using the definition of $G(t, \boldsymbol{x})$:

$$\begin{aligned} \log p_0^{\mathrm{SDE}}(\boldsymbol{x}_0) = {}& \log p_T^{\mathrm{SDE}}(\boldsymbol{x}_T) \\ & - \int_0^T \left(-\mathrm{div}_{\boldsymbol{x}}\left(f(t, \boldsymbol{x}_t) - g^2(t)\boldsymbol{s}(t, \boldsymbol{x}_t)\right) + \frac{1}{2}g^2(t)\|\boldsymbol{s}(t, \boldsymbol{x}_t)\|^2\right)dt \\ & + \frac{1}{2}\int_0^T g^2(t)\|\boldsymbol{s}(t, \boldsymbol{x}_t) - \nabla_{\boldsymbol{x}} \log p_t^{\mathrm{SDE}}(\boldsymbol{x}_t)\|^2 dt \\ & - \int_0^T g(t)\nabla_{\boldsymbol{x}} \log p_t^{\mathrm{SDE}}(\boldsymbol{x}_t)^T d\mathrm{W}_t. \end{aligned} \tag{57}$$

We can take the expectation of both sides of Equation 57 w.r.t. $\boldsymbol{x} \sim \nu(\boldsymbol{x}|\boldsymbol{x}_0)$, where $\nu(\boldsymbol{x}|\boldsymbol{x}_0)$ is a path measure of $\boldsymbol{x}$ starting at $\boldsymbol{x}_0$. Note that the LHS of Equation 57 is constant w.r.t $\nu(\boldsymbol{x}|\boldsymbol{x}_0)$ and

thus it is equal to its expectation.

$$\underbrace{\mathbb{E}[\log p_0^{\text{SDE}}(\boldsymbol{x}_0)]}_{=\log p_0^{\text{SDE}}(\boldsymbol{x}_0)} = \underbrace{\mathbb{E}[\log p_T^{\text{SDE}}(\boldsymbol{x}_T)]}_{\text{"first term"}}$$

$$- \underbrace{\mathbb{E}\left[\int_0^T \left(-\text{div}_{\boldsymbol{x}}\left(f(t,\boldsymbol{x}_t) - g^2(t)\boldsymbol{s}(t,\boldsymbol{x}_t)\right) + \frac{1}{2}g^2(t)\|\boldsymbol{s}(t,\boldsymbol{x}_t)\|^2\right)dt\right]}_{\text{"second term"}}$$

$$+ \underbrace{\mathbb{E}\left[\frac{1}{2}\int_0^T g^2(t)\|\boldsymbol{s}(t,\boldsymbol{x}_t) - \nabla_{\boldsymbol{x}}\log p_t^{\text{SDE}}(\boldsymbol{x}_t)\|^2 dt\right]}_{\text{"third term"}}$$

$$- \underbrace{\mathbb{E}\left[\int_0^T g(t)\nabla_{\boldsymbol{x}}\log p_t(\boldsymbol{x}_t)^T d\text{W}_t\right]}_{\text{"fourth term"}}.$$

$$(58)$$

**First term.** Since the expectation is taken w.r.t $\nu(\boldsymbol{x}|\boldsymbol{x}_0)$, we have

$$\mathbb{E}_{\boldsymbol{x}\sim\nu(\boldsymbol{x}|\boldsymbol{x}_0)}[\log p_T^{\text{SDE}}(\boldsymbol{x}_T)] = \mathbb{E}_{\boldsymbol{x}_T\sim p_{T|0}(\boldsymbol{x}_T|\boldsymbol{x}_0)}[\log p_T^{\text{SDE}}(\boldsymbol{x}_T)], \tag{59}$$

where $p_{T|0}$ is the forward transition probability of equation 23.

**Second term.** Using Fubini's theorem we have

$$\mathbb{E}_{\boldsymbol{x}\sim\nu(\boldsymbol{x}|\boldsymbol{x}_0)}\left[\int_0^T F(t,\boldsymbol{x}_t)dt\right] = \int_0^T \left(\mathbb{E}_{\boldsymbol{x}_t\sim\nu(\boldsymbol{x}_t|\boldsymbol{x}_0)}F(t,\boldsymbol{x}_t)\right)dt$$

$$= \int_0^T \left(\mathbb{E}_{\boldsymbol{x}_t\sim p_{t|0}(\boldsymbol{x}_t|\boldsymbol{x}_0)}F(t,\boldsymbol{x}_t)\right)dt. \tag{60}$$

After substituting for $F$, we get

$$\mathbb{E}_{\boldsymbol{x}_t\sim p_{t|0}(\boldsymbol{x}_t|\boldsymbol{x}_0)}F(t,\boldsymbol{x}_t) = \mathbb{E}_{\boldsymbol{x}_t\sim p_{t|0}(\boldsymbol{x}_t|\boldsymbol{x}_0)}\left(-\text{div}_{\boldsymbol{x}}\left(f(t,\boldsymbol{x}_t) - g^2(t)\boldsymbol{s}(t,\boldsymbol{x}_t)\right) + \frac{1}{2}g^2(t)\|\boldsymbol{s}(t,\boldsymbol{x}_t)\|^2\right). \tag{61}$$

Note that for any $t$ the divergence term under the expectation can equivalently be written in one of three ways (integration by parts; assumptions 8 and 14 in Appendix A):

$$- \mathbb{E}_{\boldsymbol{x}_t\sim p_{t|0}(\boldsymbol{x}_t|\boldsymbol{x}_0)}\text{div}_{\boldsymbol{x}}\left(f(t,\boldsymbol{x}_t) - g^2(t)\boldsymbol{s}(t,\boldsymbol{x}_t)\right)$$

$$\stackrel{(i)}{=} \mathbb{E}_{\boldsymbol{x}_t\sim p_{t|0}(\boldsymbol{x}_t|\boldsymbol{x}_0)}\left[\left(f(t,\boldsymbol{x}_t) - g^2(t)\boldsymbol{s}(t,\boldsymbol{x}_t)\right)^T \nabla_{\boldsymbol{x}_t}\log p_{t|0}(\boldsymbol{x}_t|\boldsymbol{x}_0)\right] \tag{62}$$

$$\stackrel{(ii)}{=} \mathbb{E}_{\boldsymbol{x}_t\sim p_{t|0}(\boldsymbol{x}_t|\boldsymbol{x}_0)}\left[-\text{div}_{\boldsymbol{x}}f(t,\boldsymbol{x}_t) - g^2(t)\boldsymbol{s}(t,\boldsymbol{x}_t)^T\nabla_{\boldsymbol{x}_t}\log p_{t|0}(\boldsymbol{x}_t|\boldsymbol{x}_0)\right],$$

which holds due to applying integration by parts either

- $(i)$ : to $f(t,\boldsymbol{x}_t) - g^2(t)\boldsymbol{s}(t,\boldsymbol{x}_t)$ and $p_{t|0}(\boldsymbol{x}_t|\boldsymbol{x}_0)$, or

- $(ii)$ : to $\boldsymbol{s}(t,\boldsymbol{x}_t)$ and $p_{t|0}(\boldsymbol{x}_t|\boldsymbol{x}_0)$.

**Third term.**

$$\mathbb{E}\left[\frac{1}{2}\int_0^T g^2(t)\|\boldsymbol{s}(t,\boldsymbol{x}_t) - \nabla_{\boldsymbol{x}}\log p_t^{\text{SDE}}(\boldsymbol{x}_t)\|^2 dt\right]$$

$$= \frac{1}{2}\int_0^T g^2(t)\left(\mathbb{E}_{\boldsymbol{x}_t\sim p_{t|0}(\boldsymbol{x}_t|\boldsymbol{x}_0)}\|\boldsymbol{s}(t,\boldsymbol{x}_t) - \nabla_{\boldsymbol{x}}\log p_t^{\text{SDE}}(\boldsymbol{x}_t)\|^2\right)dt \tag{63}$$

**Fourth term.** Using Assumption 12 (Appendix A) and the fact that $g(t)\nabla_{\boldsymbol{x}}\log p_t^{\text{SDE}}(\boldsymbol{x}_t)$ is W adapted, we have

$$\mathbb{E}\left[\int_0^T g(t)\nabla_{\boldsymbol{x}}\log p_t^{\text{SDE}}(\boldsymbol{x}_t)^T d\mathrm{W}_t\right] = 0. \tag{64}$$

Combining all four terms yields the claim. $\qquad\square$

**Corollary 1** (ELBO for Linear SDE). *For an SDE with linear drift (equation 1), for any $\boldsymbol{x}_0 \in \mathbb{R}^D$, assuming $p_T^{\text{SDE}} = \mathcal{N}(\mathbf{0}, \sigma_T^2 \boldsymbol{I}_D)$ we have*

$$\log p_0^{\text{SDE}}(\boldsymbol{x}_0) = \underbrace{\frac{T}{2}\mathbb{E}_{t,\boldsymbol{\varepsilon}}g^2(t)\|\boldsymbol{s}(t,\boldsymbol{x}_t) - \nabla_{\boldsymbol{x}}\log p_t^{\text{SDE}}(\boldsymbol{x}_t)\|^2}_{\geq 0} + \text{ELBO}(\boldsymbol{x}_0) \tag{65}$$

*where $t \sim \mathcal{U}(0,T)$, $\boldsymbol{\varepsilon} \sim \mathcal{N}(\mathbf{0}, \boldsymbol{I}_D)$, $\boldsymbol{x}_t = \alpha_t \boldsymbol{x}_0 + \sigma_t \boldsymbol{\varepsilon}$ and*

$$\text{ELBO}(\boldsymbol{x}_0) = C - \frac{e^{\lambda_{min}}}{2}\|\boldsymbol{x}_0\|^2 - \frac{T}{2}\mathbb{E}_{t,\boldsymbol{\varepsilon}}\left(-\frac{d\lambda_t}{dt}\right)\|\sigma_t \boldsymbol{s}(t,\alpha_t\boldsymbol{x}_0 + \sigma_t\boldsymbol{\varepsilon}) + \boldsymbol{\varepsilon}\|^2 \tag{66}$$

*and $C = -\frac{D}{2}\left(1 + \log(2\pi\sigma_0^2)\right)$.*

*Proof.* In the linear SDE case (equation 1) we have $p_{t|0}(\boldsymbol{x}_t|\boldsymbol{x}_0) = \mathcal{N}(\boldsymbol{x}_t|\alpha_t\boldsymbol{x}_0, \sigma_t^2\boldsymbol{I}_D)$. Using Proposition 3, we have

$$\text{ELBO}(\boldsymbol{x}_0) = \mathbb{E}_{\boldsymbol{x}_T \sim p_{T|0}(\boldsymbol{x}_T|\boldsymbol{x}_0)}[\log p_T^{\text{SDE}}(\boldsymbol{x}_T)] + T\mathbb{E}_{t,\boldsymbol{x}_t}L(t,\boldsymbol{x}_t) \tag{67}$$

and we choose

$$
\begin{aligned}
L(t,\boldsymbol{x}) &= -\frac{1}{2}g^2(t)\|\boldsymbol{s}(t,\boldsymbol{x})\|^2 + L_3(t,\boldsymbol{x}) \\
&= -\frac{1}{2}g^2(t)\|\boldsymbol{s}(t,\boldsymbol{x})\|^2 + \text{div}_{\boldsymbol{x}}(f(t,\boldsymbol{x})) + g^2(t)\boldsymbol{s}(t,\boldsymbol{x})^T\nabla_{\boldsymbol{x}}\log p_{t|0}(\boldsymbol{x}|\boldsymbol{x}_0) \\
&= f(t)D - \frac{1}{2}g^2(t)\|\boldsymbol{s}(t,\boldsymbol{x}) - \nabla_{\boldsymbol{x}}\log p_{t|0}(\boldsymbol{x}|\boldsymbol{x}_0)\|^2 + \frac{1}{2}g^2(t)\|\nabla_{\boldsymbol{x}}\log p_{t|0}(\boldsymbol{x}|\boldsymbol{x}_0)\|^2
\end{aligned}
\tag{68}
$$

Since $\nabla_{\boldsymbol{x}}\log p_{t|0}(\boldsymbol{x}|\boldsymbol{x}_0) = \frac{\alpha_t\boldsymbol{x}_0 - \boldsymbol{x}}{\sigma_t^2}$ and $\boldsymbol{x}_t \sim p_{t|0}(\boldsymbol{x}_t|\boldsymbol{x}_0)$ is equivalent to $\boldsymbol{x}_t = \alpha_t\boldsymbol{x}_0 + \sigma_t\boldsymbol{\varepsilon}$ for $\boldsymbol{\varepsilon} \sim \mathcal{N}(\mathbf{0}, \boldsymbol{I}_D)$, we have $\nabla_{\boldsymbol{x}}\log p_{t|0}(\boldsymbol{x}_t|\boldsymbol{x}_0) = \frac{-\boldsymbol{\varepsilon}}{\sigma_t}$ and

$$
\begin{aligned}
\text{ELBO}(\boldsymbol{x}_0) &= -\frac{T}{2}\mathbb{E}_{t,\boldsymbol{\varepsilon}}g^2(t)\left\|\boldsymbol{s}(t,\alpha_t\boldsymbol{x}_0 + \sigma_t\boldsymbol{\varepsilon}) + \frac{\boldsymbol{\varepsilon}}{\sigma_t}\right\|^2 \\
&\quad + \mathbb{E}_{\boldsymbol{x}_T \sim p_{T|0}(\boldsymbol{x}_T|\boldsymbol{x}_0)}[\log p_T^{\text{SDE}}(\boldsymbol{x}_T)] + DT\mathbb{E}_{t,\boldsymbol{x}_t}f(t) + \frac{T}{2}\mathbb{E}_{t,\boldsymbol{\varepsilon}}g^2(t)\left\|\frac{\boldsymbol{\varepsilon}}{\sigma_t}\right\|^2 \\
&= -\frac{T}{2}\mathbb{E}_{t,\boldsymbol{\varepsilon}}\left(-\frac{d\lambda_t}{dt}\right)\|\sigma_t\boldsymbol{s}(t,\alpha_t\boldsymbol{x}_0 + \sigma_t\boldsymbol{\varepsilon}) + \boldsymbol{\varepsilon}\|^2 \\
&\quad + \underbrace{\mathbb{E}_{\boldsymbol{\varepsilon}}[\log p_T^{\text{SDE}}(\alpha_T\boldsymbol{x}_0 + \sigma_T\boldsymbol{\varepsilon})]}_{\text{``first term''}} + \underbrace{DT\mathbb{E}_t\left(f(t) - \frac{1}{2}\frac{d\lambda_t}{dt}\right)}_{\text{``second term''}}
\end{aligned}
\tag{69}
$$

**First term**

$$
\begin{aligned}
\mathbb{E}_{\boldsymbol{\varepsilon}}[\log p_T^{\text{SDE}}(\alpha_T\boldsymbol{x}_0 + \sigma_T\boldsymbol{\varepsilon})] &= -\frac{D}{2}\log(2\pi\sigma_T^2) - \frac{1}{2\sigma_T^2}\mathbb{E}_{\boldsymbol{\varepsilon}}\|\alpha_T\boldsymbol{x}_0 + \sigma_T\boldsymbol{\varepsilon}\|^2 \\
&= -\frac{D}{2}\log(2\pi\sigma_T^2) - \frac{1}{2\sigma_T^2}\left(\alpha_T^2\|\boldsymbol{x}_0\|^2 + \sigma_T^2 D\right) \\
&= -\frac{D}{2}\left(1 + \log(2\pi\sigma_T^2)\right) - \frac{e^{\lambda_{\min}}}{2}\|\boldsymbol{x}_0\|^2
\end{aligned}
\tag{70}
$$

**Second term**

$$DT\mathbb{E}_t\left(f(t) - \frac{1}{2}\frac{d\lambda_t}{dt}\right) = D\int_0^T \frac{d\log\sigma_t}{dt}dt = D\left(\log\sigma_T - \log\sigma_0\right) \tag{71}$$

Combining all the terms yields the claim. $\qquad\square$

We can now use Theorem 6 to prove Theorem 4.

**Theorem 4** (Approximate Augmented Forward SDE). *Let $s(t, \boldsymbol{x}_t)$ be the model approximating the score function and $\boldsymbol{x}_0 \in \mathbb{R}^D$ given. Define an auxiliary process $\omega$ starting at $\omega_0 = 0$. If*

$$d\begin{bmatrix}\boldsymbol{x}_t\\\omega_t\end{bmatrix} = \begin{bmatrix}f(t)\boldsymbol{x}_t\\-f(t)D + g^2(t)\left(\frac{1}{2}\|\ \boldsymbol{s}(t,\boldsymbol{x}_t)\ \|^2 + \mathrm{div}_{\boldsymbol{x}}\ \boldsymbol{s}(t,\boldsymbol{x}_t)\ \right)\end{bmatrix}dt + g(t)\begin{bmatrix}\boldsymbol{I}_D\\\boldsymbol{s}(t,\boldsymbol{x}_t)\end{bmatrix}^T d\mathrm{W}_t. \tag{10}$$

*Then*

$$\omega_T = \log p_T^{\mathrm{SDE}}(\boldsymbol{x}_T) - \log p_0^{\mathrm{SDE}}(\boldsymbol{x}_0) + \mathrm{Y}_{\boldsymbol{x}_0}, \tag{11}$$

*where $\mathrm{Y}_{\boldsymbol{x}_0}$ is a random variable such that*

$$\mathbb{E}\mathrm{Y}_{\boldsymbol{x}_0} = \frac{T}{2}\mathbb{E}_{t\sim\mathcal{U}(0,T)}\mathbb{E}_{\boldsymbol{x}_t\sim p(\boldsymbol{x}_t|\boldsymbol{x}_0)}g^2(t)\|\ \boldsymbol{s}(t,\boldsymbol{x}_t)\ -\ \nabla_{\boldsymbol{x}}\log p_t^{\mathrm{SDE}}(\boldsymbol{x}_t)\ \|^2 \geq 0. \tag{12}$$

*Furthermore, $\mathrm{Y}_{\boldsymbol{x}_0}$ can be written as $\mathrm{Y}_{\boldsymbol{x}_0} = \mathrm{Y}_1 + \mathrm{Y}_2$, where*

$$\mathrm{Y}_1 = \frac{1}{2}\int_0^T g^2(t)\|\nabla_{\boldsymbol{x}}\log p_t^{\mathrm{SDE}}(\boldsymbol{x}_t) - \boldsymbol{s}(t,\boldsymbol{x}_t)\|^2 dt$$

*and*

$$\mathbb{E}\mathrm{Y}_2 = 0;\ \mathrm{Var}(\mathrm{Y}_2) = \int_0^T g^2(t)\mathbb{E}_{\boldsymbol{x}_t\sim p(\boldsymbol{x}_t|\boldsymbol{x}_0)}\|\boldsymbol{s}(t,\boldsymbol{x}_t) - \nabla_{\boldsymbol{s}}\log p_t^{\mathrm{SDE}}(\boldsymbol{x}_t)\|^2 dt.$$

For the non-linear drift (equation 23), we have (the difference highlighted in blue)

$$d\begin{bmatrix}\boldsymbol{x}_t\\\omega_t\end{bmatrix} = \begin{bmatrix}f(t,\boldsymbol{x}_t)\\-\mathrm{div}_{\boldsymbol{x}}\ f(t,\boldsymbol{x}_t) + g^2(t)\left(\frac{1}{2}\|\ \boldsymbol{s}(t,\boldsymbol{x})\ \|^2 + \mathrm{div}_{\boldsymbol{x}}\ \boldsymbol{s}(t,\boldsymbol{x})\ \right)\end{bmatrix}dt + g(t)\begin{bmatrix}\boldsymbol{I}_D\\\boldsymbol{s}(t,\boldsymbol{x}_t)\end{bmatrix}^T d\mathrm{W}_t.$$

*Proof.* Using Theorem 6 we have

$$\begin{aligned}\log p_0^{\mathrm{SDE}}(\boldsymbol{x}_0) &= \log p_T^{\mathrm{SDE}}(\boldsymbol{x}_T) - \int_0^T G(t,\boldsymbol{x}_t)dt - \int_0^T g(t)\nabla_{\boldsymbol{x}}\log p_t^{\mathrm{SDE}}(\boldsymbol{x}_t)^T d\mathrm{W}_t,\\ &= \log p_T^{\mathrm{SDE}}(\boldsymbol{x}_T) - \int_0^T d\omega_t + \mathrm{Y}_1 + \mathrm{Y}_2,\end{aligned} \tag{72}$$

where

$$\mathrm{Y}_1 = \frac{1}{2}\int_0^T g^2(t)\|\nabla_{\boldsymbol{x}}\log p_t^{\mathrm{SDE}}(\boldsymbol{x}_t) - \boldsymbol{s}(t,\boldsymbol{x}_t)\|^2 dt \tag{73}$$

and

$$\mathrm{Y}_2 = \int_0^T g(t)\left(\boldsymbol{s}(t,\boldsymbol{x}_t) - \nabla_{\boldsymbol{x}}\log p_t^{\mathrm{SDE}}(\boldsymbol{x}_t)\right)d\mathrm{W}_t. \tag{74}$$

Since $\int_0^T d\omega_t = \omega_T - \omega_0 = \omega_T$, we have

$$\log p_0^{\mathrm{SDE}}(\boldsymbol{x}_0) = \log p_T^{\mathrm{SDE}}(\boldsymbol{x}_T) - \omega_T + \mathrm{Y} \tag{75}$$

for $\mathrm{Y} = \mathrm{Y}_1 + \mathrm{Y}_2$. $\qquad\square$

Similarly to Theorem 6 we can derive the dynamics of $\log p_t^{\mathrm{SDE}}(\boldsymbol{x}_t)$ under the approximate reverse SDE (Equation 24).

**Theorem 7** (Approximate augmented reverse SDE). *Let $\boldsymbol{x}$ be a random process following Equation 24, then*

$$d\begin{bmatrix} \boldsymbol{x}_t \\ \log p_t^{\mathrm{SDE}}(\boldsymbol{x}_t) \end{bmatrix} = \begin{bmatrix} f(t, \boldsymbol{x}_t) - g^2(t)\boldsymbol{s}(t, \boldsymbol{x}_t) \\ \tilde{F}(t, \boldsymbol{x}_t) \end{bmatrix} dt + g(t)\begin{bmatrix} \boldsymbol{I}_d \\ \nabla_{\boldsymbol{x}} \log p_t^{\mathrm{SDE}}(\boldsymbol{x}_t)^T \end{bmatrix} d\overline{\mathbf{W}}_t, \quad (76)$$

*where*

$$\tilde{F}(t, \boldsymbol{x}) = -\operatorname{div}_{\boldsymbol{x}} f(t, \boldsymbol{x}) - g^2(t)\underbrace{\left(\Delta_{\boldsymbol{x}} \log p_t^{\mathrm{SDE}}(\boldsymbol{x}) - \operatorname{div}_{\boldsymbol{x}} \boldsymbol{s}(t, \boldsymbol{x})\right)}_{=0 \text{ when } \boldsymbol{s}(t, x) = \nabla_{\boldsymbol{x}} \log p_t^{\mathrm{SDE}}(\boldsymbol{x})} - \frac{1}{2}g^2(t)\|\nabla_{\boldsymbol{x}} \log p_t^{\mathrm{SDE}}(\boldsymbol{x})\|^2.$$

$$(77)$$

*Proof.* The approximate reverse SDE can equivalently be written as

$$d\boldsymbol{x}_t = -\left(f(T - t, \boldsymbol{x}_t) - g^2(T - t)\boldsymbol{s}(T - t, \boldsymbol{x}_t)\right) dt + g(T - t)d\mathbf{W}_t \quad (78)$$

for $dt > 0$ and W running forward in time. We will apply Itô's lemma (Appendix C) to $h(t, \boldsymbol{x}) = \log p_{T-t}^{\mathrm{SDE}}(\boldsymbol{x})$. From Equation 47 we have:

$$\frac{\partial}{\partial t} h(t, \boldsymbol{x}) = \operatorname{div}_{\boldsymbol{x}} f(T - t, \boldsymbol{x}) + \frac{1}{2}g^2(T - t)\Delta_{\boldsymbol{x}} \log p_{T-t}^{\mathrm{SDE}}(\boldsymbol{x}) - g^2(T - t)\operatorname{div}_{\boldsymbol{x}} \boldsymbol{s}(T - t, \boldsymbol{x})$$

$$+ \nabla_{\boldsymbol{x}} \log p_{T-t}^{\mathrm{SDE}}(\boldsymbol{x})^T \left(f(T - t, \boldsymbol{x}) - g^2(T - t)\boldsymbol{s}(T - t, \boldsymbol{x}) + \frac{1}{2}g^2(T - t)\nabla_{\boldsymbol{x}} \log p_{T-t}^{\mathrm{SDE}}(\boldsymbol{x})\right)$$

$$(79)$$

Therefore the drift of $h(t, \boldsymbol{x}_t)$ is given by

$$\frac{\partial}{\partial t} h(t, \boldsymbol{x}_t) + \nabla_{\boldsymbol{x}} h(t, \boldsymbol{x}_t)^T \left(-f(T - t, \boldsymbol{x}_t) + g^2(T - t)\boldsymbol{s}(T - t, \boldsymbol{x}_t)\right) + \frac{1}{2}g^2(T - t)\Delta_{\boldsymbol{x}} h(t, \boldsymbol{x}_t)$$

$$= \operatorname{div}_{\boldsymbol{x}} f(T - t, \boldsymbol{x}_t) + g^2(T - t)\Delta_{\boldsymbol{x}} \log p_{T-t}^{\mathrm{SDE}}(\boldsymbol{x}_t) - g^2(T - t)\operatorname{div}_{\boldsymbol{x}} \boldsymbol{s}(T - t, \boldsymbol{x}_t)$$

$$+ \frac{1}{2}g^2(T - t)\|\nabla_{\boldsymbol{x}} \log p_{T-t}^{\mathrm{SDE}}(\boldsymbol{x}_t)\|^2$$

$$(80)$$

and therefore for $\boldsymbol{x}$ following the approximate reverse SDE (Equation 24), we have

$$d\log p_t^{\mathrm{SDE}}(\boldsymbol{x}_t) = \tilde{F}(t, \boldsymbol{x}_t)dt + g(t)\nabla_{\boldsymbol{x}} \log p_t^{\mathrm{SDE}}(\boldsymbol{x}_t)^T d\overline{\mathbf{W}}_t, \quad (81)$$

where $dt < 0$, $\overline{\mathbf{W}}$ is running backwards in time and

$$\tilde{F}(t, \boldsymbol{x}) = -\operatorname{div}_{\boldsymbol{x}} f(t, \boldsymbol{x}) - g^2(t)\Delta_{\boldsymbol{x}} \log p_t^{\mathrm{SDE}}(\boldsymbol{x}) + g^2(t)\operatorname{div}_{\boldsymbol{x}} \boldsymbol{s}(t, \boldsymbol{x}) - \frac{1}{2}g^2(t)\|\nabla_{\boldsymbol{x}} \log p_t^{\mathrm{SDE}}(\boldsymbol{x})\|^2.$$

$$(82)$$

$\square$

Theorem 7 defines the exact dynamics of $\log p_t^{\mathrm{SDE}}(\boldsymbol{x}_t)$. However, $d\log p_t^{\mathrm{SDE}}(\boldsymbol{x}_t)$ depends on $\nabla_{\boldsymbol{x}} \log p_t^{\mathrm{SDE}}(\boldsymbol{x}_t)$, which we cannot access in practice. We only have access to the approximation $\boldsymbol{s}(t, \boldsymbol{x})$. We now show that replacing the true $\nabla_{\boldsymbol{x}} \log p_t^{\mathrm{SDE}}(\boldsymbol{x}_t)$ with $\boldsymbol{s}$ no longer provides exact likelihood estimates, but an "upper bound in expectation".

**Theorem 3** (Approximate Augmented Reverse SDE). *Let $\boldsymbol{s}(t, \boldsymbol{x})$ be an approximation of the score function. Let $\boldsymbol{x}_T \sim p_T$ and define an auxiliary process $r$ starting at $r_T = \log p_T^{\mathrm{SDE}}(\boldsymbol{x}_T)$. If*

$$d\begin{bmatrix} \boldsymbol{x}_t \\ r_t \end{bmatrix} = \begin{bmatrix} f(t)\boldsymbol{x}_t - g^2(t)\,\boldsymbol{s}(t, \boldsymbol{x}_t) \\ -f(t)D - \frac{1}{2}g^2(t)\|\,\boldsymbol{s}(t, \boldsymbol{x}_t)\,\|^2 \end{bmatrix} dt + g(t)\begin{bmatrix} \boldsymbol{I}_D \\ \boldsymbol{s}(t, \boldsymbol{x}_t) \end{bmatrix}^T d\overline{\mathbf{W}}_t, \quad (7)$$

*then $\boldsymbol{x}_0 \sim p_0^{\mathrm{SDE}}(\boldsymbol{x}_0)$ and*

$$r_0 = \log p_0^{\mathrm{SDE}}(\boldsymbol{x}_0) + \mathrm{X}, \quad (8)$$

*where $\mathrm{X}$ is a random variable such that the bias of $r_0$ is given by*

$$\mathbb{E}\mathrm{X} = \frac{T}{2}\mathbb{E}_{t \sim \mathcal{U}(0,T), \boldsymbol{x}_t \sim p_t^{\mathrm{SDE}}(\boldsymbol{x}_t)}\left[g^2(t)\|\,\boldsymbol{s}(t, \boldsymbol{x}_t)\, - \,\nabla_{\boldsymbol{x}} \log p_t^{\mathrm{SDE}}(\boldsymbol{x}_t)\,\|^2\right] \geq 0. \quad (9)$$

Furthermore, X can be written as $X = X_1 + X_2$, where

$$X_1 = \int_0^T g^2(t) \left( \mathrm{div}_{\boldsymbol{x}}\, \boldsymbol{s}(t, \boldsymbol{x}_t) + \frac{1}{2}\|\boldsymbol{s}(t, \boldsymbol{x}_t)\|^2 - \Delta_{\boldsymbol{x}} \log p_t^{\mathrm{SDE}}(\boldsymbol{x}_t) - \frac{1}{2}\|\nabla_{\boldsymbol{x}} \log p_t^{\mathrm{SDE}}(\boldsymbol{x}_t)\|^2 \right) dt$$

and

$$\mathbb{E}X_2 = 0;\ \mathrm{Var}(X_2) = \int_0^T g^2(t) \mathbb{E}_{\boldsymbol{x}_t \sim p_t^{\mathrm{SDE}}(\boldsymbol{x}_t)}\|\boldsymbol{s}(t, \boldsymbol{x}_t) - \nabla_{\boldsymbol{x}} \log p_t^{\mathrm{SDE}}(\boldsymbol{x}_t)\|^2 dt$$

For the non-linear drift (equation 23), we have (the difference highlighted in blue)

$$d \begin{bmatrix} \boldsymbol{x}_t \\ r_t \end{bmatrix} = \begin{bmatrix} f(t, \boldsymbol{x}_t) - g^2(t)\, \boldsymbol{s}(t, \boldsymbol{x}_t) \\ -\mathrm{div}_{\boldsymbol{x}}\, f(t, \boldsymbol{x}_t) - \frac{1}{2}g^2(t)\|\, \boldsymbol{s}(t, \boldsymbol{x}_t)\, \|^2 \end{bmatrix} dt + g(t) \begin{bmatrix} \boldsymbol{I}_D \\ \boldsymbol{s}(t, \boldsymbol{x}_t)^T \end{bmatrix} d\overline{\mathbf{W}}_t,$$

*Proof.*

$$\log p_0^{\mathrm{SDE}}(\boldsymbol{x}_0) = \log p_T^{\mathrm{SDE}}(\boldsymbol{x}_T) - \int_0^T d\log p_t^{\mathrm{SDE}}(\boldsymbol{x}_t) \tag{83}$$

From Theorem 7 we have

$$\int_0^T d\log p_t^{\mathrm{SDE}}(\boldsymbol{x}_t) = \int_0^T \tilde{F}(t, \boldsymbol{x}_t) dt + \int_0^T g(t)\nabla_{\boldsymbol{x}} \log p_t^{\mathrm{SDE}}(\boldsymbol{x}_t)^T d\overline{\mathbf{W}}_t$$

$$= \int_0^T \left( -\mathrm{div}_{\boldsymbol{x}}\, f(t, \boldsymbol{x}_t) - \frac{1}{2}g^2(t)\|\boldsymbol{s}(t, \boldsymbol{x}_t)\|^2 \right) dt \tag{84}$$

$$+ \int_0^T g(t)\boldsymbol{s}(t, \boldsymbol{x}_t)^T d\overline{\mathbf{W}}_t + X_1 + X_2,$$

where

$$X_1 = \int_0^T g^2(t) \left( \mathrm{div}_{\boldsymbol{x}}\, \boldsymbol{s}(t, \boldsymbol{x}_t) + \frac{1}{2}\|\boldsymbol{s}(t, \boldsymbol{x}_t)\|^2 - \Delta_{\boldsymbol{x}} \log p_t^{\mathrm{SDE}}(\boldsymbol{x}_t) - \frac{1}{2}\|\nabla_{\boldsymbol{x}} \log p_t^{\mathrm{SDE}}(\boldsymbol{x}_t)\|^2 \right) dt \tag{85}$$

and

$$X_2 = \int_0^T g(t) \left( \nabla_{\boldsymbol{x}} \log p_t^{\mathrm{SDE}}(\boldsymbol{x}_t) - \boldsymbol{s}(t, \boldsymbol{x}_t) \right)^T d\overline{\mathbf{W}}_t \tag{86}$$

Note that from Assumption 13 (Appendix A) we have

$$\mathbb{E}X_2 = 0. \tag{87}$$

Furthermore, using Fubini's theorem, we have

$$\mathbb{E}X_1 = \int_0^T g^2(t) \mathbb{E}_{\boldsymbol{x}_t \sim p_t^{\mathrm{SDE}}(\boldsymbol{x}_t)} \left( \mathrm{div}_{\boldsymbol{x}}\, \boldsymbol{s}(t, \boldsymbol{x}_t) + \frac{1}{2}\|\boldsymbol{s}(t, \boldsymbol{x}_t)\|^2 - \Delta_{\boldsymbol{x}} \log p_t^{\mathrm{SDE}}(\boldsymbol{x}_t) \right.$$

$$\left. - \frac{1}{2}\|\nabla_{\boldsymbol{x}} \log p_t^{\mathrm{SDE}}(\boldsymbol{x}_t)\|^2 \right) dt. \tag{88}$$

Now rewrite

$$\mathbb{E}_{\boldsymbol{x}_t \sim p_t^{\mathrm{SDE}}(\boldsymbol{x}_t)} \left( \mathrm{div}_{\boldsymbol{x}}\, \boldsymbol{s}(t, \boldsymbol{x}_t) - \Delta_{\boldsymbol{x}} \log p_t^{\mathrm{SDE}}(\boldsymbol{x}_t) \right)$$

$$= \mathbb{E}_{\boldsymbol{x}_t \sim p_t^{\mathrm{SDE}}(\boldsymbol{x}_t)} \mathrm{div}_{\boldsymbol{x}} \left( \boldsymbol{s}(t, \boldsymbol{x}_t) - \nabla_{\boldsymbol{x}} \log p_t^{\mathrm{SDE}}(\boldsymbol{x}_t) \right)$$

$$\overset{(i)}{=} \mathbb{E}_{\boldsymbol{x}_t \sim p_t^{\mathrm{SDE}}(\boldsymbol{x}_t)} \left( -\boldsymbol{s}(t, \boldsymbol{x}_t) + \nabla_{\boldsymbol{x}} \log p_t^{\mathrm{SDE}}(\boldsymbol{x}_t) \right)^T \nabla_{\boldsymbol{x}} \log p_t^{\mathrm{SDE}}(\boldsymbol{x}_t) \tag{89}$$

$$= \mathbb{E}_{\boldsymbol{x}_t \sim p_t^{\mathrm{SDE}}(\boldsymbol{x}_t)} \left( -\boldsymbol{s}(t, \boldsymbol{x}_t)^T \nabla_{\boldsymbol{x}} \log p_t^{\mathrm{SDE}}(\boldsymbol{x}_t) + \|\nabla_{\boldsymbol{x}} \log p_t^{\mathrm{SDE}}(\boldsymbol{x}_t)\|^2 \right),$$

where $(i)$ is applying integration by parts (Assumptions 8 and 11 in Appendix A). Substituting back to Equation 88, we get

$$\mathbb{E}X_1 = \int_0^T \frac{1}{2}g^2(t)\mathbb{E}_{\boldsymbol{x}_t \sim p_t^{\mathrm{SDE}}(\boldsymbol{x}_t)} \left( \|\boldsymbol{s}(t, \boldsymbol{x}_t)\|^2 - 2\boldsymbol{s}(t, \boldsymbol{x}_t)^T \nabla_{\boldsymbol{x}} \log p_t^{\mathrm{SDE}}(\boldsymbol{x}_t) + \|\nabla_{\boldsymbol{x}} \log p_t^{\mathrm{SDE}}(\boldsymbol{x}_t)\|^2 \right) dt$$

$$= \int_0^T \frac{1}{2}g^2(t)\mathbb{E}_{\boldsymbol{x}_t \sim p_t^{\mathrm{SDE}}(\boldsymbol{x}_t)}\|\boldsymbol{s}(t, \boldsymbol{x}_t) - \nabla_{\boldsymbol{x}} \log p_t^{\mathrm{SDE}}(\boldsymbol{x}_t)\|^2 dt. \tag{90}$$

Therefore

$$
\log p_0^{\mathrm{SDE}}(\boldsymbol{x}_0) = \log p_T^{\mathrm{SDE}}(\boldsymbol{x}_T) - \int_0^T \left( - \operatorname{div}_{\boldsymbol{x}} f(t, \boldsymbol{x}_t) - \frac{1}{2} g^2(t) s(t, \boldsymbol{x}_t) \right) dt
$$
$$
- \int_0^T g(t) s(t, \boldsymbol{x}_t)^T d\overline{\mathrm{W}}_t - \mathrm{X}, \tag{91}
$$

where

$$
\mathbb{E}\mathrm{X} = \int_0^T \frac{1}{2} g^2(t) \mathbb{E}_{\boldsymbol{x}_t \sim p_t^{\mathrm{SDE}}(\boldsymbol{x}_t)} \| s(t, \boldsymbol{x}_t) - \nabla_{\boldsymbol{x}} \log p_t^{\mathrm{SDE}}(\boldsymbol{x}_t) \|^2 dt \geq 0. \tag{92}
$$

$\square$

## H  PROOF OF THEOREM 5

**In the following sections, we assume the linear drift SDE** Equation 1. In Theorem 5 we explicitly assume Gaussian forward transition densities, which are only guaranteed in the linear drift SDE.

**Theorem 5** (Mode-tracking ODE). *Let $t \in (0, T]$ and $\boldsymbol{x}_t \in \mathbb{R}^D$ a noisy sample. If there exists a smooth curve $s \mapsto \boldsymbol{y}_s$ such that $p(\boldsymbol{y}_s | \boldsymbol{x}_t) = \max_{\boldsymbol{x}_s} p(\boldsymbol{x}_s | \boldsymbol{x}_t)$, then $\boldsymbol{y}_t = \boldsymbol{x}_t$ and for $s < t$*

$$
\frac{d}{ds} \boldsymbol{y}_s = f(s) \boldsymbol{y}_s - g^2(s) \boxed{\nabla_{\boldsymbol{y}} \log p_s(\boldsymbol{y}_s)} - \frac{1}{2} g^2(s) \boxed{\boldsymbol{A}(s, \boldsymbol{y}_s)^{-1}} \boxed{\nabla_{\boldsymbol{y}} \Delta_{\boldsymbol{y}} \log p_s(\boldsymbol{y}_s)}, \tag{16}
$$

*where $\boldsymbol{A}(s, \boldsymbol{y}) = \left( \nabla_{\boldsymbol{y}}^2 \log p_s(\boldsymbol{y}) - \psi(s) \boldsymbol{I}_D \right)$, $\psi(s) = \frac{1}{\sigma_s^2} \frac{e^{\lambda_t}}{e^{\lambda_s} - e^{\lambda_t}}$, and $\Delta_{\boldsymbol{y}} = \sum_i \frac{\partial^2}{\partial y_i^2}$ is the Laplace operator. In particular:*

$$
p(\boldsymbol{y}_0 | \boldsymbol{x}_t) = \max_{\boldsymbol{x}_0} p(\boldsymbol{x}_0 | \boldsymbol{x}_t). \tag{17}
$$

Note that without assuming invertibility of $\boldsymbol{A}$, Equation 16 becomes

$$
\boldsymbol{A}(s, \boldsymbol{y}_s) \left( \dot{\boldsymbol{y}}_s - f(s) \boldsymbol{y}_s + g^2(s) \nabla_{\boldsymbol{y}} \log p_s(\boldsymbol{y}_s) \right) = -\frac{1}{2} g^2(s) \nabla_{\boldsymbol{y}} \Delta_{\boldsymbol{y}} \log p_s(\boldsymbol{y}_s) \tag{93}
$$

*Proof.* We begin by noting that for linear SDE (equation 1) $p_{t|s}$ is Gaussian for $s < t$ and therefore

$$
\log p_{s|t}(\boldsymbol{y}_s | \boldsymbol{x}_t) = \log p_{t|s}(\boldsymbol{x}_t | \boldsymbol{y}_s) + \log p_s(\boldsymbol{y}_s) - \log p_t(\boldsymbol{x}_t)
$$
$$
= C - \frac{\|\boldsymbol{x}_t - \tilde{f}(s) \boldsymbol{y}_s\|^2}{2\tilde{g}^2(s)} + \log p_s(\boldsymbol{y}_s) - \log p_t(\boldsymbol{x}_t), \tag{94}
$$

where $\tilde{f}(s) = \frac{\alpha_t}{\alpha_s}$ and $\tilde{g}^2(s) = \sigma_t^2 - \tilde{f}^2(s) \sigma_s^2$ (See Appendix A.1 in Kingma et al. (2021)). Since $p_{s|t}(\boldsymbol{y}_s | \boldsymbol{x}_t) = \max_{\boldsymbol{x}_s} p_{s|t}(\boldsymbol{x}_s | \boldsymbol{x}_t)$, it must hold that

$$
\nabla_{\boldsymbol{y}_s} \log p_{s|t}(\boldsymbol{y}_s | \boldsymbol{x}_t) = 0 \text{ for all } s < t. \tag{95}
$$

Therefore

$$
\frac{d}{ds} \left( \nabla_{\boldsymbol{y}_s} \log p_{s|t}(\boldsymbol{y}_s | \boldsymbol{x}_t) \right) = 0 \text{ for all } s < t. \tag{96}
$$

From Equation 94 we have

$$
\nabla_{\boldsymbol{y}_s} \log p_{s|t}(\boldsymbol{y}_s | \boldsymbol{x}_t) = \nabla_{\boldsymbol{y}_s} \log p_s(\boldsymbol{y}_s) + \frac{\tilde{f}(s)}{\tilde{g}^2(s)} \left( \boldsymbol{x}_t - \tilde{f}(s) \boldsymbol{y}_s \right) = 0 \tag{97}
$$

and thus

$$
\frac{d}{ds} \left( \nabla_{\boldsymbol{y}_s} \log p_s(\boldsymbol{y}_s) - \psi(s) \boldsymbol{y}_s + \phi(s) \boldsymbol{x}_t \right) = 0, \tag{98}
$$

where $\psi(s) = \frac{\tilde{f}^2(s)}{\tilde{g}^2(s)}$ and $\phi(s) = \frac{\tilde{f}(s)}{\tilde{g}^2(s)}$. Note that

$$
\frac{d}{ds} \nabla_{\boldsymbol{y}_s} \log p_s(\boldsymbol{y}_s) = \frac{\partial}{\partial s} \nabla_{\boldsymbol{y}_s} \log p_s(\boldsymbol{y}_s) + \nabla_{\boldsymbol{y}_s}^2 \log p_s(\boldsymbol{y}_s) \dot{\boldsymbol{y}}_s \tag{99}
$$

and we can use Equation 26 to re-write the first term

$$
\begin{aligned}
\frac{\partial}{\partial s}\nabla_{\boldsymbol{y}_s}\log p_s(\boldsymbol{y}_s) &= \nabla_{\boldsymbol{y}_s}\frac{\partial}{\partial s}\log p_s(\boldsymbol{y}_s)\\
&= \nabla_{\boldsymbol{y}_s}\left(-f(s)D+\frac{1}{2}g^2(s)\Delta_{\boldsymbol{y}_s}\log p_s(\boldsymbol{y}_s)-\nabla_{\boldsymbol{y}_s}\log p_s(\boldsymbol{y}_s)^T(f(s)\boldsymbol{y}_s-\frac{1}{2}g^2(s)\nabla_{\boldsymbol{y}_s}\log p_s(\boldsymbol{y}_s))\right)\\
&= \frac{1}{2}g^2(s)\nabla_{\boldsymbol{y}_s}\Delta_{\boldsymbol{y}_s}\log p_s(\boldsymbol{y}_s)-f(s)\nabla_{\boldsymbol{y}}^2\log p_s(\boldsymbol{y}_s)\boldsymbol{y}_s\\
&\quad -f(s)\nabla_{\boldsymbol{y}}\log p_s(\boldsymbol{y}_s)+g^2(s)\nabla_{\boldsymbol{y}}^2\log p_s(\boldsymbol{y}_s)\nabla_{\boldsymbol{y}}\log p_s(\boldsymbol{y}_s)\\
&= \frac{1}{2}g^2(s)\nabla_{\boldsymbol{y}_s}\Delta_{\boldsymbol{y}_s}\log p_s(\boldsymbol{y}_s)-f(s)\nabla_{\boldsymbol{y}}\log p_s(\boldsymbol{y}_s)\\
&\quad -\nabla_{\boldsymbol{y}}^2\log p_s(\boldsymbol{y}_s)\left(f(s)\boldsymbol{y}_s-g^2(s)\nabla_{\boldsymbol{y}}\log p_s(\boldsymbol{y}_s)\right)
\end{aligned}
\tag{100}
$$

and thus

$$
\begin{aligned}
\frac{d}{ds}\nabla_{\boldsymbol{y}_s}\log p_s(\boldsymbol{y}_s) &= \frac{1}{2}g^2(s)\nabla_{\boldsymbol{y}_s}\Delta_{\boldsymbol{y}_s}\log p_s(\boldsymbol{y}_s)-f(s)\nabla_{\boldsymbol{y}}\log p_s(\boldsymbol{y}_s)\\
&\quad +\nabla_{\boldsymbol{y}}^2\log p_s(\boldsymbol{y}_s)\left(\dot{\boldsymbol{y}}_s-f(s)\boldsymbol{y}_s+g^2(s)\nabla_{\boldsymbol{y}}\log p_s(\boldsymbol{y}_s)\right)
\end{aligned}
\tag{101}
$$

For the remaining terms we first note

$$
\tilde{f}(s)=\frac{\alpha_t}{\alpha_s}=\exp\{\log\alpha_t-\log\alpha_s\}=\exp\{\int_s^t\frac{d}{du}\log\alpha_u\}=\exp\{\int_s^t f(u)\}
\tag{102}
$$

and in particular $\frac{d}{ds}\log\tilde{f}(s)=-f(s)$. Similarly

$$
\begin{aligned}
\tilde{g}^2(s)=\sigma_t^2-\tilde{f}^2(s)\sigma_s^2 &=\alpha_t^2\left(\frac{\sigma_t^2}{\alpha_t^2}-\frac{\sigma_s^2}{\alpha_s^2}\right)=\alpha_t^2\left(e^{-\lambda_t}-e^{-\lambda_s}\right)=\alpha_t^2\int_s^t\frac{d}{du}e^{-\lambda_u}du\\
&=\alpha_t^2\int_s^t\left(-\frac{d\lambda_u}{du}\right)e^{-\lambda_u}du=\alpha_t^2\int_s^t\left(-\frac{d\lambda_u}{du}\right)\frac{\sigma_u^2}{\alpha_u^2}du=\alpha_t^2\int_s^t\frac{g^2(u)}{\alpha_u^2}du\\
&=\int_s^t\tilde{f}^2(u)g^2(u)du
\end{aligned}
\tag{103}
$$

and in particular $\frac{d}{ds}\log\tilde{g}^2(s)=\frac{1}{\tilde{g}^2(s)}\frac{d}{ds}\tilde{g}^2(s)=-\psi(s)g^2(s)$. Therefore

$$
\begin{aligned}
\frac{d}{ds}\left(-\psi(s)\boldsymbol{y}_s+\phi(s)\boldsymbol{x}_t\right) &= -\psi'(s)\boldsymbol{y}_s-\psi(s)\dot{\boldsymbol{y}}_s+\phi'(s)\boldsymbol{x}_t\\
&= -\psi(s)\dot{\boldsymbol{y}}_s+\phi'(s)\boldsymbol{x}_t-\left(\phi'(s)\tilde{f}(s)-f(s)\psi(s)\right)\boldsymbol{y}_s\\
&= -\psi(s)\left(\dot{\boldsymbol{y}}_s-f(s)\boldsymbol{y}_s\right)+\phi'(s)\left(\boldsymbol{x}_t-\tilde{f}(s)\boldsymbol{y}_s\right)\\
&= -\psi(s)\left(\dot{\boldsymbol{y}}_s-f(s)\boldsymbol{y}_s\right)+\phi(s)\frac{d}{ds}\left(\log\phi(s)\right)\left(\boldsymbol{x}_t-\tilde{f}(s)\boldsymbol{y}_s\right).
\end{aligned}
\tag{104}
$$

From Equation 97, we have

$$
\phi(s)\left(\boldsymbol{x}_t-\tilde{f}(s)\boldsymbol{y}_s\right)=-\nabla_{\boldsymbol{y}}\log p_s(\boldsymbol{y}_t)
\tag{105}
$$

and

$$
\frac{d}{ds}\left(\log\phi(s)\right)=\frac{d}{ds}\log\tilde{f}(s)-\frac{d}{ds}\log\tilde{g}^2(s)=-f(s)+\psi(s)g^2(s).
\tag{106}
$$

Thus

$$
\begin{aligned}
\frac{d}{ds}\left(-\psi(s)\boldsymbol{y}_s+\phi(s)\boldsymbol{x}_t\right) &= -\psi(s)\left(\dot{\boldsymbol{y}}_s-f(s)\boldsymbol{y}_s\right)+\left(f(s)-\psi(s)g^2(s)\right)\nabla_{\boldsymbol{y}}\log p_s(\boldsymbol{y}_s)\\
&= -\psi(s)\left(\dot{\boldsymbol{y}}_s-f(s)\boldsymbol{y}_s+g^2(s)\nabla_{\boldsymbol{y}}\log p_s(\boldsymbol{y}_s)\right)+f(s)\nabla_{\boldsymbol{y}}\log p_s(\boldsymbol{y}_s).
\end{aligned}
\tag{107}
$$

Putting it all together, we have

$$
\begin{aligned}
0 &= \frac{d}{ds}\left(\nabla_{\boldsymbol{y}}\log p(\boldsymbol{y}_s|\boldsymbol{x}_t)\right)\\
&= \frac{1}{2}g^2(s)\nabla_{\boldsymbol{y}}\Delta_{\boldsymbol{y}}\log p_s(\boldsymbol{y}_s) - f(s)\nabla_{\boldsymbol{y}}\log p_s(\boldsymbol{y}_s)\\
&\quad + \nabla_{\boldsymbol{y}}^2\log p_s(\boldsymbol{y}_s)\left(\dot{\boldsymbol{y}}_s - f(s)\boldsymbol{y}_s + g^2(s)\nabla_{\boldsymbol{y}}\log p_s(\boldsymbol{y}_s)\right)\\
&\quad - \psi(s)\left(\dot{\boldsymbol{y}}_s - f(s)\boldsymbol{y}_s + g^2(s)\nabla_{\boldsymbol{y}}\log p_s(\boldsymbol{y}_s)\right) + f(s)\nabla_{\boldsymbol{y}}\log p_s(\boldsymbol{y}_s)\\
&= \frac{1}{2}g^2(s)\nabla_{\boldsymbol{y}}\Delta_{\boldsymbol{y}}\log p_s(\boldsymbol{y}_s) + \left(\nabla_{\boldsymbol{y}}^2\log p_s(\boldsymbol{y}_s) - \psi(s)\boldsymbol{I}_D\right)\left(\dot{\boldsymbol{y}}_s - f(s)\boldsymbol{y}_s + g^2(s)\nabla_{\boldsymbol{y}}\log p_s(\boldsymbol{y}_s)\right),
\end{aligned}
\tag{108}
$$

or equivalently for $\boldsymbol{A}(s,\boldsymbol{y}) = \nabla_{\boldsymbol{y}}^2\log p_s(\boldsymbol{y}) - \psi(s)\boldsymbol{I}_D$

$$
\dot{\boldsymbol{y}}_s = f(s)\boldsymbol{y}_s - g^2(s)\nabla_{\boldsymbol{y}}\log p_s(\boldsymbol{y}_s) - \frac{1}{2}g^2(s)\boldsymbol{A}(s,\boldsymbol{y}_s)^{-1}\nabla_{\boldsymbol{y}}\Delta_{\boldsymbol{y}}\log p_s(\boldsymbol{y}_s)
\tag{109}
$$

if $\boldsymbol{A}(s,\boldsymbol{y}_s)$ is invertible for all $s < t$ and

$$
\begin{aligned}
\psi(s) &= \frac{\tilde{f}^2(s)}{\tilde{g}^2(s)} = \frac{\alpha_t^2}{\alpha_s^2\left(\sigma_t^2 - \tilde{f}^2(s)\sigma_s^2\right)} = \frac{\alpha_t^2}{\alpha_s^2\sigma_t^2 - \alpha_t^2\sigma_s^2} = \frac{1}{\alpha_s^2}\frac{1}{\frac{\sigma_t^2}{\alpha_t^2} - \frac{\sigma_s^2}{\alpha_s^2}}\\
&= \frac{1}{\alpha_s^2}\frac{1}{e^{-\lambda_t} - e^{-\lambda_s}} = \frac{e^{\lambda_s}}{\alpha_s^2}\frac{e^{\lambda_t}}{e^{\lambda_s} - e^{\lambda_t}} = \frac{1}{\sigma_s^2}\frac{e^{\lambda_t}}{e^{\lambda_s} - e^{\lambda_t}}.
\end{aligned}
\tag{110}
$$

$\square$

# I  MODE-SEEKING ODE IN THE GAUSSIAN CASE

We will prove the claims from Remark 1. We recall it for completeness.

**Remark 1** (High-density ODE or HD-ODE)**.** *If $p_0$ is Gaussian, then equation 16 becomes*

$$
d\boldsymbol{y}_s = \left(f(s)\boldsymbol{y}_s - g^2(s)\,\boxed{\nabla_{\boldsymbol{x}}\log p_s(\boldsymbol{y}_s)}\,\right)ds,
\tag{18}
$$

*i.e. the drift term of reverse SDE equation 2. If $\boldsymbol{y}_t = \boldsymbol{x}_t$ and equation 18 holds for $s < t$, then*

$$
\boldsymbol{y}_0 = \arg\max_{\boldsymbol{x}_0} p(\boldsymbol{x}_0|\boldsymbol{x}_t) + \mathcal{O}(e^{-\lambda_{\max}}).
\tag{19}
$$

*Proof.* We first note that when $p_0$ Gaussian and the SDE is linear (equation 1) then $p_s$ are Gaussian $\forall s$. In particular $\nabla_{\boldsymbol{x}}\Delta_{\boldsymbol{x}}\log p_s(\boldsymbol{x}) = 0$ for al $s \in [0,T]$ and $\boldsymbol{x} \in \mathbb{R}^D$. Therefore equation 16 becomes equation 18. We will now study $\boldsymbol{y}_s$ following equation 18. Recalling Equation 101:

$$
\begin{aligned}
\frac{d}{ds}\nabla_{\boldsymbol{y}}\log p_s(\boldsymbol{y}_s) &= \frac{1}{2}g^2(s)\overset{0}{\cancel{\nabla_{\boldsymbol{y}}\Delta_{\boldsymbol{y}}\log p_s(\boldsymbol{y}_s)}} - f(s)\nabla_{\boldsymbol{y}}\log p_s(\boldsymbol{y}_s)\\
&\quad + \nabla_{\boldsymbol{y}}^2\log p_s(\boldsymbol{y}_s)\left(\dot{\boldsymbol{y}}_s - f(s)\boldsymbol{y}_s + g^2(s)\nabla_{\boldsymbol{y}}\log p_s(\boldsymbol{y}_s)\right)\\
&= -f(s)\nabla_{\boldsymbol{y}}\log p_s(\boldsymbol{y}_s) + \nabla_{\boldsymbol{y}}^2\log p_s(\boldsymbol{y}_s)\left(\dot{\boldsymbol{y}}_s - f(s)\boldsymbol{y}_s + g^2(s)\nabla_{\boldsymbol{y}}\log p_s(\boldsymbol{y}_s)\right).
\end{aligned}
\tag{111}
$$

If we then assume Equation 18, we have

$$
\frac{d}{ds}\nabla_{\boldsymbol{y}}\log p_s(\boldsymbol{y}_s) = -f(s)\nabla_{\boldsymbol{y}}\log p_s(\boldsymbol{y}_s)
\tag{112}
$$

and to simplify further, using the fact that $f(s) = \frac{d}{ds}\log\alpha_s$

$$
\frac{d}{ds}\left(\alpha_s\nabla_{\boldsymbol{y}}\log p_s(\boldsymbol{y}_s)\right) = \frac{d}{ds}\alpha_s\nabla_{\boldsymbol{y}}\log p_s(\boldsymbol{y}_s) + \alpha_s\frac{d}{ds}\nabla_{\boldsymbol{y}}\log p_s(\boldsymbol{y}_s) = 0.
\tag{113}
$$

Hence, for $\boldsymbol{y}_s$ satisfying Equation 18, we have

$$\alpha_s \nabla_{\boldsymbol{y}} \log p_s(\boldsymbol{y}_s) = \alpha_t \nabla_{\boldsymbol{y}} \log p_t(\boldsymbol{y}_t) \ \text{ for all } \ s < t. \tag{114}$$

We can thus rewrite

$$\begin{aligned}
\dot{\boldsymbol{y}}_s &= f(s)\boldsymbol{y}_s - g^2(s)\nabla_{\boldsymbol{y}} \log p_s(\boldsymbol{y}_s) \\
&= f(s)\boldsymbol{y}_s - \frac{g^2(s)}{\alpha_s}\alpha_s \nabla_{\boldsymbol{y}} \log p_s(\boldsymbol{y}_s) \\
&= f(s)\boldsymbol{y}_s - \frac{g^2(s)}{\alpha_s}\alpha_t \nabla_{\boldsymbol{y}} \log p_t(\boldsymbol{y}_t).
\end{aligned} \tag{115}$$

Furthermore

$$\begin{aligned}
\frac{d}{ds}\left(\frac{\boldsymbol{y}_s}{\alpha_s}\right) &= \frac{\dot{\boldsymbol{y}}_s \alpha_s - \boldsymbol{y}_s \alpha_s'}{\alpha_s^2} = \frac{\left(\frac{\alpha_s'}{\alpha_s}\boldsymbol{y}_s - \frac{g^2(s)}{\alpha_s}\alpha_t \nabla_{\boldsymbol{y}} \log p_t(\boldsymbol{y}_t)\right)\alpha_s - \boldsymbol{y}_s \alpha_s'}{\alpha_s^2} \\
&= -\frac{g^2(s)}{\alpha_s^2}\alpha_t \nabla_{\boldsymbol{y}} \log p_t(\boldsymbol{y}_t) = \frac{d\lambda_s}{ds}e^{-\lambda_s}\alpha_t \nabla_{\boldsymbol{y}} \log p_t(\boldsymbol{y}_t)
\end{aligned} \tag{116}$$

and we can solve

$$\begin{aligned}
\frac{\boldsymbol{y}_t}{\alpha_t} - \frac{\boldsymbol{y}_0}{\alpha_0} &= \int_0^t \left(\frac{d\lambda_s}{ds}e^{-\lambda_s}\alpha_t \nabla_{\boldsymbol{y}} \log p_t(\boldsymbol{y}_t)\right)ds = \left(\int_0^t \frac{d\lambda_s}{ds}e^{-\lambda_s}ds\right)\alpha_t \nabla_{\boldsymbol{y}} \log p_t(\boldsymbol{y}_t) \\
&= \left(\int_{\lambda_{\max}}^{\lambda_t} e^{-\lambda}d\lambda\right)\alpha_t \nabla_{\boldsymbol{y}} \log p_t(\boldsymbol{y}_t) = \left(e^{-\lambda_{\max}} - e^{-\lambda_t}\right)\alpha_t \nabla_{\boldsymbol{y}} \log p_t(\boldsymbol{y}_t).
\end{aligned} \tag{117}$$

Leveraging that $\alpha_0 = 1$, we get

$$\begin{aligned}
\boldsymbol{y}_0 &= \frac{\boldsymbol{y}_t}{\alpha_t} + e^{-\lambda_t}\alpha_t \log p_t(\boldsymbol{y}_t) - e^{-\lambda_{\max}}\alpha_t \nabla_{\boldsymbol{y}} \log p_t(\boldsymbol{y}_t) \\
&= \frac{\boldsymbol{y}_t + \sigma_t^2 \nabla_{\boldsymbol{x}} \log p_t(\boldsymbol{x}_t)}{\alpha_t} - e^{-\lambda_{\max}}\alpha_t \nabla_{\boldsymbol{x}} \log p_t(\boldsymbol{x}_t) \\
&= \mathbb{E}\left[\boldsymbol{x}_0 | \boldsymbol{x}_t\right] - e^{-\lambda_{\max}}\alpha_t \nabla_{\boldsymbol{x}} \log p_t(\boldsymbol{x}_t) \\
&= \arg\max_{\boldsymbol{x}_0} p(\boldsymbol{x}_0 | \boldsymbol{x}_t) + \mathcal{O}(e^{-\lambda_{\max}}).
\end{aligned} \tag{118}$$

$\square$

## J  NON-SMOOTH MODE-TRACKING CURVE

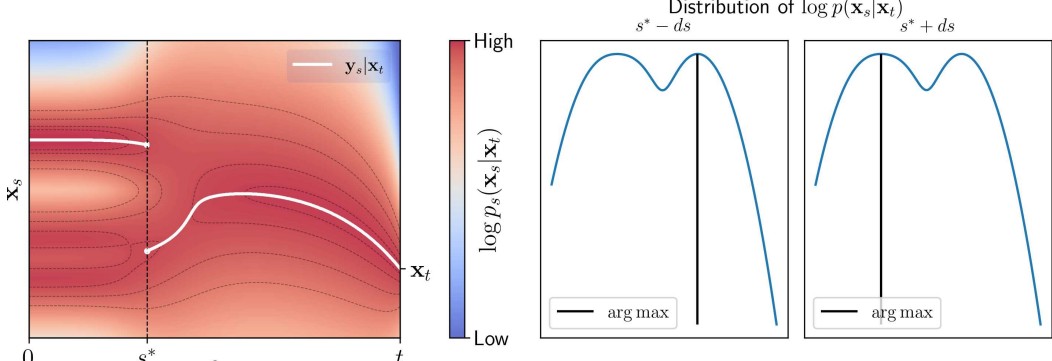

Figure 11: **The mode tracking curve need not be continuous.** Left: mode-tracking curve with a discontinuous jump at $s^*$. Right: distribution of $\log p_{s|t}(\boldsymbol{x}_s|\boldsymbol{x}_t)$ for values around $s^*$. At $s = s^*$ the arg max changes discontinuously.

An important assumption in Theorem 5 is that for a fixed $t \in (0, T]$ and a noisy point $\boldsymbol{x}_t \in \mathbb{R}^D$, there exists a smooth curve $s \mapsto \boldsymbol{y}_s$ such that

$$p_{s|t}(\boldsymbol{y}_s|\boldsymbol{x}_t) = \max_{\boldsymbol{x}_s} p_{s|t}(\boldsymbol{x}_s|\boldsymbol{x}_t). \tag{119}$$

It is an assumption that need not hold. To demonstrate we define the data distribution as 1D mixture of 3 gaussians $p = \sum_{i=1}^{3} w_i \mathcal{N}(\mu_i, \sigma^2)$, where $\mu_1 = -2.5$, $\mu_2 = -1.5$, $\mu_3 = 1$ and $\sigma^2 = 0.1$ and weights $w_1 = w_2 = 0.274$ and $w_3 = 0.45$. We model the distrbution with a VP-SDE (Song et al., 2021c), where $\sigma_t^2 = \frac{1}{1+e^{\lambda_t}} = 1 - \alpha_t^2$. We then choose $\boldsymbol{x}_t = -2.5$ and $t$ such that $\lambda_t = -8$ and visualize $\log p(\boldsymbol{x}_s|\boldsymbol{x}_t)$ for all $s < t$ and all $\boldsymbol{s}_t \in [-4, 3.5]$ and the mode-tracking curve $\boldsymbol{y}_s|\boldsymbol{x}_t$ in white (Figure 11 left).

The mode-tracking curve exhibits a discontinuous jump at $s^*$ such that $\lambda_{s^*} \approx 1.28$. The distribution $p_{s|t}(\boldsymbol{x}_s|\boldsymbol{x}_t)$ has the mode at $\boldsymbol{x}_s \approx 0.86$ for $s = s^* - ds$ and at $\boldsymbol{x}_s \approx -1.81$ for $s = s^* + ds$ (Figure 11 right).

## K    COST OF MODE-TRACKING

Evaluation of the drift of Equation 16 requires evaluating

$$\underbrace{\boldsymbol{A}(s, \boldsymbol{y}_s)^{-1}}_{\text{Hessian factor}} \underbrace{\nabla_{\boldsymbol{y}} \Delta_{\boldsymbol{y}} \log p_s(\boldsymbol{y}_s)}_{\text{Laplacian factor}},$$

where $\boldsymbol{A}(s, \boldsymbol{y}) = \left(\nabla_{\boldsymbol{y}}^2 \log p_s(\boldsymbol{y}) - \psi(s)\boldsymbol{I}_D\right)$, $\psi(s) = \frac{1}{\sigma_s^2} \frac{e^{\lambda_t}}{e^{\lambda_s} - e^{\lambda_t}}$, and $\Delta_{\boldsymbol{y}} = \sum_i \frac{\partial^2}{\partial y_i^2}$ is the Laplace operator. We will discuss the factors separately assuming that we use a model $\boldsymbol{s}_\theta(t, \boldsymbol{x}) \approx \nabla_{\boldsymbol{x}} \log p_t(\boldsymbol{x})$.

**Hessian factor**    To evaluate $\boldsymbol{A}(s, \boldsymbol{y})$, we need to estimate $\nabla_{\boldsymbol{y}}^2 \log p_s(\boldsymbol{y})$, which is the Jacobian matrix of the score function w.r.t. spatial argument $\boldsymbol{y}$. This can be done using automatic differentiation and it requires $D$ Jacobian-vector products (JVPs), where $D$ is the dimensionality of the data and each JVP is roughly twice as expensive as score function evaluation (Meng et al., 2021). In summary, evaluating $\boldsymbol{A}(s, \boldsymbol{y}_s)^{-1}$ requires roughly $2D$ score function evaluations plus the inversion of a $D \times D$ matrix, which is $\mathcal{O}(D^3)$.

**Laplacian factor**    Evaluation of $\nabla_{\boldsymbol{y}} \Delta_{\boldsymbol{y}} \log p_s(\boldsymbol{y}_s) = \nabla_{\boldsymbol{y}} \operatorname{div}_{\boldsymbol{y}} \nabla_y \log p_s(\boldsymbol{y}_s)$ requires evaluating the gradient of the divergence of the score function. Exact evaluation of the divergence would again require $D$ JVPs (Meng et al., 2021). However, one might approximate it with a single JVP using the Hutchinson's trick (Hutchinson, 1989; Grathwohl et al., 2019). One can thus approximate $\nabla_{\boldsymbol{y}} \Delta_{\boldsymbol{y}} \log p_s(\boldsymbol{y}_s) = \nabla_{\boldsymbol{y}} \operatorname{div}_{\boldsymbol{y}} \nabla_y \log p_s(\boldsymbol{y}_s)$ using a single JVP followed by a backward pass.

In summary, the bottleneck of the evaluation of $\boldsymbol{A}(s, \boldsymbol{y}_s)^{-1} \nabla_{\boldsymbol{y}} \Delta_{\boldsymbol{y}} \log p_s(\boldsymbol{y}_s)$ is the evaluation of $\boldsymbol{A}(s, \boldsymbol{y}_s)^{-1}$, which scales worse than linearly with the dimension of the data. For example, for CIFAR10 data, the evaluation of each step of Equation 16 would be at least 6000x more expensive than the evaluation of the score function. For 256x256 images it would be roughly 400000x more expensive.

## L    PROOF OF LEMMA 1

*Proof.* From Equation 26, we have that

$$\frac{\partial}{\partial t} \log p_t(\boldsymbol{z}) = -\operatorname{div}_{\boldsymbol{z}} f_1(t, \boldsymbol{z}) - \nabla_{\boldsymbol{z}} \log p_t(\boldsymbol{z})^T f_1(t, \boldsymbol{z}).$$

Therefore

$$\begin{aligned}
\frac{d}{dt} \log p_t(\boldsymbol{z}_t) &= \frac{\partial}{\partial t} \log p_t(\boldsymbol{z}_t) + \nabla_{\boldsymbol{x}} \log p_t(\boldsymbol{z}_t)^T \frac{d}{dt} \boldsymbol{z}_t \\
&= -\operatorname{div}_{\boldsymbol{z}} f_1(t, \boldsymbol{z}_t) + \nabla_{\boldsymbol{x}} \log p_t(\boldsymbol{z}_t)^T \left(f_2(t, \boldsymbol{z}_t) - f_1(t, \boldsymbol{z}_t)\right).
\end{aligned}$$

$\square$

## M  CIFAR MODELS HYPERPARAMETERS

In subsection 3.2 we train diffusion models on CIFAR10 data. Specifically, these models are Variance Preserving (VP) SDEs with a linear log-SNR noise schedule and $\varepsilon$-parametrization (where the model is directly conditioned on $\lambda = \log \mathrm{SNR}(t)$ as opposed to $t$, as suggested by Kingma & Gao (2024)). $\varepsilon_\theta$ is parametrized as a UNET using the implementation from `docs.kidger.site/equinox/examples/unet/` with hyperparameters: `is_biggan=True`, `dim_mults=(1, 2, 2, 2)`, `hidden_size=128`, `heads=8`, `dim_head=16`, `dropout_rate=0.1`, `num_res_blocks=4`, `attn_resolutions=[16]`; trained for 2M steps, 128 batch size, and the adaptive noise schedule from Kingma & Gao (2024) with EMA weight 0.99.

The two model variants are:

- CIFAR10-ML - trained with maximum likelihood (ML), i.e. unweighted ELBO;
- CIFAR10-SQ - optimized for **S**ample **Q**uality, i.e. trained with weighted ELBO with $w(\lambda) = \mathrm{sigmoid}(-\lambda + 2)$ as recommended by Kingma & Gao (2024).

## N  QUANTITATIVE ANALYSIS OF LIKELIHOODS OF SAMPLES GENERATED WITH ALGORITHM 1

In Table 2 we provide the values of $\mathbb{E}[-\log p_0(\boldsymbol{x}_0)]$ (in bits-per-dim) for different models and sampling strategies. In all cases $\log p_0(\boldsymbol{x}_0)$ was estimated using the PF-ODE (Equation 3) to ensure a fair comparison. The values are mean ± one standard deviation. We see that HD sampling (algorithm 1) generates samples with higher density (lower NLL) than regular samples across different models and values of the threshold parameter $t$. Note that for different models, values of the threshold parameter $t$ in HD sampling is in different ranges. This is due to the fact that different models use different SDEs and different noise schedules.

The models used are

- CIFAR10 - Models from subsection 3.2 with hyperparameters as defined in Appendix M. Used 1024 samples for each sampling strategy.
- ImageNet64 - Checkpoint provided by Karras et al. (2022), i.e. Variance Exploding (VE) SDE with a noise schedule satisfying $\sigma = t$. "Original" sampling strategy is the stochastic Heun sampler proposed by the authors. Used 192 samples for each sampling strategy with default hyperparameters.
- FFHQ256 and Church256 - Checkpoints provided by Song et al. (2021c), i.e. VE SDE with exponential noise schedule. "Original" sampling strategy is the Predictor-Corrector sampler recommended by the authors with default hyperparameters. Used 192 samples for each sampling strategy.

## O  STABLE DIFFUSION SAMPLES

In Figure 12 we provide a comparison of regular samples and high-density samples generated with algorithm 1 using the Stable Diffusion v2.1 model (Rombach et al., 2021) for multiple values of the threshold parameter $t$. Interestingly, even though the diffusion process happens in the latent space, we see similar behavior to pixel-space diffusion models discussed in section 5. Specifically, the high-density samples exhibit cartoon-like features or are blurry images, depending on the threshold parameter $t$ value.

## P  WHY DO CARTOONS AND BLURRY IMAGES OCCUPY HIGH-DENSITY REGIONS?

*Local intrinsic dimension* (LID) is a measure of an image's complexity and can be interpreted as "the number of local factors of variation" (Kamkari et al., 2024). For image data, PNG compression

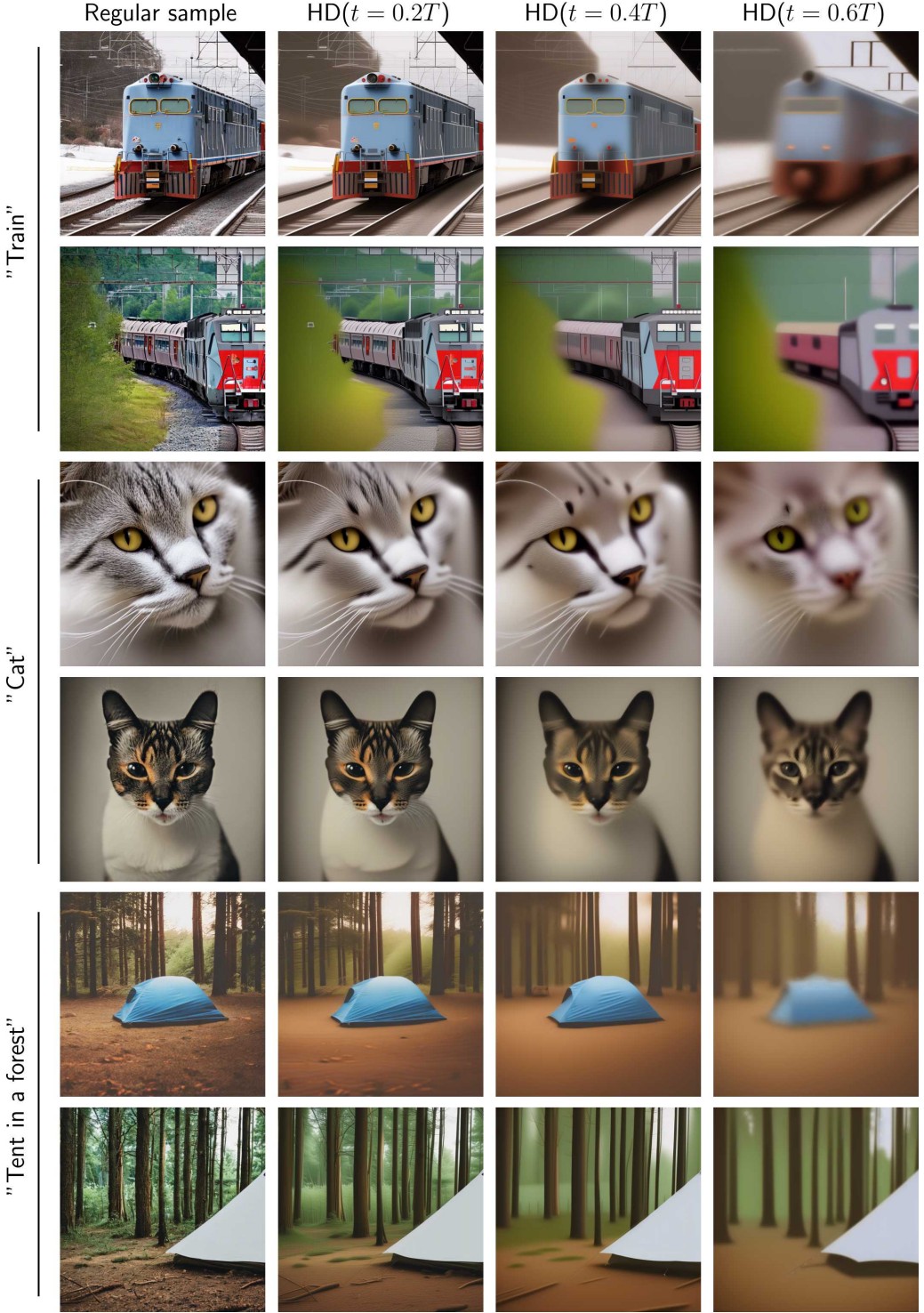

Figure 12: Regular vs High-Density samples on the Stable Diffusion v2.1 model. We added the prefix "A photo of" to each prompt to obtain more realistic images.

| Model | Sampling | NLL (bpd) |
|---|---|---|
| CIFAR10-ML | PF-ODE | $4.17 \pm 0.49$ |
| | Rev-SDE | $4.44 \pm 0.42$ |
| | Rev-SDE (Theorem 1) | $4.30* \pm 0.41$ |
| | HD($t = 0.3T$) | $2.65 \pm 0.62$ |
| | HD($t = 0.45T$) | $1.57 \pm 0.44$ |
| | HD($t = 0.5T$) | $1.25 \pm 0.38$ |
| | HD($t = 0.55T$) | $0.98 \pm 0.36$ |
| | HD($t = 0.7T$) | $0.24 \pm 0.27$ |
| CIFAR10-SQ | PF-ODE | $4.55 \pm 0.46$ |
| | Rev-SDE | $4.23 \pm 0.42$ |
| | Rev-SDE (Theorem 1) | $4.16* \pm 0.42$ |
| | HD($t = 0.3T$) | $2.74 \pm 0.61$ |
| | HD($t = 0.45T$) | $1.61 \pm 0.39$ |
| | HD($t = 0.5T$) | $1.37 \pm 0.34$ |
| | HD($t = 0.55T$) | $1.15 \pm 0.32$ |
| | HD($t = 0.7T$) | $0.46 \pm 0.30$ |
| ImageNet64 | Original | $3.16 \pm 0.81$ |
| (Karras et al., 2022) | HD($t = 0.0125T$) | $-2.10 \pm 0.33$ |
| | HD($t = 0.05T$) | $-2.74 \pm 0.38$ |
| FFHQ256 | Original | $1.01 \pm 0.41$ |
| (Song et al., 2021c) | HD($t = 0.5T$) | $-2.58 \pm 0.65$ |
| | HD($t = 0.3T$) | $-4.01 \pm 0.96$ |
| Church256 | Original | $0.77 \pm 0.23$ |
| (Song et al., 2021c) | HD($t = 0.5T$) | $-0.66 \pm 0.34$ |
| | HD($t = 0.3T$) | $-1.15 \pm 0.15$ |

Table 2: Comparison of NLL (in bits-per-dim) for different models and sampling methods. "*" denotes that the likelihood was estimated with Theorem 1.

size is commonly used as a proxy for LID; that is, higher PNG compression size indicates higher LID (Kamkari et al., 2024).

In Figure 8, we observed a strong correlation between the model's log-likelihood estimation and the image's file size after PNG compression. This relationship can be attributed to a deeper connection between LID and the likelihood of data with varying levels of added noise (Tempczyk et al., 2022; Kamkari et al., 2024). Since diffusion models estimate densities across varying noise levels, their likelihood estimates naturally correlate with LID, providing an intuitive explanation for the observed relationship.

This finding suggests that the diffusion model's (negative) log-likelihood estimates can be interpreted as a measure of the number of local factors of variation. This insight helps explain why simple images, such as cartoons or blurry images, often exhibit higher likelihoods than complex, high-detail images.

Consider the samples generated with Stable Diffusion v2.1 in Figure 13. After zooming in, one can observe that high-density samples exhibit significantly less local detail and thus a lower intrinsic dimension.

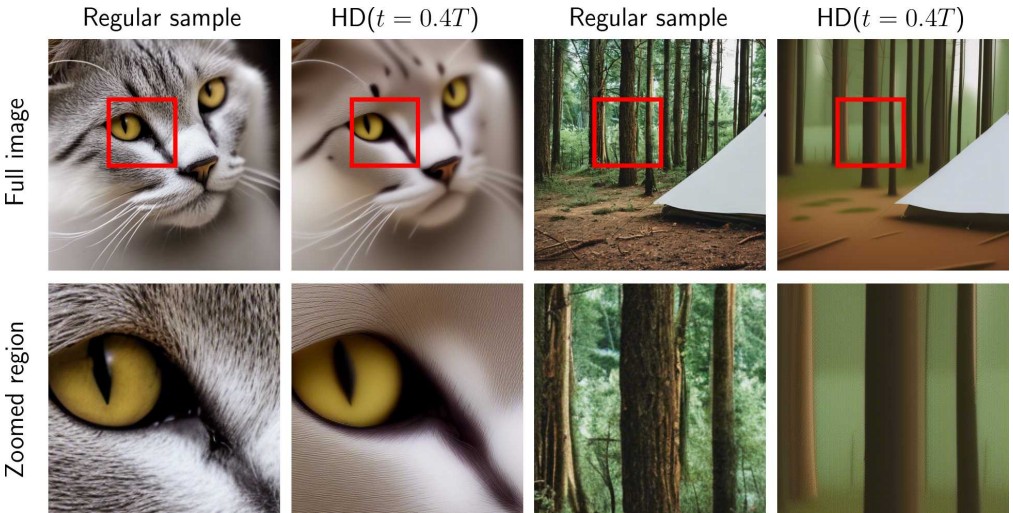

Figure 13: High-density samples have much less detail than regular samples and thus a lower local intrinsic dimension.

## Q    CARTOON GENERATION

Recently, Zhao et al. (2023) proposed to alter the generation in guided diffusion models to bias the sampler to produce cartoon-like images. Specifically, the proposed method modifies classifier-free guidance (Ho & Salimans, 2022) by replacing the intermediate noisy model input corresponding to the null-guidance with the so-called "noise disturbance".

The main difference between our results, Zhao et al. (2023), and other cartoon-generation methods such as Chen et al. (2020), is that our aim was not to build a cartoon generator. Our goal was to study high-density regions of diffusion models and we developed a method (algorithm 1) to efficiently generate points from such regions. The fact that these samples turned out to exhibit cartoon-like features was a surprising discovery that we believe is of interest to a wider research community. Especially in light of a recent report that inspired our study, which attributes the success of guided diffusion models to their ability to avoid low-density regions (Karras et al., 2024a). We show that targeting the highest possible densities is not desirable either in high-quality image generation tasks.

## R    LIMITATIONS

**Stochastic likelihood tracking**    While the likelihood estimation methods introduced in section 2 and section 3 apply to any diffusion SDE, regardless of how the score function is parametrized, there are inherent limitations regarding stochastic sampling. Our novel method for likelihood estimation introduced in Theorem 1 is beneficial as compared to the PF-ODE (Equation 3) as it does not require estimating any higher-order derivatives and is *free* when doing stochastic sampling. However, stochastic sampling usually requires more iterations than deterministic sampling (Song et al., 2021c). Therefore, even though it is roughly twice as expensive to evaluate each step in the augmented PF-ODE (Equation 3) as compared to the augmented reverse SDE (Equation 4), PF-ODE may require fewer total steps.

**Mode-tracking**    In Theorem 5 we derived an exact ODE, which follows the mode exactly. However, there are three limitations:

- The result only holds for SDEs with a linear drift as the proof relies on Gaussian forward transition probabilities;
- Finding the exact mode is only guaranteed when a smooth mode-tracking curve exists and it is difficult to verify in practice and does not always hold (Appendix J);
- The ODE is prohibitively expensive, especially in higher dimensions. See Appendix K.

