# OpenReview forum: "Diffusion Models as Cartoonists: The Curious Case of High Density Regions"
_ICLR.cc/2025/Conference — ICLR 2025 Poster_

### Official Review · Reviewer_KEPG · 2024-10-28

**Soundness:** 3
**Presentation:** 4
**Contribution:** 2
**Rating:** 6
**Confidence:** 3

**Summary:**

The paper presents a perspective on studying likelihoods for diffusion models. They introduce a stochastic way to keep track of the likelihood. They introduce a likelihood maximizing ODE that maximizes the likelihood of a path for any given starting condition. They attempt to argue that they can use this ODE to demonstrate that the high likelihood region of diffusion models results in cartoon like images.

**Strengths:**

- The paper is well written and the use of good figures and colors in equations really helps the presentation
- They present a stochastic way to keep track of the likelihood and a likelihood maximizing ODE

**Weaknesses:**

- Up to section 4.2 everything was very reasonable and nicely introduced, however the use of the ODE in remark 1 for sampling is a big mistake. Real world data is multimodal, for instance CIFAR 10 has at least 10 modes. To use the assumption that the data is gaussian in order to use remark 1 results in incorrect results. Specifically this ODE doesn't sample from the correct target distribution, and therefore we observe those cartoon like images.
- Additionally when comparing the likelihood of images sampled normally or from the HP-ODE the likelihoods were measured using different methods. In section 4.3 it is explained how they compute the likelihood of $p(y_0|x_t)$. Empirically they show that it results in higher likelihood than $p(x_0 |x_t)$, however since these likelihoods are being evaluated in different ways this comparison doesn't make sense. If done correctly the sample $y_0$ would be evaluated using the same way as $x_0$, so we would evaluate the likelihood of the **same distribution**
- The paper has a limited amount of quantitative evidence
- There is no explanation on the hyperparameters that were used during sampling

**Questions:**

Please see weaknesses

---

> ### Author Response · Authors · 2024-11-12
> **Response**
>
> We thank the Reviewer for their scrutiny of our work. We address the raised concerns below.
>
> 1. *To use the assumption that the data is gaussian in order to use remark 1 results in incorrect results.*
>
> In Remark 1 we show that HP-ODE generates the mode in the case of Gaussian data. For non-gaussian data we are no longer guaranteed to find the mode using HP-ODE, but in section 4.3 we empirically show that it generates points with higher likelihoods than regular samples.
>
> **We do not use the Gaussian assumption for real-world data experiments and we explicitly emphasize it in lines 349-351: "Even though equation (18) finds the mode of $p(x_0|x_t)$ only in the Gaussian case [...], we will empirically show that even for non-Gaussian data it finds points with much higher likelihoods than regular samples."**
>
> 2. *Specifically this ODE doesn't sample from the correct target distribution, and therefore we observe those cartoon like images.*
>
> Indeed, HP-ODE does not generate samples from the target distribution, which is evident is Figures 7 and 9. This was precisely the point of "High probability" seeking ODE: To find samples with larger likelihoods than those of regular samples.
>
> This was the question we were exploring (see e.g. lines 028 or 290-291): What happens if we explicitly bias the sampler towards higher likelihood regions? Deviation from the data distribution was expected.
>
> 3. *Additionally when comparing the likelihood of images sampled normally or from the HP-ODE the likelihoods were measured using different methods. In section 4.3 it is explained how they compute the likelihood of. Empirically they show that it results in higher likelihood than , however since these likelihoods are being evaluated in different ways this comparison doesn't make sense.*
>
> Indeed, we explain in section 4.3 that $p(y_0|x_t)$ is evaluated differently from $p(x_0|x_t)$. The sole reason for this is computational. We mention this briefly in a footnote (line 377), but are happy to elaborate on it.
>
> The goal is to generate many samples $x_0 \sim p(x_0|x_t)$ and for each $x_0$ compare $\log p(x_0|x_t)$ with $\log p(y_0|x_t)$. To generate diverse samples $x_0$ conditioned on the same $x_t$, the sampling needs to be done stochastically (deterministic PF-ODE sampling would always generate the same sample). As we show in Equation (20), the main computational challenge is estimating  $\log p_0(x_0) - \log p_t(x_t)$. Since $\log p_0(x_0) - \log p_t(x_t)=-\int_0^t d \log p_s(x_s)$, we can estimate it "for free" using Theorem 1.
>
> 4. *If done correctly the sample would be evaluated using the same way as, so we would evaluate the likelihood of the same distribution and results would not be as reported.*
>
> That said, we can estimate both $\log p(y_0|x_t)$ and $\log p(x_0|x_t)$ using the augmented probability flow ODE as is standard practice (albeit more expensive than our proposed approach).
>
> **Motivated by your concerns we now repeated the experiment** in Section 4.3 (Figure 7), but estimated both $\log p(x_0|x_t)$ and $\log p(y_0|x_t)$ using the same standard method, i.e. PF-ODE (equation 3). Specifically, we use PF-ODE to esimate $\log p_0(x_0)$, $\log p_0(y_0)$ and $\log p_t(x_t)$ and use equation (20) to estimate $\log p(y_0|x_t)$ and $\log p(x_0|x_t)$.
>
> **The results are almost identical to the ones that we reported in the paper. The correlation between two different methods of estimation is at 0.998.** We again find that for all samples $x_0 \sim p(x_0|x_t)$ we have $\log p(x_0|x_t) < \log p(y_0|x_t)$. This is to be expected, because (as we have shown in section 3.2) the difference between $\log p$ estimation using our novel method (Theorem 1) and the standard approach (Equation 3) is very small. However, we chose to use our novel method, because it does not incur any additional cost unlike the standard approach.
>
> 5. *Unfortunately the final conclusion of the paper is just incorrect. It is not true that diffusion models sample away from the high likelihood regions.*
>
> We hope that the above explanations convinced you that our analysis in the paper does in fact demonstrate that regular diffusion model samples have lower likelihoods than 1) samples generated by HP-ODE (Figures 7 and 9); and blurry images (Figure 10).
>
> ---
>
> Many thanks for the opportunity to clarify some key aspects of HP-ODE. We hope we have resolved all your concerns and would be grateful if this could be reflected in an improved score. If you have any further questions or concerns or suggestions, we are more than happy to address them.

---

> > ### Author Response · Authors · 2024-11-13
> > **Response - part 2**
> >
> > 1. *To use the assumption that the data is gaussian [...] results in incorrect results*
> >
> > We would like to clarify further that we derive the mode-tracking ODE without assuming a Gaussian distribution (see Theorem 5). However, as discussed in lines 319-320, evaluating this ODE directly is computationally too costly in practice, so we opt to use an approximation. Specifically, while the simplified ODE (Equation 18) does not guarantee finding the mode, it allows us to evaluate the results empirically, as demonstrated in Section 4.3.
> >
> > The purpose of Remark 1 is to illustrate that, under the assumption of a Gaussian data distribution, the simplified ODE (18) aligns with the optimal solution by naturally eliminating the troublesome higher-order terms. Although our data is not Gaussian, this simplification is appealing, so we apply the simplified ODE even in the non-Gaussian case to explore its empirical performance.
> >
> > We believe that this reasoning is similar in spirit to the "single-point" distribution assumption (Appendix B.2 in [1]). In that work, the authors use the "ideal derivative trick" to derive parameters for the procedure under the assumption that the data distribution consists of a single point. They then empirically verify that this procedure performs well on real-world distributions, which consist of tens of thousands of points.
> >
> > Similarly, we observe that when the data distribution is Gaussian, the troublesome higher-order terms vanish. Empirically, we find that even for non-Gaussian data, this approximation produces high-likelihood points, though not necessarily the exact modes, as would be guaranteed in the Gaussian case.
> >
> > We would be happy to elaborate on this in the paper.
> >
> > 5. *It is not true that diffusion models sample away from the high-likelihood regions.*
> >
> > We take this opportunity to emphasize further that **in high dimensional distributions, samples are low-density by nature** and one always needs a special approach to obtain high-density points (which are not samples anymore).
> > In our paper, we demonstrate this in the context of diffusion, but the general phenomenon of "typical" samples not covering the highest likelihood regions is more widespread [2, 3].
> >
> > **The challenge was to design a procedure, which will find high likelihood samples and the qualitative analysis of these samples revealed surprising insights.**
> > However, **the existence of samples with likelihoods higher than those of typical samples was to be expected and can be illustrated with an example below.**
> >
> > Consider the example of a Gaussian mixture.
> > It is informative because 1) it can approximate any distribution (given enough modes), and 2) we know exactly where the mode is (mean of the component with the highest weight).
> > We will consider a $D$-dimensional mixture with $K$ components of equal variance with the first component having the highest weight.
> >
> > Concretely: $p=\sum_i^K w_i \mathcal{N}(\mu_i, I_D)$, where $w_1=\frac{2}{K+1}$ and $w_i=\frac{1}{K + 1}$ for $i\geq 2$;
> > and $\mu_i$ the $i$-th row of $I_D$, i.e. the $j$-th element of $\mu_i$ is $\delta_i^j$.
> >
> > We now sample $N=8192$ points from that mixture, evaluate their loglikelihoods, and compare them with the loglikelihood of the mode ($\mu_1$).
> > For $D=50$ and $K=8$, we find that the "typical" values of loglikelihoods are in the range $[-96.03, -55.07]$, whereas the loglikelihood of the mode is $-45.25$, which is clearly larger than the largest "typical" value.
> >
> > The discrepancy grows as we increase the dimension $D$.
> > For example for $D=3 * 32 * 32$ (dimensionality corresponding to CIFAR10 images) and for a larger number of components $K=512$, the typical range of loglikelihood is $[-4490.378, -4237.26]$ and the loglikelihood of the mode is $-2822.286$.
> > In this case the difference between the loglikelihood of the mode and the highest typical value is over $5$ times larger than the difference between the highest and lowest typical values.
> >
> > We hope that this additional example showed that it is not uncommon in high dimensional distributions that there exist high-likelihood points that are never sampled.
> > **This also shows that there are no contradictions in our findings.** The cartoon-like or blurry images do have high likelihoods, but they are not "typical" and therefore do not appear in regular sampling.
> >
> > Thank you for your feedback that has helped us clarify these aspects.
> >
> > ---
> >
> > [1] Dockhorn et al. "GENIE: Higher-Order Denoising Diffusion Solvers" (NeurIPS 2022)
> >
> > [2] Nalisnick et al. "Do deep generative models know what they don’t know?" (ICLR 2019)
> >
> > [3] Nalisnick et al. "Detecting out-of-distribution inputs to deep generative models using typicality" (arXiv:1906.02994)

---

> ### Comment · Reviewer_KEPG · 2024-11-15
>
> I thank the authors for the insightful comments. I would like to apologize for the initially bad review, I think some of the points discussed here are perhaps under/miss-explained in the main paper. I agree that in high dimensions we might "miss" the highest likelihood point, however there is an important distinction between *high likelihood points* and *high probability regions*. In many parts of the paper we keep talking about sampling high probability  (some examples are line 107, the whole high probability seeking ODE section and the conclusion of the paper in line 533).
>
> As it might be true that some deep high likelihood regions are avoided, these are low probability. While other regions with a more "standard" likelihood (meaning that is neither low or the highest (the set up considered here)) are actually high probability. So diffusion models still sample from high probability regions, while avoiding the low probability regions. This can be seen for instance in the GMM case you provided, if we make a large enough ball around the heavy mode then many samples would be in this region.
>
> I think it is important to have these explained correctly in order to avoid these confusions.
>
> Could you please provide a full table detailing $\mathbb{E}_{x}[ -\log(p_0(x)))]$ where $-\log(p_0(x))$ is measured using the PF ode for the following 3 cases:
> 1. x is sampled using the traditional PF ODE
> 2. x is sampled using the SDE
> 3. x is sampled using your HP-ODE (it would be nice if we get several values of t, but at least one would be nice)
> Additionally it would be good to add what happens when the measurement is done using theorem 1. It would also be nice to have the variances. This experiment is very important to have, because in the current set up we are only getting to see a single sample. (edit: for this table bits per dimension is also good, i.e. NLL/dim)
>
> For instance:
> - Figure 7 just makes a claim without detailing any of the values and the result only apply for the 3 images there presented
> - Figure 8 doesn't provide any details of how many samples were used to compute these estimates. I suspect the results are correct but this information is missing
> - In Figure 9 we only get to see the samples, but no information about their likelihoods or how they compare with other samples is provided
>
> Intuitively the results here presented can make sense, but no quantitative evidence is provided and this is a *big downside* of this work. The theoretical insights are very interesting, however since assumptions were made we need to properly show that these resulted in the claimed results. If these two are resolved I would be willing to *significantly* increase my score.

---

> ### Author Response · Authors · 2024-11-16
> **Response**
>
> We thank the Reviewer for their engagement in the discussion and valuable feedback!
>
> 1. *there is an important distinction between high likelihood points and high probability regions*
>
> We agree. Let us formalize this: For a random variable $X$ with probability **density of a point** $x$, $p_X(x)$, the **probability of a set** $A$ is $P_X(A)=\int_A p_X(x)d\mu(x)$, where $\mu$ is the Lebesgue measure on $\mathbb{R}^D$.
>
> We believe we have reached a consensus that there can exist a typical set $T$ and a high-density set $H$ such that $P_X(T) \approx 1$, $P_X(H)\approx 0$ and that $\sup\lbrace\log p_X(x)|x\in H\rbrace > \sup\lbrace\log p_X(x)|x\in T\rbrace$.
>
> In other words $H$ is low probability, but it contains points of larger density than all elements of $T$.
> What we study in the paper are exactly such sets $H$ (which we call high-density regions).
> The regular samples from the model belong to the typical set, which does not contain the highest density points, but is of course high-probability.
>
> Thank you for this comment as it helps to improve the clarity of the paper. We will change the wording and use "high-density regions" throughout the paper and change the name of Algorithm 1 to "high-density (HD) sampling".
>
> 2. *Could you please provide a full table detailing $\mathbb{E} [-\log p_0(x_0)]$*
>
> Below we provide the table with the values of $\mathbb{E} [-\log p_0(x_0)]$ (in bits-per-dim) for different models and sampling strategies. In all cases $\log p_0(x_0)$ was estimated using the PF-ODE. The values are mean ± one standard deviation. We see that **HD sampling generates samples with higher density than regular samples accross different models and values of the threshold parameter $t$**.
>
> |Model|Sampling|NLL (bpd)|
> |-|-|-|
> |CIFAR10-ML|PF-ODE|4.17 ± 0.49|
> ||Rev-SDE|4.44 ± 0.42|
> ||Rev-SDE(Th 1)|4.30 ± 0.41|
> ||HD(0.3T)|2.65 ± 0.62|
> ||HD(0.45T)|1.57 ± 0.44|
> ||HD(0.5T)|1.25 ± 0.38|
> ||HD(0.55T)|0.98 ± 0.36|
> ||HD(0.7T)|0.24 ± 0.27|
> ||||
> |CIFAR10-SQ|PF-ODE|4.55 ± 0.46|
> ||Rev-SDE|4.23 ± 0.42|
> ||Rev-SDE(Th 1)|4.16 ± 0.42|
> ||HD(0.3T)|2.74 ± 0.61|
> ||HD(0.45T)|1.61 ± 0.39|
> ||HD(0.5T)|1.37 ± 0.34|
> ||HD(0.55T)|1.15 ± 0.32|
> ||HD(0.7T)|0.46 ± 0.30|
> ||||
> |IMAGENET64 [3]|Original|3.16 ± 0.81|
> ||HD(0.0125T)|-2.10 ± 0.33|
> ||HD(0.05T)|-2.74 ± 0.38|
> ||||
> |FFHQ256 [2]|Original|1.01 ± 0.41|
> ||HD(0.5T)|-2.58 ± 0.65|
> ||HD(0.3T)|-4.01 ± 0.96|
> ||||
> |CHURCH256 [2]|Original|0.77 ± 0.23|
> ||HD(0.5T)|-0.66 ± 0.34|
> ||HD(0.3T)|-1.15 ± 0.15|
>
> where:
> * CIFAR10-ML - maximum likelihood training, i.e. unweighed ELBO
> * CIFAR10-SQ - (Sample Quality) weighted ELBO training as recommended in [1]
> * Rev-SDE(Th 1) - $\log p$ evaluated with Theorem 1. Note that it can only be used for Rev-SDE sampling.
> * "Original" sampling - same sampling procedure as in the corresponding paper, i.e. Predictor-Corrector sampling for FFHQ256 and CHURCH256 [2]; and stochastic Heun sampler for IMAGENET64 [3]. We chose original samplers to faithfully reproduce the "regular samples". Note that neither of these samplers are the Reverse SDE, so we could not use Theorem 1 to evaluate $\log p_0$.
> * For CIFAR10 models we used 1024 samples in each experiment. For the remaining models we used 192 samples and only two values of $t$ for each model due to computational budget constraints.
> * Values of $t$ is HD sampling are in different ranges for different models, because they use different SDEs and different noise schedules. CIFAR10 uses VP SDE with linear log-SNR schedule, [2] uses VE SDE and exponential schedule and [3] uses VE SDE with $\sigma=t$.
>
> 3. *Figure 8 doesn't provide any details of how many samples were used*
>
> To generate Figure 8, we used 192 samples and for each created 7 versions by applying different strengths of blur ($\sigma \in \lbrace 0, 1, 2, 5, 10, 20, 50\rbrace$) so in total 192 * 7=1344 samples were used. For every image, $\log p_0$ was estimated with PF-ODE.
>
> The distributions of $\log p_0$ for different blur levels are on the right of Figure 10.
>
> 4. *In Figure 9 we only get to see the samples, but no information about their likelihoods or how they compare with other samples is provided*
>
> On the right-hand side of Figure 9 we show the plots of distributions of $\log p_0$ for all models, for regular samples and for HD for the two presented values of $t$ for each model.
> These density plots were based on $\log p_0$ evaluated on 192 samples. The means and standard deviations are as in the table above.
>
> We thank the Reviewer again for your thorough review, and hope that our clarifications and these additional details have convincingly addressed your concerns. We will appreciate if the same can be reflected in your revised score. Many thanks!
>
> ---
>
> [1] Kingma et al. "Understanding Diffusion Objectives as the ELBO with Simple Data Augmentation" (NeurIPS 2024)
>
> [2] Song et al. "Score-Based Generative Modeling through Stochastic Differential Equations" (ICLR 2021)
>
> [3] Karras et al. "Elucidating the Design Space of Diffusion-Based Generative Models" (NeurIPS 2022)

---

> > ### Comment · Reviewer_KEPG · 2024-11-16
> >
> > Dear Authors, I thank you a lot for providing this evidence and for confirming our understanding of the conclusions in this paper. I have increased my score. I believe this results should be included in the paper as the main piece of evidence of your claims. Given these corrections I think its a good paper. As a final comment I think you should release the details on the hyperparmeters used for sampling for different datasets and models to provide the readers with all the details. Thanks again for clarifying my questions and running the necessary experiments

---

> > > ### Author Response · Authors · 2024-11-17
> > > **Response**
> > >
> > > We thank the Reviewer for their continued engagement and an increased score! We have uploaded an updated paper with the suggested changes. Specifically,
> > > * We now consistently use "high-density" instead of "high-probability" in the text, abbreviations (HP->HD), and in the figures (e.g. Figure 7);
> > > * We added a new section in the appendix: Appendix M: *Cifar Models Hyperparameters* and refer to it in the main text when we introduce them in section 3.2;
> > > * We added a new section in the appendix, Appendix L: *Quantitative analysis of likelihoods of samples generated with Algorithm 1*, where we added the table of average likelihoods from this discussion. We explained that all are evaluated using the PF-ODE to ensure a fair comparison. We also explained the different sampling strategies for different baselines.
> > > * We added details in the text specifying the numbers of samples used to generate Figures 7, 8, 9, and 10.
> > >
> > > Thank you again for your thoughtful review and for acknowledging the strengths of our paper. We have carefully addressed all your concerns and incorporated your suggestions into the revised version. Considering these improvements and your positive assessment that it is a good paper, we kindly request reconsideration of the current score, as it suggests the work does not meet the acceptance threshold.

---

> > > > ### Comment · Reviewer_KEPG · 2024-11-25
> > > >
> > > > Dear authors upon careful consideration and re reading the new manuscript. Considering the efforts to make the experimental and theoretical paper as complete as possible, the new insights on the density landscape and how it relates to the amount of information in an image. I decided to increase my score, thanks for addressing the reviewers concerns

---

> > > > > ### Author Response · Authors · 2024-11-25
> > > > >
> > > > > Dear Reviewer,
> > > > >
> > > > > Thank you for your continued engagement in the review process and for contributing to this discussion. We sincerely appreciate your valuable suggestions and are grateful for your increased support of this work. Once again, thank you for your time and effort.

---

### Official Review · Reviewer_nSFN · 2024-11-01

**Soundness:** 4
**Presentation:** 4
**Contribution:** 3
**Rating:** 8
**Confidence:** 4

**Summary:**

The authors offer a theoretical framework for estimating the log probability of samples that follow SDE in diffusion models. They develop novel forward and reverse augmented dynamics that estimates log probability not only for the case where $\nabla \log p_{t}(x)$ is known precisely but for the case where the score function is known. Then, the authors provide novel upper and lower bounds on the bias of the proposed estimator and analyze them on the experimental data. Lastly, the authors apply their proposed theory to analyze the diffusion probability landscape. They propose a simple yet effective way to generate samples with the highest log probability and show that such samples are unrealistic and blurry.

**Strengths:**

* The authors develop a novel theory of augmented SDEs. They provided a clear and detailed derivation and coupled it with the bias estimation
* Landscape analysis gives valuable insight into the structure of high-probability samples and provides a theoretical justification for the known fact that distorted images tend to have a higher likelihood
* The paper is well-structured and easy to read

**Weaknesses:**

1. It isn't clear whether analysis from Section 5 can lead to the creation of better stochastic samplers or improve the quality of image generation. Overall practical implications of this work are quite poor
2. There is no intuition behind observations from Figure 4. More precisely, what does it mean that the model optimized for sample quality yields a smaller difference between $p_{0}^{ODE}$ and $p_{0}^{SDE}$?
3. Experiments were conducted on rather small and outdated diffusion models. It would greatly improve the scope of the work if the experiments were performed on the frontier models

**Questions:**

1. Have the authors thought about how the proposed theory can be used to improve the quality of image generation?
2. Can we use the proposed estimation for the log probability to evaluate model quality the same way as [1] (Table 2)? Will there be any difference between SDE and ODE estimation? Which one is better to compare different models?

[1] Yang Song, Jascha Sohl-Dickstein, Diederik P Kingma, Abhishek Kumar, Stefano Ermon, and Ben Poole. Score-based generative modeling through stochastic differential equations. arXiv preprint arXiv:2011.13456, 2020c.

---

> ### Author Response · Authors · 2024-11-19
> **Response**
>
> We thank the Reviewer for acknowledging the strengths of the paper and insightful questions. We address the raised concerns below.
>
> 1. *It isn't clear whether analysis from Section 5 can lead to the creation of better stochastic samplers or improve the quality of image generation. Overall practical implications of this work are quite poor [...] Have the authors thought about how the proposed theory can be used to improve the quality of image generation?*
>
> This is a valid point and we address it under "How can this study lead to improved sampling strategies?" in the Global Response.
>
> 2. *There is no intuition behind observations from Figure 4. More precisely, what does it mean that the model optimized for sample quality yields a smaller difference between $p_0^{\text{SDE}}$ and $p_0^{\text{ODE}}$?*
>
> Thank you for the opportunity to elaborate. As we mention in Lines 181-182, the SDE model and the ODE models are not equivalent in practice (because the score function is not learned perfectly). Therefore, for a learned score function $\mathbf{s}(t, x) \approx \nabla_x \log p_t(x)$, we have two different probability distributions $p_0^{\text{SDE}}$ and $p_0^{\text{ODE}}$.
>
> As we know from [1], if the score function was learned perfectly, i.e. $\mathbf{s}(t, x) = \nabla_x \log p_t(x)$, then we would have $p_0^{\text{SDE}} \equiv p_0^{\text{ODE}}$. Therefore, the discrepancy between $p_0^{\text{SDE}}$ and $p_0^{\text{ODE}}$ can be interpreted as a measure of self-consistency of the diffusion model.
>
> In section 3, we derive a tractable upper bound on this discrepancy as measured by $KL[p_0^{\text{SDE}} || p_0^{\text{ODE}}]$, i.e. $\mathcal{R}^U(\mathbf{s})$. Empirically, we found that $\mathcal{R}^U(\mathbf{s})$ is smaller for a model optimized for sample quality than for a model optimized with maximum likelihood training. This was a surprising discovery because it suggests that, the principled learning objective, i.e. maximum likelihood, leads to a model, which is worse in terms of the self-consistency as defined above. Conversely, the model optimized for sample quality, i.e. trained with a weighed ELBO, with the weighting function chosen empirically [3], leads to a model, which is more self-consistent.
>
> Furthermore, we believe that $\mathcal{R}^U(\mathbf{s})$ might become a useful tool for evaluating and comparing diffusion models. Especially, since FID is expensive to compute in practice and is sensitive to the choice of a random seed [4].
>
> 3. *It would greatly improve the scope of the work if the experiments were performed on the frontier models*
>
> Thank you for this suggestion. Based on this, we now extended the study with Stable Diffusion v2.1. Please see "Lack of analysis of more sophisticated models." in the Global Response for more details.
>
> 4. *Can we use the proposed estimation for the log probability to evaluate model quality the same way as [1] (Table 2)? Will there be any difference between SDE and ODE estimation? Which one is better to compare different models?*
>
> Very good question!
>
> First, our main novelty in likelihood estimation in stochastic SDEs are theorems 1 and 3. They show that during stochastic sampling, one can get an estimate of $\log p_0^{\text{SDE}}(x_0)$. However, **this only holds, when the model is generating the sample** and we can keep track of the likelihood along the stochastic trajectory. If we are presented with a sample $x_0$ (e.g. from a test set like in Table 2 in [1]), theorems [1] and [3] do not apply, because the model did not generate the sample.
>
> In that case, we would need to apply the augmented forward stochastic dynamics, which apply for any given input $x_0$ (in particular, it can be from some test set). However, in that case, we show (In Equation (13)) that this ends up being equivalent to the known formula for the lower bound on $\log p_0^{\text{SDE}}(x_0)$ [2].
>
> Finally, in that case, you can use both and the results most likely will be different for two reasons: 1) For $\log p_0^{\text{SDE}}(x_0)$ we only have a bound, whereas $\log p_0^{\text{ODE}}(x_0)$ we can estimate exactly; and 2) as we mention in the paper, $\log p_0^{\text{SDE}}$ and $\log p_0^{\text{ODE}}$ correspond to two different models, which are not equivalent in practice! In section 3 we even propose a tractable way to quantify the Kullback-Leibler divergence between them.
>
> Thank you again for a thorough review, we hope that we addressed all concerns and would kindly ask you to reconsider the score given the new evidence and clarifications.
>
> ---
>
> [1] Song et al. "Score-based generative modeling through stochastic differential equations". (ICLR 2021)
>
> [2] Song et al. "Maximum Likelihood Training of Score-Based Diffusion Models" (NeurIPS 2021)
>
> [3] Kingma et al. "Understanding Diffusion Objectives as the ELBO with Simple Data Augmentation" (NeurIPS 2024)
>
> [4] Karras et al. "Elucidating the Design Space of Diffusion-Based Generative Models" (NeurIPS 2022)

---

> > ### Author Response · Authors · 2024-11-26
> > **Gentle reminder**
> >
> > Dear Reviewer nSFN,
> >
> > As the deadline for the paper revision approaches, we wanted to follow up to ensure that our responses have adequately addressed your concerns. If there are any additional questions or points for discussion, we would be happy to address them promptly.
> >
> > We look forward to your feedback and sincerely appreciate your time and effort in reviewing our work.

---

> > > ### Author Response · Authors · 2024-11-30
> > > **Nearing the end of the discussion period**
> > >
> > > Dear Reviewer nSFN,
> > >
> > > With the discussion period coming to a close, we’d like to confirm whether we have successfully resolved your concerns. If the updates meet your expectations, would you consider revising your score to acknowledge the improved clarity and quality of the manuscript? If there are remaining issues, we are happy to engage further. For your convenience, we’ve outlined below the concerns you raised and our responses to them:
> > >
> > > * Intuition behind observations from Figure 4
> > >   * We discussed the findings from Figure 4 in the discussion above and hope that these have been helpful;
> > > * Experiments conducted on small and outdated diffusion models
> > >   * We have extended the study with Stable Diffusion v2.1;
> > > * Using our novel theory on likelihood estimation to evaluate models
> > >   * We hope that our answer above has been helpful and answered your question;
> > > * Practical insights from the High-density sampler
> > >   * Below, we paste our elaboration on this question from a discussion with Reviewer p2yw:
> > >
> > > 1. **The importance of identifying the need to avoid high-density regions.** While it might seem straightforward in hindsight, our finding that high-density regions should be avoided is, in itself, significant. This conclusion complements the low-density claims of [1] and opens up new questions about how likelihood relates to perceptual quality.
> > >
> > > 2. **Novel insights into the relationship between likelihood and perceptual quality.** In addition to this broader conclusion, we show in Figure 8 and elaborate on further in Appendix P that likelihood appears to correlate with the level of detail in an image. This novel finding links likelihood to perceptual quality—a connection that had not been fully explored in prior work—and offers a new lens through which to design sampling techniques.
> > >
> > > 3. **Practical challenges in implementation:** Even if the goal is conceptually clear, there are numerous practical hurdles to avoiding high-density regions:
> > >
> > >     * **Determining the optimal likelihood range:** Identifying the appropriate range of likelihood values for avoiding high-density regions is not a straightforward task and may vary based on the model or application context.
> > >     * **Class-dependent variations in likelihood:** For conditional generation, the likelihood distributions of different classes can vary significantly, complicating the determination of a universal or class-specific optimal range.
> > >     * **Efficient sampling within the optimal range:** Even once the ideal range is identified, efficiently sampling from this range is a separate and challenging problem. Regular sampling followed by filtering based on likelihood can introduce prohibitive computational costs. This suggests that injecting the optimal likelihood range directly into the design of the sampler itself might be a more practical solution. However, exploring this direction lies beyond the scope of our current work.
> > >
> > > We hope this clarification underscores the nuanced nature of the challenges associated with avoiding high-density regions and highlights how our findings provide both foundational insights and open questions for future exploration.
> > >
> > > ---
> > >
> > > [1] Karras et al. "Guiding a diffusion model with a bad version of itself" (NeurIPS 2024)

---

### Official Review · Reviewer_p2yw · 2024-11-08

**Soundness:** 2
**Presentation:** 2
**Contribution:** 2
**Rating:** 5
**Confidence:** 3

**Summary:**

This paper investigates the high-density regions of diffusion models, discovering that samples in these regions often appear as unrealistic cartoon-like or blurry images, despite the absence of such images in the training data. The authors propose a novel framework based on augmented stochastic differential equations (SDEs) to estimate the likelihoods of generated samples. This approach enables efficient, high-likelihood sampling without additional computational cost. The authors also introduce a high-probability sampling method that consistently yields higher-likelihood images than traditional sampling techniques, while their empirical analysis shows that images with less detail (e.g., blurred images) tend to achieve higher likelihood scores. The paper contributes to a better understanding of the diffusion model probability landscape and the relationship between likelihood and image quality.

**Strengths:**

The paper introduces a novel framework for estimating likelihoods within diffusion models using augmented stochastic differential equations (SDEs) and high-probability samplers. This is an important advancement because it allows the exploration of high-likelihood regions without increasing computational costs. By deriving density estimates through augmented SDEs, the authors provide a theoretically efficient approach to analyzing model outputs across noise levels, setting a foundation for future studies in likelihood-based generative modeling. The findings reveal that high-likelihood samples often resemble cartoonish or blurry images—even though such images are not present in the training data.

**Weaknesses:**

1. Although the paper sheds light on high-density regions and likelihood estimation, it lacks a clear discussion on how these findings could practically inform the design or improvement of diffusion models in applied settings. Without recommendations on balancing high-likelihood sampling with image quality, it’s challenging to draw valuable insights from the findings, particularly for practitioners focused on real-world applications.

2. The paper describes the emergence of cartoon-like images in high-likelihood samples but provides limited exploration into why or how this phenomenon occurs.

3. The experiments are primarily conducted on well-known datasets like CIFAR-10 and FFHQ-256. Expanding the study to other types of diffusion models or additional data domains would strengthen the generalizability of the findings. Additionally, clarifying any specific architectural limitations of the proposed high-probability sampler could benefit those aiming to extend the approach.

4. While the paper emphasizes the likelihood of generated samples, it does not address diversity within these high-likelihood outputs. The authors mention blurry and cartoon-like images, but they do not provide an assessment of diversity or how the high-likelihood sampling might affect the overall variance in generated outputs. It’s unclear if the method leads to mode collapse or reduces the richness of the model’s output space.

**Questions:**

1. Does the high-likelihood sampling technique impact the diversity of generated samples, potentially leading to mode collapse or other reductions in output variability?

2. For applications that prioritize both high likelihood and visual quality, what adjustments to the high-probability sampler could mitigate the production of low-quality images?

---

> ### Author Response · Authors · 2024-11-19
> **Response**
>
> We thank the Reviewer for their insightful review! We address the raised concerns below.
>
> 1. *The paper describes the emergence of cartoon-like images in high-likelihood samples but provides limited exploration into why or how this phenomenon occurs.*
>
> Thank you for raising this point. Based on your question, we have now revised the manuscript and added Appendix P: "Why do cartoons and blurry images occupy high-density regions?", where we provide an explanation for why this behavior occurs.
> We argue that the diffusion model's estimate of the (negative) loglikelihood can be interpreted as a notion of the amount of detail in an image. Therefore highest-density regions will be occupied by images with the least amount of detail, such as cartoons or blurry images.
>
> 2. *Although the paper sheds light on high-density regions and likelihood estimation, it lacks a clear discussion on how these findings could practically inform the design or improvement of diffusion models in applied settings. Without recommendations on balancing high-likelihood sampling with image quality, it’s challenging to draw valuable insights from the findings, particularly for practitioners focused on real-world applications*
>
> This is a good point. We address this in the Global Response under "How can this study lead to improved sampling strategies". We propose that, while the high-density sampler does not produce high-quality samples, our findings, alongside [1], suggest that the highest-quality images may correspond to the lower range of "moderate" likelihood values. We leave a deeper investigation of this for future work.
>
> 3. *The experiments are primarily conducted on well-known datasets like CIFAR-10 and FFHQ-256. Expanding the study to other types of diffusion models or additional data domains would strengthen the generalizability of the findings.*
>
> Another good point! Based on your suggestion we extended the study to include the Stable Diffusion v2.1 model. Please see "Lack of analysis of more sophisticated models." section in the Global Response.
>
> 4. *Additionally, clarifying any specific architectural limitations of the proposed high-probability sampler could benefit those aiming to extend the approach.*
>
> Good question. We have not encountered any architectural limitations. The qualitative analysis shows similar phenomena for different types of SDEs and noise schedules in Section 5. Furthermore, the results now extend to the Stable Diffusion model, which not only is text-guided but also operates in the latent space as opposed to the pixel space like the other baselines.
>
> 5. *While the paper emphasizes the likelihood of generated samples, it does not address diversity within these high-likelihood outputs.*
>
> Thank you for drawing attention to this. We provide some intuition regarding this in Figure 6, where we show that the threshold parameter $t$ in the high-density sampler allows for controlling the diversity/likelihood tradeoff. In terms of real-world data, a similar observation can be made. The higher the $t$, the lower the diversity of the generated outputs. This is caused by the fact that the highest-density samples have the least degrees of freedom (we discuss this in more detail in Appendix P) of the revised manuscript.
>
> Considering the two extremes: when $t \approx 0$, then HD sampling coincides with regular sampling (whether it's PF-ODE or Rev SDE). Then the variability is the same as of the base diffusion model. On the other hand, when $t \approx T$, then as can be seen in the bottom row of Figure 1, the outputs are monochromatic patches, which have significantly less variation than natural images.
>
> 6. *For applications that prioritize both high likelihood and visual quality, what adjustments to the high-probability sampler could mitigate the production of low-quality images?*
>
> This is a very interesting question. As we show in Section 5 and now elaborate on in the Global Response under "How can this study lead to improved sampling strategies?", we argue that it might be impossible to have both. This is caused by the fact that the highest likelihoods are occupied by images with the least detail. We leave the exploration of optimization of perceptual quality from the perspective of likelihood for future work.
>
> We thank the Reviewer again for their suggestions and questions. We hope that we have addressed all concerns and that you will consider increasing your score. For a full summary of updates in the manuscript, please refer to the Global Response. We remain dedicated to addressing any further concerns, questions, or suggestions you may have.
>
> ---
>
> [1] Karras et al. "Guiding a diffusion model with a bad version of itself" (NeurIPS 2024)

---

> > ### Comment · Reviewer_p2yw · 2024-11-25
> >
> > Thank you for your response. However, I feel it does not directly address the question of how your findings could lead to the design of improved samplers. While your study presents a practical high-density sampler that consistently generates images with higher likelihood than usual samplers, the actionable strategies for improving sampler quality—particularly to address the real-world challenge of enhancing perceptual quality—remain unclear. This is a critical issue that I believe most applications prioritize. Could you elaborate on how your observations might inform the design of future samplers, beyond simply avoiding high-density regions?

---

> ### Author Response · Authors · 2024-11-25
>
> Thank you for your thoughtful comments and for emphasizing this important point. We appreciate the opportunity to clarify our findings and their implications.
>
> The main takeaway from Section 5 of our paper is that, across different diffusion SDEs and models — now also including the latent diffusion model (Stable Diffusion v2.1) as per your suggestion — we consistently observe that high-density regions contain unrealistic images. This finding is particularly surprising in light of [1], which attributes the success of diffusion models to their ability to avoid low-density regions.
>
> Combining these observations — the insights from [1], our findings, and now also the explanation of this phenomenon inspired by your question (Appendix P in the revised manuscript) — we hypothesize that the highest-quality samples likely occupy regions of moderate likelihood (neither too low nor too high). Thus, samplers explicitly targeting these regions may produce the best results. However, we do not propose a specific solution, as addressing this challenge is nontrivial and requires an additional in-depth investigation that we leave for future work.
>
> While the high-density sampler does not directly lead to high-quality sampling, it was crucial in making these observations, which we believe will ultimately inform the design of better samplers.
>
> Finally, **we wish to emphasize that the high-density sampler and the empirical findings in Section 5 are only part of this work's contributions.** Our theoretical developments in Sections 2 and 3, along with the practical computational tool for estimating likelihood in stochastic diffusion models, offer broader potential applications. For instance, as noted in lines 156–157, they could enable importance sampling without additional computational cost.
>
> We hope that this addresses your remaining concern and that you will consider stronger support for this work. Thank you again for your feedback and for the opportunity to expand on this aspect of our study.
>
> ---
>
> [1] Karras et al. "Guiding a diffusion model with a bad version of itself" (NeurIPS 2024)

---

> > ### Author Response · Authors · 2024-11-26
> >
> > We would also like to point out that "simply avoiding high-density regions" is not as simple as it may seem. While the phrase might suggest straightforward implementation, our findings reveal that this task involves not only conceptual insights but also significant practical challenges, as outlined below:
> >
> > 1. **The importance of identifying the need to avoid high-density regions.** While it might seem straightforward in hindsight, our finding that high-density regions should be avoided is, in itself, significant. This conclusion complements the low-density claims of [1] and opens up new questions about how likelihood relates to perceptual quality.
> >
> > 2. **Novel insights into the relationship between likelihood and perceptual quality.** In addition to this broader conclusion, we show in Figure 8 and elaborate on further in Appendix P that likelihood appears to correlate with the level of detail in an image. This novel finding links likelihood to perceptual quality—a connection that had not been fully explored in prior work—and offers a new lens through which to design sampling techniques.
> >
> > 3. **Practical challenges in implementation:** Even if the goal is conceptually clear, there are numerous practical hurdles to avoiding high-density regions:
> >
> >     * **Determining the optimal likelihood range:** Identifying the appropriate range of likelihood values for avoiding high-density regions is not a straightforward task and may vary based on the model or application context.
> >     * **Class-dependent variations in likelihood:** For conditional generation, the likelihood distributions of different classes can vary significantly, complicating the determination of a universal or class-specific optimal range.
> >     * **Efficient sampling within the optimal range:** Even once the ideal range is identified, efficiently sampling from this range is a separate and challenging problem. Regular sampling followed by filtering based on likelihood can introduce prohibitive computational costs. This suggests that injecting the optimal likelihood range directly into the design of the sampler itself might be a more practical solution. However, exploring this direction lies beyond the scope of our current work.
> >
> > We hope this clarification underscores the nuanced nature of the challenges associated with avoiding high-density regions and highlights how our findings provide both foundational insights and open questions for future exploration.
> >
> > ---
> >
> > [1] Karras et al. "Guiding a diffusion model with a bad version of itself" (NeurIPS 2024)

---

> > > ### Author Response · Authors · 2024-11-30
> > > **Nearing the end of the discussion period**
> > >
> > > Dear Reviewer p2yw,
> > >
> > > As we near the conclusion of the discussion period, we wanted to check if we have sufficiently addressed your concerns. If so, would you be open to updating your score to reflect the enhanced clarity and quality of the manuscript? If there are still unresolved issues, we are more than willing to discuss them further. For your reference, we’ve summarized below the concerns you highlighted and the steps we’ve taken to address them:
> > >
> > > * Practical insights from the high-density sampler
> > >   * We have provided an elaborate discussion above on our insights and laid down the open research questions;
> > > * Explanation of cartoons and burry images phenomenon
> > >   * Appendix P - we have now included a discussion explaining this phenomenon;
> > > * Study limited to simple diffusion models
> > >   * Appendix O - we have extended the study with a latent diffusion model: Stable Diffusion v2.1;
> > > * Lack of discussion on architectural limitations
> > >   * We discussed above that we did not encounter any;
> > > * Diversity of high-density samples
> > >   * We have discussed above how the choice of the threshold parameter $t$ controls the diversity of the samples.

---

### Official Review · Reviewer_NhXG · 2024-11-08

**Soundness:** 3
**Presentation:** 3
**Contribution:** 3
**Rating:** 6
**Confidence:** 3

**Summary:**

This paper introduces new augmented SDE for likelihood estimation in SDE sampling of diffusion models. Moreover, a novel theoretical mode-tracking approach is proposed in order to locate the exact mode of the generative distribution and introduce a high-probability sampler capable of generating samples with higher likelihood than all the other samples, under theoretically proved guarantees.
Finally, the main findings of the proposes analysis lies in the discovery of Cartoon-like images that lies in high-likelihood regions, even if the model has never been trained with images of the same style.

**Strengths:**

- The paper is well written and it develops new theoretical insights, especially the augmented SDE for tracking likelihood evolution are innovative and pratical since they don't introduce any additional computational cost.

- The proposed high-probability sampler is a nice tool to generate high-likelihood samples that are not discoverable by traditional sampling techniques

- The paper analize the diffusion probability landscape finding analyzing the different images in high-likelihood regions, experimentally demonstrating that the proposed theoretical tools can be used to perform analyisis of this landscape.

**Weaknesses:**

- The proposed mode-tracking approach has a very high computational cost and this is not clearly discussed. The discussion on how this mode-tracking approach scale and its computational limitations should be discussed and quantified

- The high-likelihood samples discovered by the proposed analysis does not seem to have a real practical advantage, being cartoon drawings or blurry images. It is not discussed how these insight can be leveraged to improve sample generation strategies or how these high-density samples can be practically useful.

- The limitations of the work are not discussed at all, a limitations section covering all the main potential limitations should be added.

- The work does not report any implementation details, code and implementation are not submitted in the supplementary materials. It would be important to openly release the contribution upon acceptance to ensure reproducibility and improve transparency.

- The analysis of high-density regions is done on small and simple diffusion models trained on restricted datasets with limited variability. Moreover the selected models and study is done on uncoditional sampling without any guidance from text. In my opinion it would be valuable to explore real-world and more complex diffusion models (such as SD, Flux etc), especially focusing on the impact of text.

**Questions:**

- The related work does not take into consideration a very relevant paper "Null-text Guidance in Diffusion Models is
Secretly a Cartoon-style Creator", Zhao et al which is not discussed. In particular it would be relevant to highlight some connection and insights with this previous related work.

- See weaknesses

---

> ### Author Response · Authors · 2024-11-19
> **Response**
>
> We thank the Reviewer for their thorough review and support of this work! We address the raised concerns below.
>
> 1. *The proposed mode-tracking approach has a very high computational cost and this is not clearly discussed. The discussion on how this mode-tracking approach scale and its computational limitations should be discussed and quantified*
>
> This is a good point, thank you for bringing our attention to this. In the revised version of the paper, we added Appendix K: "Cost of mode-tracking", where we discuss in detail how the mode-tracking ODE scales and why it is prohibitively expensive, especially in larger dimensions. For example, on CIFAR10 dataset, mode tracking ODE is at least 6000x times more expensive to evaluate than the PF-ODE (or HD-ODE). On 256x256 images, it becomes 400000x more expensive than PF-ODE (or HD-ODE).
>
> 2. *The high-likelihood samples discovered by the proposed analysis does not seem to have a real practical advantage.*
>
> This is a good point that we address in the Global Response under "How can this study lead to improved sampling strategies?". We argue, that while the high-density sampler does not produce high-quality samples, our findings together with [1] suggest that the highest-quality images may have likelihoods in the lower end of "moderate" likelihood values. We leave this exploration for future work.
>
> 3. *The limitations of the work are not discussed at all, a limitations section covering all the main potential limitations should be added.*
>
> Thank you for pointing this out. We have now added the Limitations section in Appendix R: "Limitations" and refer to it in the conclusion. The main limitations are the inherent limitations of stochastic sampling as compared to deterministic carry over to likelihood estimation; and the impracticality of the exact mode-tracking ODE for general, non-Gaussian distributions. However, as our experiments suggest, we can dispense with the computationally expensive term to get significantly higher-likelihood samples than regular ones.
>
> 4. *The work does not report any implementation details, code and implementation are not submitted in the supplementary materials. It would be important to openly release the contribution upon acceptance to ensure reproducibility and improve transparency.*
>
> In the revised version of the paper, we added appendices M: "CIFAR models hyperparameters" and N: "Quantitative analysis of likelihoods of samples generated with algorithm 1", where we provide all the hyperparameters we used for our CIFAR10 models and other baselines used. Specifically, the CIFAR models we trained from scratch, whereas for the other baselines: ImageNet64, FFHQ256, Churches256 and now Stable Diffusion, we used the original implementations with default sampling parameters. We implemented high-density sampling on top of each of them.
> We are committed to releasing the code of our experiments upon acceptance.
>
> 5. *The analysis of high-density regions is done on small and simple diffusion models trained on restricted datasets with limited variability. Moreover the selected models and study is done on uncoditional sampling without any guidance from text. In my opinion it would be valuable to explore real-world and more complex diffusion models (such as SD, Flux etc), especially focusing on the impact of text.*
>
> Thank you for this suggestion. We address this in the Global Response under "Lack of analysis of more sophisticated models.". Based on your suggestion, we added an experiment with Stable Diffusion v2.1 and confirmed that even text-guided, latent diffusion models exhibit similar behavior, i.e. high-density samples correspond to blurry and cartoon-like images.
>
> 6. *The related work does not take into consideration a very relevant paper "Null-text Guidance in Diffusion Models is Secretly a Cartoon-style Creator", Zhao et al which is not discussed. In particular it would be relevant to highlight some connection and insights with this previous related work.*
>
> Thank you for pointing this out and we apologize for this omission. We have now included a discussion on the comparison with the paper you provided in Appendix Q "Cartoon generation" and we point to it in the related works section. The main difference between is in the motivation. Unlike existing models, our aim was not to build the best possible cartoon generator. We studied the high-density regions and discovered that they comprise of cartoon-like and blurry images.
>
> We would like to thank the Reviewer again for their thoughtful and constructive review. We hope that we have addressed all concerns and you will consider an even stronger support for this work. For a full summary of the updates in the manuscript, please refer to the Global Response. We are committed to engaging further in case you have any further concerns, questions, or suggestions.
>
> ---
>
> [1] Karras et al. "Guiding a diffusion model with a bad version of itself" (NeurIPS 2024)

---

> ### Comment · Reviewer_NhXG · 2024-11-22
> **Response to Authors**
>
> Dear Authors,
>
> Thanks a lot for your answers, I particularly appreciate the effort made with running additional experiments with SD2.1. I think this work is a nice analysis, toward a better understanding of the probability landscape of diffusion models.

---

> > ### Author Response · Authors · 2024-11-23
> > **Thank you**
> >
> > Dear Reviewer,
> >
> > Many thanks for your kind response, appreciating our efforts and acknowledging the significance of this work.
> >
> > We are grateful for your insightful review, thoughtful remarks, and constructive suggestions for additional analyses, which have significantly reinforced the merits of the work. We would greatly appreciate it if you could kindly consider revisiting your score to reflect the improved clarity and quality resulting from these enhancements. Thank you again for your time and feedback.

---

> > > ### Author Response · Authors · 2024-11-30
> > > **Nearing the end of the discussion period**
> > >
> > > Dear Reviewer NhXG,
> > >
> > > Since we are approaching the end of the discussion period we wanted to ask whether we have adequately addressed your concerns? If so, will you consider raising your score to reflect the improved clarity and quality of the manuscript? If not, we are happy to discuss further. Below we again summarize the concerns you raised and our attempts to address them:
> > >
> > > * Unclear cost of mode-tracking
> > >   * Appendix K with a detailed breakdown of the cost of mode-tracking;
> > > * Missing limitations
> > >   * Appendix R with a discussion on limitaions and a reference in the main text;
> > > * Missing implementation details
> > >   * Appendix M with implementation details, and working on full code release upon acceptance;
> > > * Missing more sophisticated models
> > >   * Appendix O with experiments on Stable Diffusion v2.1;
> > > * Missing discussion with related work on cartoon generation
> > >   * Appendix Q with the discussion and a reference in the main text.
> > > * Practical insights from the high-density sampler
> > >   * Below, we paste our elaboration on this question from a discussion with Reviewer p2yw:
> > >
> > > 1. **The importance of identifying the need to avoid high-density regions.** While it might seem straightforward in hindsight, our finding that high-density regions should be avoided is, in itself, significant. This conclusion complements the low-density claims of [1] and opens up new questions about how likelihood relates to perceptual quality.
> > >
> > > 2. **Novel insights into the relationship between likelihood and perceptual quality.** In addition to this broader conclusion, we show in Figure 8 and elaborate on further in Appendix P that likelihood appears to correlate with the level of detail in an image. This novel finding links likelihood to perceptual quality—a connection that had not been fully explored in prior work—and offers a new lens through which to design sampling techniques.
> > >
> > > 3. **Practical challenges in implementation:** Even if the goal is conceptually clear, there are numerous practical hurdles to avoiding high-density regions:
> > >
> > >     * **Determining the optimal likelihood range:** Identifying the appropriate range of likelihood values for avoiding high-density regions is not a straightforward task and may vary based on the model or application context.
> > >     * **Class-dependent variations in likelihood:** For conditional generation, the likelihood distributions of different classes can vary significantly, complicating the determination of a universal or class-specific optimal range.
> > >     * **Efficient sampling within the optimal range:** Even once the ideal range is identified, efficiently sampling from this range is a separate and challenging problem. Regular sampling followed by filtering based on likelihood can introduce prohibitive computational costs. This suggests that injecting the optimal likelihood range directly into the design of the sampler itself might be a more practical solution. However, exploring this direction lies beyond the scope of our current work.
> > >
> > > We hope this clarification underscores the nuanced nature of the challenges associated with avoiding high-density regions and highlights how our findings provide both foundational insights and open questions for future exploration.
> > >
> > > ---
> > >
> > > [1] Karras et al. "Guiding a diffusion model with a bad version of itself" (NeurIPS 2024)

---

### Author Response · Authors · 2024-11-19
**Global Response (1/2)**

We thank the reviewers for their insightful comments and suggestions to improve this work, and the area, program, and general chairs for their great service to the community.

**Positive Aspects.** The Reviewers variously acknowledged many strengths of the paper, including clear structure and good writing (NhXG, nSFN, KEPG); novel theoretical insights (NhXG, p2yw, nSFN, KEPG); analysis of the probability landscape (NhXG, p2yw, nSFN).

**Concerns and Questions.** The Reviewers also provided constructive feedback, especially regarding the practicality of the high-density sampler and the lack of evaluation on more sophisticated models. Below we summarize the main concerns.

### How can this study lead to improved sampling strategies?
In our paper, we developed an efficient method for generating high-density samples. Reviewers (NhXG, p2yw, nSFN) asked how our findings can be leveraged to design improved samplers. Indeed, the high-density sampler does not produce high-quality images. Its purpose was to study high-density regions and we concluded that such regions do not contain high-quality samples. We have also provided additional intuition for why it happens (Appendix P in the revised manuscript).

Our study of high-density regions was inspired by [1], where the authors attributed the success of diffusion models to their ability to avoid low-density regions. Our analysis in section 5 shows that for high-quality image generation, the highest densities should also be avoided. Therefore, in the quest to improve perceptual quality, one might need to target "moderate" densities and perhaps even on the lower end (given our new discussion section in Appendix P, where we argue that negative loglikelihood measures the amount of detail in an image). We leave this exciting research direction for future work.

**We also take this opportunity to emphasize that the high-density sampler is not the sole contribution of this work. The likelihood estimation tools we introduced in sections 2 and 3 have other potential applications.** As we mention in line 157, one can e.g. use stochastic sampling for importance sampling as we now show that it is possible to generate both a sample and its likelihood estimate for no extra cost.

### Lack of analysis of more sophisticated models.
Reviewers (NhXG, p2yw, nSFN) suggested that it would improve the quality of the study if we included more sophisticated models in our analysis. We would like to point out that in our study we experimented with diffusion models up to 256x256 resolution, which we believe are relatively large.

However, based on this suggestion, we included an experiment with Stable Diffusion v2.1 [2], which is a text-conditioned latent diffusion model generating images in 768x768 resolution. Interestingly, even though this is a text-guided diffusion model operating in the latent space, we observed similar behavior to the diffusion SDEs in the pixel space. Specifically, high-density samples decode to images that are blurry or exhibit cartoon-like features. Please see Appendix O in the revised manuscript for details.

### Updated manuscript.
We uploaded a revised version of the manuscript with addressed concerns from all Reviewers and below we summarize the changes made.

1. **Change in terminology: "high-probability" -> "high-density".** to avoid any confusion and emphasize that we refer to sets of points, which have high probability density.
2. **Model hyperparameters** - We added Appendix M: "CIFAR models hyperparameters", where we detail the hyperparameters and the training procedure used for CIFAR models in our experiments.
3. **Computational cost of mode-tracking** - We added K: "Cost of mode-tracking", where we discuss in detail the cost of evaluating the exact mode-tracking ODE introduced in Theorem 5.
    * Specifically, we show that evaluating a single step in solving the mode-tracking ODE scales worse than linearly with the dimension of the input $D$, which for CIFAR10 amounts to computation equivalent to over 6000 score function evaluations and to over 400000 score function evaluations for 256x256 data. In comparison, PF-ODE and HD-ODE both require a single score function evaluation per step.
4. **More details on the experiments** - We provide in the text details regarding sample sizes used in the generation of Figures 7, 8, 9, and 10.
5. **A quantitative analysis of HD sampling** - We added Appendix N: "Quantitative analysis of likelihoods of samples generated with algorithm 1", where provide a table containing $\mathbb{E}[-\log p_0(x_0)]$ for all baselines and different sampling strategies and different values of the threshold parameter $t$.
    * We reinforce our claims that the HD sampler indeed generates points with higher likelihoods than regular samples across different models.

---

> ### Author Response · Authors · 2024-11-19
> **General Response (2/2)**
>
> 6. **Additional experiments using Stable Diffusion** - We added Appendix O: "Stable Diffusion Samples", where we describe an experiment conducted with stable diffusion, which extends the score of the study to 1) latent diffusion models and 2) text-guided diffusion models.
>     * We found that even though the SDE governs the evolution of the latent representation of an image, we observe similar behavior as in pixel-space diffusion models. Specifically, we found that high-density samples decode to images, which are either blurry or exhibit cartoon-like features
> 7. **Explanation of cartoons and blurry images occupying the high-density regions** - We added Appendix P: "Why do cartoons and blurry images occupy high-density regions?", where we note the connection between local intrinsic dimension and diffusion model's likelihood estimation.
>     * We argue that the diffusion model's estimate of the (negative) log-likelihood can be interpreted as the amount of detail in an image. Therefore, since blurry and cartoon-like images have much less detail than regular images, they lie in high-density regions.
> 8. **Addition of cartoon generation methods to related work** - We added Appendix Q: "Cartoon generation", where we discuss the connection between our results and other cartoon generation methods.
>     * The main difference is the motivation. Unlike the existing cartoon-generation method, our aim was not to build the best possible cartoon generator. Rather, we set out to conduct a theoretical analysis of the high-density regions and made a surprising discovery that they consist mainly of cartoons and blurry images.
> 9. **Updated conclusion** - We added to the first part of the conclusion: "While [1] argued that avoiding low-density regions is crucial for the success of diffusion models, our analysis reveals that high-density regions should also be avoided in high-quality image generation."
>     * We added this to emphasize the takeaway from section 5.
> 10. **Limitations** - We added Appendix R, where we discuss the limitations of our work, which are split into two groups:
>     * stochastic likelihood estimation - the inherent limitations of stochastic sampling as compared to deterministic carry over to likelihood estimation;
>     * exact-mode tracking ODE holds only for linear SDEs and is too expensive to be used in practice.
>
> We thank the Reviewers again for their constructive feedback!
>
> ---
>
> [1] Karras et al. "Guiding a diffusion model with a bad version of itself" (NeurIPS 2024)
>
> [2] Rombach et al. "High-resolution image synthesis with latent diffusion models" (CVPR 2022)

---

### Meta-Review · Area_Chair_DA25 · 2024-12-22

**Metareview:**

This paper proposes a mode-tracking method to sample data from high-density regions from a diffusion model. To this end, a likelihood estimation method is proposed and theoretical guarantees are given. Empirically, this paper found that the high likelihood examples are more cartoon-like drawings or blurry images. The reviewers agree that the paper has novel technical and theoretical contributions, but the application of the proposed method is unclear. The authors have partially addressed this concern in the rebuttal. Overall, I think it is a solid paper and recommend accepting it to ICLR.

**Additional Comments On Reviewer Discussion:**

The main concerns raised by reviewers are 1)it is unclear how the proposed method lead to improved samplers, 2) the experiments are only done on vanilla diffusion models on pixels, not including those work on the latent space, such as stable diffusion, 3) the practical advantage of the proposed method is fully clear. The authors addressed these concerns in the rebuttal and state that the proposed work could suggest avoiding sampling high-density regions. Also, the authors included experiments on stable diffusion and found similar phenomena. Overall, after rebuttal and discussion, the concerns from reviewers are well addressed, but the practical application of this work might need further investigations in the future. I recommend acceptance mainly because of its solid theoretical foundations and novel method.

---

### Decision · Program_Chairs · 2025-01-22

Accept (Poster)